# Disentangling and Re-evaluating The Effectiveness of Graph Structure Learning For GNNs

## Abstract

Graph Structure Learning (GSL) has been widely adopted in the design of Graph Neural Networks (GNNs), with similarity-based graph learning emerging as the most popular approach for node classification. However, which component of GSL really enhances GNN performance remains underexplored. In this paper, we disentangle its effects and present a comprehensive analysis. Specifically, we propose a novel framework that can decompose GSL into three steps: (1) GSL bases (*i.e.,* processed node embeddings for construction) generation, (2) new graph construction, and (3) view fusion. Through empirical studies and theoretical analysis, we demonstrate that applying graph convolution to the newly constructed graphs does not increase the Mutual Information (MI) between node embeddings and labels. Our findings reveal that model performance is primarily driven by the quality of GSL bases rather than the graph construction methods. To validate them, we conduct extensive experiments with 450 GSL variants and benchmark them against GNN baselines within the same search space for GSL bases. Results show that similarity-based graph construction has negligible or even adverse impacts on GNN performance, while pre-trained GSL bases provide significant performance gains. These findings verify and confirm our analysis, underscoring the critical role of GSL bases and highlighting the need to simplify the other two GSL steps.

## 1   Introduction

Graph Neural Networks (GNNs) [17] are effective in capturing structural information from non-Euclidean data, which can be used in many applications such as recommendation [50, 49], telecommunication [29], bio-informatics [54, 12, 13], and social networks [34, 24]. However, conventional GNNs suffer from issues including heterophily [30, 31], over-squashing [5], adversarial attacks [15, 22], and missing or noisy structures [21, 28]. To address these issues, Graph Structure Learning (GSL), especially the similarity-based method that reconstructs or refines the original graph structures, has been widely used in enhancing GNN performance and robustness [60]. Even though GSL is believed to improve GNN performance, it introduces more hyperparameters and adds plenty of computational cost in both the construction process and the learning process. In addition, recent studies [39, 36] have shown that GSL methods cannot consistently outperform baseline GNNs with the same hyperparameter tuning strategy. Therefore, an in-depth analysis of the effectiveness and necessity of GSL is highly needed.

To have a detailed understanding of each component in GSL, we propose a new framework that can break down GSL into 3 steps: **(1) GSL Bases Generation.** GSL bases are the processed node embeddings that serve as inputs for the structure construction of new graphs. They are built by either graph-aware or graph-agnostic models with fixed or learnable parameters. **(2) Graph Structure Construction.** Based on the GSL bases, new structures are constructed with similarity-based [14, 38],

structural-based [55, 27], or optimization-based approaches [15] [1], followed by graph refinements. **(3) View Fusion.** To incorporate the original graph or combine multiple GSL-generated graphs, various view fusion strategies are applied, *e.g.,* late fusion [45], early fusion [22], or separation [28]. Compared with existing categorizations of GSL [40, 60, 61, 18] that mainly focus on **step (2)**, our proposed framework is more comprehensive, and is able to disentangle the effect of each component in GSL.

More specifically, for **step (1)**, we argue that a fair comparison between GSL-enhanced GNNs and traditional GNNs should be made using the same GSL bases. Previous GSL studies often enhance node inputs with additional information before graph construction, such as pre-trained node embeddings [6, 51] or structural embeddings [38, 57]. However, these enhancements are typically absent for the inputs of GNN baselines, leading to potentially biased and invalid evaluations. For **step (2)**, we examine the effectiveness of graph convolution operations with the similarity-based graphs. Our empirical and theoretical findings indicate that the Mutual Information (MI) between convolved node representations and labels does not increase after graph convolution. This suggests that the performance improvements observed in previous similarity-based GSL methods result from the processed GSL bases (*i.e.,*, enhanced node inputs) in step (1) rather than new graph construction in step (2)[2].

We conducted extensive experiments to validate our hypothesis. To thoroughly evaluate the performance of GSL-enhanced GNNs, we implemented these methods using six GNN backbones, five GSL bases, three GSL graph construction approaches, three view fusion methods, and two types of fusion strategies, resulting in 450 different GSL variants. The results demonstrate that, within the same search space of GSL bases, there are no significant performance differences between GSL-enhanced GNNs and the corresponding baseline GNNs on node classification tasks. In addition, the results show that the pre-trained GSL bases is the component which significantly enhance GNN performance on certain datasets. This aligns with our analysis and validate our claim. In summary, our main contributions are as follows:

- **Comprehensive GSL Framework:** In Section 3, we propose a novel framework that decomposes the GSL process into three steps. This decomposition provides a more comprehensive perspective than existing categorizations, offering valuable insights into the workings of GSL.
- **Empirical and Theoretical Analysis:** In Section 4, we present both empirical evidence and theoretical analysis demonstrating that the Mutual Information (MI) between node representations and labels does not increase after applying graph convolution on similarity-based GSL graphs. This finding suggests that similarity-based GSL methods may be unnecessary.
- **Fair Re-Evaluation of GSL:** In Section 5, we conduct a fair reassessment of GSL's impact on GNN performance. Our results highlight that GSL bases play a crucial role in improving GNN performance, while similarity-based graph construction has a negligible effect. Besides, we identify the key components for effective GSL, including pretrained GSL bases, parameter separation, and early fusion strategies.

## 2 Preliminary

**Graphs.** Suppose we have an undirected graph $\mathcal{G} = \{\mathcal{V}, \mathcal{E}\}$ with node set $\mathcal{V}$ and edge set $\mathcal{E}$. Let $\boldsymbol{Y} \in \mathbb{R}^{N \times C}$ denote the node labels and $\boldsymbol{X} \in \mathbb{R}^{N \times M}$ represent the node features, where $N$ is the number of nodes, $C$ is the number of classes, and $M$ is the number of features. The graph structure is represented by an adjacency matrix $\boldsymbol{A}$, where $\boldsymbol{A}_{u,v} = \boldsymbol{A}_{v,u} = 1$ indicates the existence of an edge $e_{uv}, e_{vu} \in \mathcal{E}$ between nodes $u$ and $v$. The normalized adjacency matrix is given by $\hat{\boldsymbol{A}} = \tilde{\boldsymbol{D}}^{-\frac{1}{2}} \tilde{\boldsymbol{A}} \tilde{\boldsymbol{D}}^{-\frac{1}{2}}$, where $\tilde{\boldsymbol{D}} = \boldsymbol{D} + \boldsymbol{I}_n$ and $\tilde{\boldsymbol{A}} = \boldsymbol{A} + \boldsymbol{I}_n$ represent the degree matrix and adjacency matrix with added self-loops. The neighbors of node $u$ is denoted as $\mathcal{N}_u = \{v | e_{uv} \in \mathcal{E}\}$. Graph Structure Learning (GSL) generates a new graph topology $\boldsymbol{A}'$, where the new neighbors of node $u$ are denoted as $\mathcal{N}'_u$. **Graph-aware models** $\mathcal{M}^{\mathcal{G}}$, such as Graph Convolutional Networks (GCN) [17], are powerful in extracting structural information in graphs by message aggregation or graph filters [35]. In contrast, **graph-agnostic models** $\mathcal{M}^{\neg\mathcal{G}}$, such as Multilayer Perceptrons (MLP), only use $\boldsymbol{X}$ without considering $\mathcal{G}$. For example, the updating process of node embeddings in GCN and MLP

---

[1]Note that most of the construction methods are similarity-based, which is the main focus of our paper.

[2]As step (3) fuses the results from step (2), if step (2) is ineffective, then step (3) will also be ineffective.

can be represented as $\boldsymbol{H}^l = \sigma(\hat{\boldsymbol{A}}\boldsymbol{H}^{l-1}\boldsymbol{W}^{l-1})$ and $\boldsymbol{H}^l = \sigma(\boldsymbol{H}^{l-1}\boldsymbol{W}^{l-1})$, respectively. Here, $\boldsymbol{H}^l$ and $\boldsymbol{W}^l$ are the node embeddings and weight matrix at the $l$-th layer, respectively, and $\sigma(\cdot)$ is an activation function.

**Graph Homophily.** The concept of homophily originates from social network analysis and is defined as the tendency of individuals to connect with others who have similar characteristics [16]. A higher level of graph homophily makes the topological information of each node more informative, thereby improving the performance of graph-aware models $\mathcal{M}^{\mathcal{G}}$ [32, 33, 56]. Commonly used homophily metrics include edge homophily [2, 59] and node homophily [38]:

$$h_{\text{edge}}(\mathcal{G}, \boldsymbol{Y}) = \frac{\left|\{e_{uv} \mid e_{uv} \in \mathcal{E}, Y_u = Y_v\}\right|}{|\mathcal{E}|} \tag{1}$$

$$h_{\text{node}}(\mathcal{G}, \boldsymbol{Y}) = \frac{1}{|\mathcal{V}|} \sum_{v \in \mathcal{V}} \frac{\left|\{u \mid u \in \mathcal{N}_v, Y_u = Y_v\}\right|}{|\mathcal{N}_v|} \tag{2}$$

**Mutual Information.** Mutual Information quantifies the amount of information obtained about one random variable given another variable [1]. The mutual information between variable $X$ and $Y$ can be expressed as:

$$I(\boldsymbol{X}; \boldsymbol{Y}) = \sum_{y \in \mathcal{Y}} \sum_{x \in \mathcal{X}} p(x, y) \, \log \frac{p(x, y)}{p(x)p(y)} \tag{3}$$

where $p(x, y)$ is joint probability, and $p(x)$ and $p(y)$ are marginal probability.

Mutual information could be used to analyze the quality of input features by measuring how much information the inputs $\boldsymbol{X}$ retain about the outputs $\boldsymbol{Y}$. However, in graphs under the task of node classification, the mutual information between a discrete variable $\boldsymbol{Y}$ and a continuous variable $\boldsymbol{X}$ cannot be directly measured by Eq. (3). Therefore, in this paper, we measure the mutual information $I(\boldsymbol{X}; \boldsymbol{Y})$ based on entropy estimation from k-nearest neighbors distances following [19, 20, 41].

# 3   Graph Structure Learning

Existing studies and evaluations of GSL mainly focus on the structure construction method. However, through extensive literature review, we find that it only constitutes one step of GSL [40, 60, 61]. To comprehensively understand and disentangle GSL for GNN learning, we propose a new framework. As shown in Figure 1, our framework includes three steps: GSL bases generation, new structure construction, and view fusion. Then, the whole pipeline of GSL is: First, GSL bases $\boldsymbol{B}$ is constructed based on node features $\boldsymbol{X}$ (and input graphs $\mathcal{G}$); Then, new graph structures $\mathcal{G}'$ are constructed with the GSL bases; At last, the information from $\mathcal{G}'$ (sometimes with multiple views) and original graph $\mathcal{G}$ are combined with different view fusion strategies for GNN training. We will introduce each component in the following subsections.[3]

## 3.1   GSL Bases

The GSL bases $\boldsymbol{B}$ is defined as the pre-processed node embeddings used for new structure construction. The quality of the GSL bases plays a crucial role for the graph construction step. For node classification tasks, an effective GSL bases $\boldsymbol{B}$ should exhibit consistency among intra-class nodes, as shown in Figure 2 (left). The construction of $\boldsymbol{B}$ can be categorized into non-parametric approaches [8, 38, 63], which generate fixed $\boldsymbol{B}$, and parametric approaches [15, 6, 53], where $\boldsymbol{B}$ is learnable during training. The construction of $\boldsymbol{B}$ can also be categorized into graph-agnostic [8, 15, 63] and graph-aware approaches [38, 53, 45], based on whether the original graph information will be contained in $\boldsymbol{B}$. Combining these two perspectives, in Figure 1, we show the diagrams of four types of bases: $\boldsymbol{B} = \boldsymbol{X}$, $\boldsymbol{B} = (\boldsymbol{A})^k \boldsymbol{X}$, $\boldsymbol{B} = \text{MLP}(\boldsymbol{X})$, and $\boldsymbol{B} = \text{GNN}(\boldsymbol{X}, \boldsymbol{A})$.

## 3.2   New Structure Construction

The construction of the new structure $\mathcal{G}'$, based on $\boldsymbol{B}$, is a key element of GSL. Based on relation extraction methods, the construction of $\mathcal{G}'$ can be categorized into similarity-based [14, 38, 23], structure-based [55, 27, 63], and parametric optimization-based [15, 28, 25] approaches. Similarity-based method is the most prevalent one, and the choice of similarity measurement, such as k-Nearest

---

[3]Please refer to Appendix A for a more detailed discussion of the representative GSL methods within our proposed GSL framework.

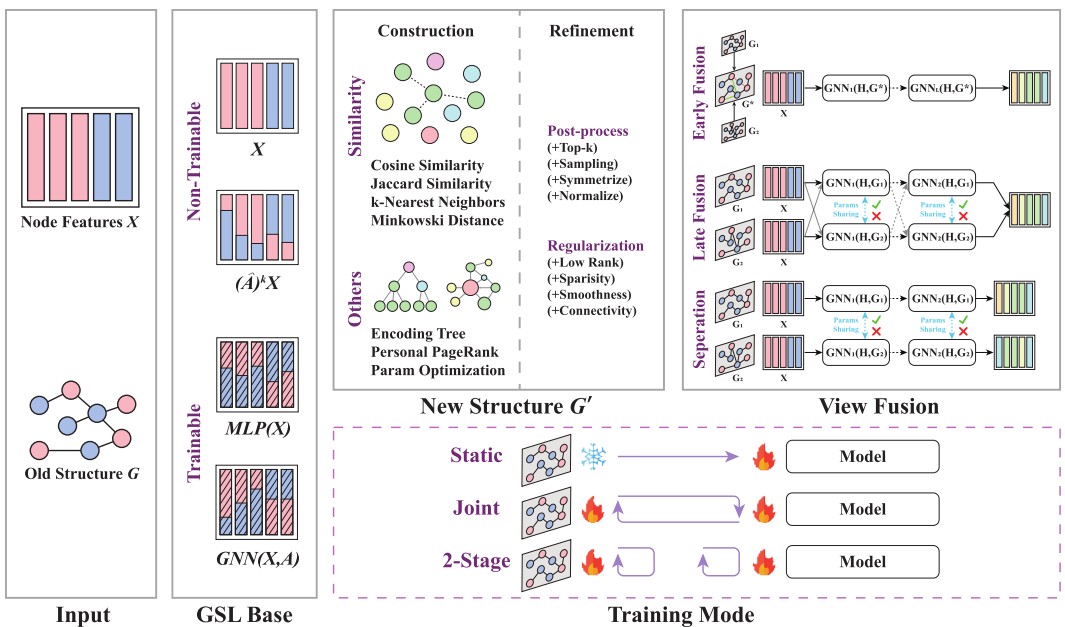

Figure 1: Our proposed GSL framework consists of three steps: GSL base generation, new structure construction, and view fusion.

Neighbors [8], cosine similarity [6], or Minkowski distance [28], plays a critical role in the quality of the reconstructed graphs. However, the initial $\mathcal{G}'$ produced by these methods often results in a coarse graph structure, which may not be optimal for GNN training. Thus, further refinements are often necessary, such as sampling [55, 22, 28], symmetrization [53, 7, 28], normalization [14, 55, 28], or applying graph regularization [14, 15, 25].

### 3.3 View Fusion

For GSL methods which have already implicitly fused the information from the original graph structure $\mathcal{G}$ into the reconstructed structure $\mathcal{G}'$ [14, 7, 63], further view fusion is unnecessary. However, for other approaches, the fusion of information from $\mathcal{G}$ and $\mathcal{G}'$ is crucial. Based on the fusion stage, methods can be classified as early fusion [22, 21, 27], late fusion [45, 28, 57], and separation [28]. Early fusion, often seen as "graph editing", modifies $\mathcal{G}$ by adding or removing edges with $\mathcal{G}'$ before training. Late fusion keeps both views as input, fusing node embeddings either at each layer or in the final layer. Separation methods, typically paired with contrastive learning, maintain multiple views without embedding fusion during GNN training. Additionally, view fusion methods can be further distinguished by whether they involve parameter sharing across layers during training.

### 3.4 Training Mode

In addition to the previous three steps, the training mode of $\mathcal{G}'$ plays a crucial role in GSL and can be categorized into static, joint, and 2-stage approaches. Most methods [15, 25, 51] use joint training where $\mathcal{G}'$ and model parameters are optimized simultaneously. In contrast, some methods [8, 45, 27] follow a 2-stage mode, iteratively updating $\mathcal{G}'$ and model parameters. While dynamic updates offer better flexibility for learning complex structures through parameter optimization, they also significantly increase computational complexity, especially during the bases and graph construction steps. To address this, other methods [43, 23, 57] opt for a static $\mathcal{G}'$ during training. Although this fixed structure may limit performance, it avoids the time-consuming process of frequent graph updates.

## 4 Effectiveness of Graph Structure Learning

Based on the framework built in the previous section, in this section, we question the necessity of similarity-based graph construction methods with theoretical analysis and extensive experiments. We introduce the motivation with an example in Section 4.1. We then explore the impact of GSL on

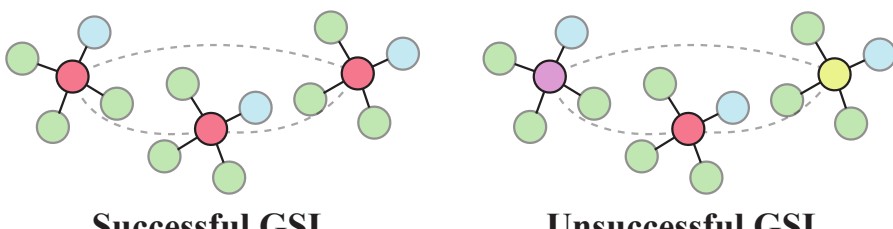

**Successful GSL**     **Unsuccessful GSL**

Figure 2: Examples of GSL that use the similarity of neighbor distribution as GSL bases for graph construction. The color of nodes indicates their labels. **Left**: new edges successfully connect intra-class (red) nodes, which share similar GSL bases (neighborhood pattern with 3 green nodes and 1 blue node). **Right**: new edges connect inter-class nodes, resulting in an unsuccessful GSL.

GNN performance through empirical observations in Section 4.2 and theoretical analysis in Section 4.3. Finally, the time complexity of GSL is discussed in Section 4.4.

## 4.1 Motivation

We revisit the effectiveness of GSL by the examples shown in Figure 2, where we use neighborhood distributions as GSL bases. Suppose a successful new edge is the one that connect intra-class nodes (Figure 2 (Left)), *i.e.,* homophilic connection [31]; and an unsuccessful edge connects inter-class nodes (Figure 2 (Right)), *i.e.,* heterophilic edge. We can see that successful and unsuccessful connections both follow node similarity principle to build edges. In other words, the same construction method can lead to totally different outcomes. Instead, the main difference comes from the GSL bases: the left example has high-quality bases, where intra-class nodes share consistent representations; however, the right example has low-quality bases, where inter-class nodes have similar embeddings.

On the other hand, this example also points out an awkward situation for GSL: when we have low-quality bases, GSL cannot work well; when we have high-quality bases, it means that the bases themselves can already provide sufficiently informative and distinguishable node embeddings for classification, and therefore, new graph construction may still be unnecessary. To further explore the effectiveness of the new graph construction step in GSL, we conduct empirical and theoretical analyses in the following subsections.

## 4.2 Empirical Observations on Synthetic Graphs

In this section, we investigate how GSL bases and the graph reconstruction methods influence GNN performance through experiments on synthetic graphs. The graph generation process is as follows.

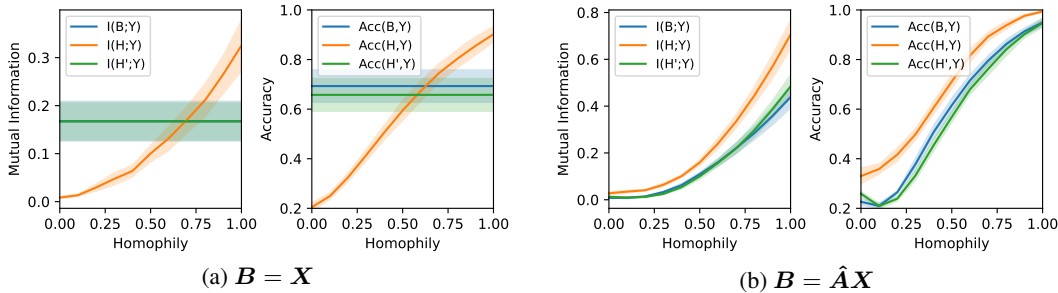

(a) $\boldsymbol{B} = \boldsymbol{X}$          (b) $\boldsymbol{B} = \hat{\boldsymbol{A}}\boldsymbol{X}$

Figure 3: Mutual information and accuracy of node classification for GSL bases $\boldsymbol{B}$, convoluted bases $\boldsymbol{H} = \hat{\boldsymbol{A}}\boldsymbol{B}$, new graph convoluted bases $\boldsymbol{H'} = \hat{\boldsymbol{A}'}\boldsymbol{B}$, across various homophily degrees. $\boldsymbol{B}$ is set to node features $\boldsymbol{X}$ in (a) and aggregated features $\hat{\boldsymbol{A}}\boldsymbol{X}$ in (b). Note that in (a), $\hat{\boldsymbol{A}'}$ only depends on $\boldsymbol{B}$ and does not change with homophily.

**Settings**   Based on CSBM-H [33] (see details in Appendix B), we generate synthetic graphs with 10 random seeds for each homophily degree $h \in \{0, 0.1, \ldots, 1.0\}$ to mitigate randomness effects. Each graph $\mathcal{G}$ contains 1000 nodes, with each node characterized by 10 features, 5 balanced classes,

and a degree sampled from the range $[2, 10]$. Then, we apply k-Nearest-Neighbors (kNN) on GSL bases $\boldsymbol{B}$ with $k = 5$ to generate new graphs, *i.e.,* $\mathcal{G}' = \text{kNN}(\boldsymbol{B})$.

The experiments are designed to answer two questions: **Q1**: Is the reconstructed graph necessary for GNN? **Q2**: How much does the reconstructed graph enhance GNN performance compared to the original graph structure?

Let us denote the original node representations (*i.e.,* original GSL bases) as $\mathbf{B}$, the original graph convoluted representations as $\mathbf{H}$, and the reconstructed graph convoluted embeddings as $\mathbf{H}'$. These bases will be separately fed into MLP, GCN, and GCN+GSL to compare model performance (prediction accuracy). We measure the quality of the bases using both the non-parametric metric mutual information $\text{I}(\cdot)$ and the parametric metric $\text{Acc}(\cdot)$. We test two different settings of GSL bases (1) graph-unaware GSL bases $\mathbf{B} = \mathbf{X}$ in Figure 3a, where $\mathbf{H}'$ does not rely on graph homophily; (2) and graph-aware GSL bases $\mathbf{B} = \hat{\mathbf{A}}\mathbf{X}$ in Figure 3b, where $\mathbf{H}'$ depends on graph homophily.

To answer **Q1**, we can compare $I(\boldsymbol{B}; \boldsymbol{Y})$ vs. $I(\boldsymbol{H}'; \boldsymbol{Y})$ and $\text{Acc}(\boldsymbol{B}; \boldsymbol{Y})$ vs. $\text{Acc}(\boldsymbol{H}'; \boldsymbol{Y})$. They compare the models which have and do not have graph reconstruction step, under the same GSL bases. To answer **Q2**, we can compare $I(\boldsymbol{H}; \boldsymbol{Y})$ vs. $I(\boldsymbol{H}'; \boldsymbol{Y})$ and $\text{Acc}(\boldsymbol{H}; \boldsymbol{Y})$ vs. $\text{Acc}(\boldsymbol{H}'; \boldsymbol{Y})$, which compare the performance of GCN and GSL-enhanced GCN. Through extensive experiments, we have the following observations.

**Observation 1. Mutual information is an effective non-parametric measure of model performance.** As shown in Figure 3a and 3b, the shape of mutual information $I(\cdot)$ curves (left) are highly similar to the curves for model accuracy $\text{ACC}(\cdot)$ (right). This shows that mutual information can effectively measure the quality of the embeddings. We will use it for theoretical analysis in the next section.

**Observation 2. Graph construction does not make significant difference.** In Figure 3, the mutual information $I(\boldsymbol{B}; \boldsymbol{Y})$ and classification accuracy $\text{ACC}(\boldsymbol{B}, \boldsymbol{Y})$ are close to $I(\boldsymbol{H}'; \boldsymbol{Y})$ and $\text{ACC}(\boldsymbol{H}', \boldsymbol{Y})$, respectively, across both graph-agnostic and graph-aware GSL bases. This suggests that the model performance does not improve significantly after applying graph convolution on the reconstructed graph $\mathcal{G}'$, which aligns with our analysis in the previous section.

**Observation 3. GSL-enhanced GCN only outperforms GCN in heterophilous graphs under graph-agnostic bases.** With graph-agnostic bases in Figure 3a, $I(\boldsymbol{H}; \boldsymbol{Y})$ and $\text{ACC}(\boldsymbol{H}, \boldsymbol{Y})$ increase as homophily increases, while $I(\boldsymbol{H}'; \boldsymbol{Y})$ and $\text{ACC}(\boldsymbol{H}', \boldsymbol{Y})$ remain constants across homophily degrees. As the reconstructed graph does not depend on homophily, the harmful connections in graphs with low homophily will not cause negative impact on $\boldsymbol{H}'$. Thus, GSL-enhanced GCN can outperform GCN. However, this effect is observed only when $\boldsymbol{B} = \boldsymbol{X}$.

Note that when $\boldsymbol{B} = \hat{\boldsymbol{A}}\boldsymbol{X}$ in Figure 3b, even when GCN+GSL outperforms GCN, its performance still remains close to MLP under the same GSL bases. This means that GSL-enhanced GNN cannot outperform the simple baselines significantly. Recent studies [36, 39] also indicate that under consistent hyperparameter tuning, GSL does not always consistently outperform classic GNN baselines. This leads us to reconsider the necessity of GSL. In addition to the above empirical observations, we proceed with a theoretical analysis on the effectiveness of GSL in the following section.

### 4.3 Theoretical Analysis

To explain the above empirical observations, in this section, we first prove that the mutual information $I(Y; H)$ between label $Y$ and aggregated features $H$ can serve as a non-parametric measurement of the effect of graph convolution. Following this, we compare the mutual information between the node labels $Y$ and either the original GSL bases $\boldsymbol{B}$ or the aggregated GSL bases $\boldsymbol{H}'$ (on $\mathcal{G}'$), to reveal the impact of GSL on model performance.

**Theorem 4.1.** *Given a graph $\mathcal{G} = \{\mathcal{V}, \mathcal{E}\}$ with node labels $\boldsymbol{Y}$ and node features $\boldsymbol{X}$, the accuracy of graph convolution on node classification $P_A$ is upper bounded by the mutual information of node label $Y$ and aggregated node features $H = AX$:*

$$P_A \leq \frac{I(Y; H) + \log 2}{\log(C)} \tag{4}$$

**Proposition 4.2.** *Consider a graph $\mathcal{G} = \{\mathcal{V}, \mathcal{E}\}$ characterized by node labels $Y$ and $n$-dimensional node bases $\mathbf{B} = (B_1, B_2, \ldots, B_n)$ with $C$ classes. Each base $B_i$ is independent and follows a class-dependent Gaussian distribution, i.e., $B_i \sim \mathcal{N}(\mu_Y, \sigma_Y)$. A new graph $\mathcal{G}' = \{\mathcal{V}, \mathcal{E}'\}$ is generated*

238  *using a non-parametric method based on the bases* $\mathbf{B}$. *For the aggregated bases* $\mathbf{B'}$ *on* $\mathcal{G'}$, *we have*
239  $\inf I(Y; \mathbf{B'}) \leq \inf I(Y; \mathbf{B})$.

240  where the proofs are shown in Appendix C.

241  Theorem 4.1 shows that the mutual information $I(Y; H)$ provides an upper bound on the accuracy
242  of graph convolution for node classification, which justifies why mutual information serves as an
243  effective measure of model performance, as demonstrated in Observation 1.

244  Based on the conclusion of mutual information in Theorem 4.1, we analyze the effectiveness of GSL.
245  Proposition 4.2 shows that the graph convolution on new graphs generated by GSL does not increase
246  the lower bound of mutual information. This explains why MLP performs similarly to, or slightly
247  better than, GCN+GSL in Observation 2 and the dilemma of GSL in Figure 2

248  To further explain Observation 3 in Section 4.2, we refer again to Proposition 4.2. In conjunction
249  with previous studies on graph homophily [38, 32, 56], we know that the performance of GCN could
250  be inferior to MLP on heterophilous graphs. Since GCN+GSL is upper bounded by the MLP on
251  the same GSL bases, when MLP outperforms GCN, GCN+GSL may also outperform GCN, as seen
252  in Figure 3a. However, even when GCN+GSL surpasses GCN in some cases, it still lags behind
253  MLP, a much simpler model, on the same GSL bases. Therefore, we hypothesize that previous GSL
254  improvements stem from the construction of the GSL bases or the introduction of additional model
255  parameters. A fair comparison of GSL with other GNNs or MLP baselines should be conducted using
256  the same GSL bases, as demonstrated in our experiments.

### 4.4  Complexity Analysis

258  After investigating the difference in the performance of GCN+GSL and GCN, we then analyze the
259  time complexity of some representative methods of GSL, such as IDGL [6], GRCN [53], GAug [55],
260  and HOG-GCN [46], as shown in Table 2. Assume the dimension of node representation is $F$ for all
261  the layers, the additional time complexity introduced by GSL generally includes: 1. Construction
262  of GSL bases: $O(|\mathcal{E}| F + |\mathcal{V}| F^2)$ for graph-aware bases or $O(|\mathcal{V}| F^2)$ for graph-agnostic bases, 2.
263  Graph construction: $O(|\mathcal{V}|^2 F)$, 3. Graph refinement: $O(|\mathcal{V}|^2)$, and 4: View Fusion $O(|\mathcal{V}|^2)$. Apart
264  from the complexity of the new graph construction in GSL, during the graph convolution, compared
265  with GNNs without using GSL, the additional complexity is further introduced by single view GSL
266  $O(|\mathcal{E'}| F)$ or multiple view GSL $O((N_{\mathcal{G}} - 1)(|\mathcal{E}| F + |\mathcal{V}| F^2))$, where $|\mathcal{E'}|$ is the additional edges
267  introduced in GSL and $N_{\mathcal{G}}$ is the number of views in GSL. Consider the fact that $|\mathcal{V}|^2 \gg |\mathcal{E}|$, we
268  have the total additional complexity of GSL by summing up all these terms: $O(|\mathcal{V}|^2 F + |\mathcal{V}| F^2)$.
269  Compared with the complexity in normal GCN $O(|\mathcal{E}| F + |\mathcal{V}| F^2)$ [4], this additional complexity
270  $O((|\mathcal{V}|^2 - |\mathcal{E}|)F)$ adds tremendous training time and grows exponentially with the number of nodes
271  in graphs, which is shown in our experiments.

## 5  Experiments

273  In this section, we examine the effectiveness of Graph Structure Learning (GSL) through extensive
274  experiments. To explore GSL's impact on Graph Neural Networks (GNNs), we compare the perfor-
275  mance of 450 GSL variants integrated with various GNN backbones in Section 5.1. Furthermore,
276  we analyze the influence of different components on GSL through an ablation study of each GSL
277  component in Section 5.2.

278  **Settings.** Our experiments include six popular GNNs as backbones: GCN [17], SGC [47], GraphSage
279  [10], and GAT [44], Mixhop [2], and ACMGNN [32]. The datasets used in our experiments
280  include heterophilous graphs: *Squirrel, Chameleon, Actor, Texas, Cornell, Wisconsin, Roman-empire,*
281  *and Amazon-ratings* [38, 42, 39], and homophilous graphs: *Cora, PubMed, and Citeseer* [52],
282  *Minesweeper, Tolokers, and Questions* [39]. We show more dataset details in Appendix D. The model
283  performance is measured by accuracy for multi-class datasets or AUC-ROC for binary-class datasets
284  on node classification tasks. We use $50\%/25\%/25\%$ random splits for training/validation/test sets.
285  For each experiment, we report the mean and standard deviation across 10 splits.

### 5.1  Performance Comparison

287  We investigate the impact of GSL on GNNs by the comparison of GNNs and the corresponding
288  GSL-enhanced GNNs (GNN+GSL). As GSL introduces significant variations in three key aspects,
289  we aim to comprehensively evaluate all possible GSL configurations through a combination of

Table 1: Performance Comparison of MLP, GNNs and the corresponding GSL-enhanced GNNs. For each GNN backbone, the best-performing method is highlighted in red, while the second-best method is highlighted in blue.

| Model | Construct | Fusion | Param Sharing | Mines. | Roman. | Amazon. | Tolokers | Questions | Squirrel | Chameleon | Actor | Texas | Cornell | Wisconsin | Cora | CiteSeer | PubMed | Rank |
|---|---|---|---|---|---|---|---|---|---|---|---|---|---|---|---|---|---|---|
| MLP | None | - | - | 79.55±1.23 | 65.45±0.99 | 46.65±0.83 | 75.94±1.38 | 74.92±1.39 | 39.29±2.22 | 43.57±4.18 | 35.40±1.38 | 80.46±6.44 | 73.78±7.34 | 85.88±7.78 | 87.97±1.80 | 76.68±2.10 | 87.39±2.18 | 3.93 |
| GCN | None | - | - | 90.07±5.79 | 81.46±1.25 | 50.89±1.16 | 84.61±0.99 | 77.68±1.10 | 41.26±2.47 | 43.24±3.86 | 34.34±1.17 | 73.08±8.68 | 67.03±10.54 | 78.24±8.32 | 87.97±1.51 | 76.75±2.30 | 89.47±0.64 | 1.36 |
| GCN | cos-graph | $\{\mathcal{G}'\}$ | - | 77.91±5.25 | 67.40±1.02 | 46.72±1.51 | 76.11±1.52 | 72.56±1.14 | 38.15±2.45 | 39.87±4.87 | 33.47±1.61 | 63.06±9.85 | 65.68±7.76 | 72.75±5.70 | 85.21±1.39 | 75.52±1.14 | 89.03±0.42 | 6.71 |
| GCN | cos-graph | $\{\mathcal{G},\mathcal{G}'\}$ | $\theta_1=\theta_2$ | 52.53±6.45 | 62.57±0.81 | 41.29±1.61 | 74.22±1.79 | 69.63±1.52 | 37.62±1.74 | 39.78±4.00 | 32.74±0.92 | 57.88±8.75 | 66.49±9.12 | 73.14±5.92 | 64.68±1.61 | 67.32±1.89 | 86.43±0.76 | 9.32 |
| GCN | cos-graph | $\{\mathcal{G},\mathcal{G}'\}$ | $\theta_1\neq\theta_2$ | 88.70±0.86 | 69.90±2.38 | 47.35±0.83 | 82.85±0.95 | 75.29±1.38 | 38.84±2.87 | 40.30±4.31 | 33.73±1.49 | 65.47±8.48 | 62.97±10.89 | 75.29±6.54 | 85.51±1.87 | 75.23±1.14 | 88.74±0.59 | 4.79 |
| GCN | cos-node | $\{\mathcal{G}'\}$ | - | 85.57±6.63 | 68.24±2.49 | 47.56±1.32 | 77.26±1.44 | 74.16±1.80 | 38.14±2.40 | 40.16±3.13 | 34.04±1.66 | 61.13±8.19 | 61.18±8.16 | 71.18±6.98 | 86.06±1.95 | 75.76±1.39 | 88.92±0.50 | 5.93 |
| GCN | cos-node | $\{\mathcal{G},\mathcal{G}'\}$ | $\theta_1=\theta_2$ | 52.53±6.45 | 62.57±0.81 | 41.29±1.61 | 74.22±1.79 | 69.63±1.52 | 37.62±1.74 | 39.78±4.00 | 32.74±0.92 | 57.88±8.75 | 66.49±9.12 | 73.14±5.92 | 64.68±1.61 | 67.32±1.89 | 86.43±0.76 | 9.36 |
| GCN | cos-node | $\{\mathcal{G},\mathcal{G}'\}$ | $\theta_1\neq\theta_2$ | 89.17±0.68 | 72.63±1.45 | 48.31±0.96 | 82.91±0.97 | 75.56±1.05 | 38.41±2.32 | 39.94±4.49 | 34.10±1.53 | 64.68±8.85 | 63.24±9.47 | 73.92±7.51 | 85.69±1.73 | 75.49±1.42 | 88.72±0.71 | 4.29 |
| GCN | kNN | $\{\mathcal{G}'\}$ | - | 82.89±6.66 | 68.44±1.83 | 47.13±1.00 | 78.92±1.79 | 73.90±1.73 | 38.15±2.02 | 40.22±3.82 | 33.94±1.24 | 63.03±8.53 | 61.35±9.28 | 72.16±7.41 | 86.08±1.62 | 75.56±1.42 | 88.59±0.58 | 5.93 |
| GCN | kNN | $\{\mathcal{G},\mathcal{G}'\}$ | $\theta_1=\theta_2$ | 52.53±6.45 | 62.57±0.81 | 41.29±1.61 | 74.22±1.79 | 69.63±1.52 | 37.62±1.74 | 39.78±4.00 | 32.74±0.92 | 57.88±8.75 | 66.49±9.12 | 73.14±5.92 | 64.68±1.61 | 67.32±1.89 | 86.43±0.76 | 9.39 |
| GCN | kNN | $\{\mathcal{G},\mathcal{G}'\}$ | $\theta_1\neq\theta_2$ | 88.96±0.73 | 72.44±1.61 | 47.06±0.83 | 83.10±0.80 | 75.61±1.19 | 37.63±1.93 | 40.18±4.76 | 33.84±1.94 | 63.87±9.68 | 62.16±9.77 | 75.49±7.29 | 85.82±1.55 | 75.50±1.30 | 88.54±0.55 | 5.00 |
| MLP | None | - | - | 79.55±1.23 | 65.45±0.99 | 46.65±0.83 | 75.94±1.38 | 74.92±1.39 | 39.29±2.22 | 43.57±4.18 | 35.40±1.38 | 80.46±6.44 | 73.78±7.34 | 85.88±7.78 | 87.97±1.80 | 76.68±2.10 | 87.39±2.18 | 3.71 |
| SGC | None | - | - | 83.45±4.47 | 78.04±0.69 | 51.38±0.68 | 84.88±1.13 | 77.39±1.23 | 41.18±2.73 | 42.35±4.10 | 34.05±1.41 | 73.63±6.94 | 70.27±9.91 | 80.59±5.13 | 88.10±1.89 | 77.52±2.20 | 89.39±0.62 | 1.57 |
| SGC | cos-graph | $\{\mathcal{G}'\}$ | - | 73.76±4.46 | 67.17±0.81 | 47.15±0.88 | 76.28±1.63 | 73.93±2.66 | 38.66±2.53 | 40.07±4.39 | 33.87±1.45 | 71.19±7.38 | 67.57±9.19 | 77.65±6.08 | 86.95±2.01 | 76.12±1.29 | 89.10±0.43 | 5.79 |
| SGC | cos-graph | $\{\mathcal{G},\mathcal{G}'\}$ | $\theta_1=\theta_2$ | 52.53±4.89 | 62.97±0.78 | 42.42±1.57 | 74.29±1.79 | 70.56±1.27 | 37.56±2.25 | 39.33±3.60 | 32.85±0.90 | 57.60±7.53 | 66.49±10.37 | 71.57±4.46 | 64.82±2.11 | 67.55±1.80 | 86.58±0.72 | 9.64 |
| SGC | cos-graph | $\{\mathcal{G},\mathcal{G}'\}$ | $\theta_1\neq\theta_2$ | 79.70±1.21 | 62.02±2.06 | 47.24±0.93 | 83.22±1.52 | 77.19±0.99 | 38.32±1.80 | 40.85±4.61 | 33.51±1.50 | 70.34±7.31 | 64.86±9.01 | 75.70±1.28 | 87.47±1.70 | 75.70±1.28 | 88.65±0.49 | 6.14 |
| SGC | cos-node | $\{\mathcal{G}'\}$ | - | 79.03±3.76 | 67.84±1.87 | 47.93±0.94 | 78.09±1.84 | 75.46±1.43 | 38.61±2.20 | 40.50±4.10 | 34.03±1.27 | 70.08±6.84 | 68.11±9.23 | 77.45±4.63 | 87.47±1.86 | 76.36±1.27 | 89.37±0.41 | 4.54 |
| SGC | cos-node | $\{\mathcal{G},\mathcal{G}'\}$ | $\theta_1=\theta_2$ | 52.53±4.89 | 62.97±0.78 | 42.42±1.57 | 74.29±1.79 | 70.56±1.27 | 37.56±2.25 | 39.33±3.60 | 32.85±0.90 | 57.60±7.53 | 66.49±10.37 | 71.57±4.46 | 64.82±2.11 | 67.55±1.80 | 86.58±0.72 | 9.57 |
| SGC | cos-node | $\{\mathcal{G},\mathcal{G}'\}$ | $\theta_1\neq\theta_2$ | 80.12±1.36 | 66.90±1.66 | 48.04±0.97 | 83.53±1.43 | 77.11±1.09 | 38.52±2.29 | 40.20±4.66 | 34.20±1.79 | 68.47±8.11 | 64.59±9.74 | 75.29±6.05 | 87.54±1.63 | 75.88±1.26 | 88.68±0.43 | 5.11 |
| SGC | kNN | $\{\mathcal{G}'\}$ | - | 75.53±4.98 | 67.94±0.70 | 47.68±0.84 | 79.45±2.06 | 74.22±2.47 | 37.32±2.10 | 39.92±3.91 | 34.05±1.55 | 72.81±6.15 | 70.00±7.98 | 77.84±6.02 | 87.82±1.77 | 76.54±1.44 | 89.19±0.42 | 4.64 |
| SGC | kNN | $\{\mathcal{G},\mathcal{G}'\}$ | $\theta_1=\theta_2$ | 52.53±4.89 | 62.97±0.78 | 42.42±1.57 | 74.29±1.79 | 70.56±1.27 | 37.56±2.25 | 39.33±3.60 | 32.85±0.90 | 57.60±7.53 | 66.49±10.37 | 71.57±4.46 | 64.82±2.11 | 67.55±1.80 | 86.58±0.72 | 9.50 |
| SGC | kNN | $\{\mathcal{G},\mathcal{G}'\}$ | $\theta_1\neq\theta_2$ | 80.78±1.08 | 64.59±1.93 | 47.48±0.99 | 83.17±1.43 | 76.80±1.09 | 36.53±2.06 | 40.17±4.24 | 34.23±1.72 | 69.26±6.77 | 65.95±8.87 | 76.08±5.92 | 87.38±1.49 | 76.02±1.22 | 88.77±0.45 | 5.79 |
| MLP | None | - | - | 79.55±1.23 | 65.45±0.99 | 46.65±0.83 | 75.94±1.38 | 74.92±1.39 | 39.29±2.22 | 43.57±4.18 | 35.40±1.38 | 80.46±6.44 | 73.78±7.34 | 85.88±7.78 | 87.97±1.80 | 76.65±2.10 | 87.39±2.18 | 4.14 |
| SAGE | None | - | - | 90.66±0.88 | 85.02±0.97 | 52.93±0.83 | 83.31±1.12 | 75.95±1.41 | 40.43±2.64 | 42.95±5.37 | 34.83±1.20 | 80.17±6.90 | 75.68±7.52 | 86.27±6.67 | 88.13±1.77 | 76.65±2.00 | 89.18±0.65 | 1.71 |
| SAGE | cos-graph | $\{\mathcal{G}'\}$ | - | 80.39±4.66 | 70.13±1.05 | 47.55±1.17 | 76.77±1.28 | 72.86±1.18 | 39.03±2.69 | 40.84±5.42 | 34.75±1.39 | 70.91±8.58 | 70.00±7.56 | 78.24±6.87 | 83.64±2.03 | 75.53±1.36 | 89.18±0.35 | 6.07 |
| SAGE | cos-graph | $\{\mathcal{G},\mathcal{G}'\}$ | $\theta_1=\theta_2$ | 53.02±6.49 | 59.98±1.73 | 39.99±2.29 | 71.57±2.28 | 66.01±3.58 | 35.05±2.41 | 38.49±3.68 | 31.32±1.04 | 60.30±7.05 | 67.57±4.59 | 76.47±5.92 | 64.58±1.74 | 67.77±1.31 | 85.53±0.51 | 9.93 |
| SAGE | cos-graph | $\{\mathcal{G},\mathcal{G}'\}$ | $\theta_1\neq\theta_2$ | 90.67±0.66 | 79.02±1.21 | 52.10±0.84 | 82.17±0.89 | 75.38±0.96 | 39.36±2.14 | 40.64±6.06 | 35.14±1.08 | 76.06±6.30 | 70.27±6.62 | 79.41±3.71 | 83.60±1.78 | 74.39±1.35 | 88.88±0.50 | 3.86 |
| SAGE | cos-node | $\{\mathcal{G}'\}$ | - | 85.26±4.64 | 71.25±1.76 | 48.96±0.87 | 78.39±1.75 | 73.01±1.11 | 38.68±2.75 | 40.81±4.51 | 35.10±1.26 | 71.47±9.47 | 68.11±7.87 | 75.49±6.32 | 84.88±1.90 | 75.58±1.04 | 89.17±0.35 | 5.64 |
| SAGE | cos-node | $\{\mathcal{G},\mathcal{G}'\}$ | $\theta_1=\theta_2$ | 53.02±6.49 | 59.98±1.73 | 39.99±2.29 | 71.59±2.28 | 66.01±3.58 | 35.05±2.41 | 38.49±3.68 | 31.32±1.04 | 60.30±7.05 | 67.57±4.59 | 76.47±5.92 | 64.58±1.74 | 67.77±1.31 | 85.53±0.51 | 9.79 |
| SAGE | cos-node | $\{\mathcal{G},\mathcal{G}'\}$ | $\theta_1\neq\theta_2$ | 90.64±0.65 | 78.60±0.98 | 52.08±0.90 | 82.02±0.88 | 75.31±1.12 | 39.18±2.54 | 40.86±6.17 | 35.18±1.24 | 74.71±5.65 | 69.73±7.43 | 80.00±5.68 | 83.96±1.65 | 74.63±1.26 | 88.93±0.64 | 3.93 |
| SAGE | kNN | $\{\mathcal{G}'\}$ | - | 82.86±3.14 | 70.74±0.80 | 48.40±1.01 | 78.12±2.17 | 72.70±1.15 | 38.93±2.84 | 39.68±5.40 | 35.09±1.14 | 70.91±9.05 | 68.92±6.88 | 75.69±6.73 | 84.40±1.75 | 75.68±1.43 | 88.86±0.44 | 6.50 |
| SAGE | kNN | $\{\mathcal{G},\mathcal{G}'\}$ | $\theta_1=\theta_2$ | 53.02±6.49 | 59.98±1.73 | 39.99±2.29 | 71.59±2.28 | 66.01±3.58 | 35.05±2.41 | 38.49±3.68 | 31.32±1.04 | 60.30±7.05 | 67.57±4.59 | 76.47±5.92 | 64.58±1.74 | 67.77±1.31 | 85.53±0.51 | 9.86 |
| SAGE | kNN | $\{\mathcal{G},\mathcal{G}'\}$ | $\theta_1\neq\theta_2$ | 90.61±0.63 | 79.16±1.15 | 51.56±1.07 | 81.66±0.87 | 75.22±0.97 | 39.20±2.39 | 40.44±5.82 | 35.13±1.38 | 74.17±6.31 | 70.54±7.32 | 79.61±6.61 | 84.05±1.63 | 74.59±1.25 | 88.67±0.55 | 4.57 |
| MLP | None | - | - | 79.55±1.23 | 65.45±0.99 | 46.65±0.83 | 75.94±1.38 | 74.92±1.39 | 39.29±2.22 | 43.57±4.18 | 35.40±1.38 | 80.46±6.44 | 73.78±7.34 | 85.88±7.78 | 87.97±1.80 | 76.68±2.10 | 87.39±2.18 | 3.86 |
| GAT | None | - | - | 90.41±1.34 | 84.51±0.84 | 52.00±2.84 | 84.37±0.96 | 77.78±1.27 | 41.67±2.51 | 43.83±3.66 | 33.73±1.77 | 75.28±8.12 | 65.41±12.14 | 77.84±7.41 | 88.02±1.92 | 76.77±2.02 | 89.21±0.67 | 2.04 |
| GAT | cos-graph | $\{\mathcal{G}'\}$ | - | 80.78±8.24 | 67.68±1.25 | 45.79±1.10 | 74.84±1.84 | 72.34±1.49 | 38.74±2.54 | 40.21±3.53 | 33.37±1.10 | 62.73±9.06 | 67.57±7.03 | 77.06±7.29 | 86.03±1.85 | 75.46±1.49 | 88.63±0.59 | 6.29 |
| GAT | cos-graph | $\{\mathcal{G},\mathcal{G}'\}$ | $\theta_1=\theta_2$ | 53.16±7.93 | 63.67±1.08 | 44.83±2.04 | 73.46±1.07 | 68.92±1.53 | 37.14±2.13 | 39.85±2.87 | 32.06±1.12 | 57.03±8.70 | 67.30±4.67 | 75.10±5.85 | 64.84±1.45 | 67.82±0.62 | 86.47±0.66 | 9.46 |
| GAT | cos-graph | $\{\mathcal{G},\mathcal{G}'\}$ | $\theta_1\neq\theta_2$ | 89.97±0.80 | 76.08±1.70 | 49.61±0.73 | 82.75±0.90 | 77.13±1.20 | 39.21±2.81 | 40.40±3.30 | 33.05±1.20 | 70.66±7.77 | 66.76±7.23 | 78.82±6.76 | 86.60±1.75 | 75.05±1.36 | 87.85±0.72 | 4.71 |
| GAT | cos-node | $\{\mathcal{G}'\}$ | - | 87.64±8.40 | 68.80±2.39 | 46.37±1.06 | 77.77±1.86 | 73.65±1.47 | 38.65±2.46 | 40.33±3.25 | 33.43±0.94 | 64.64±9.09 | 65.41±8.48 | 75.10±6.13 | 87.08±1.66 | 75.59±1.49 | 88.59±0.49 | 5.82 |
| GAT | cos-node | $\{\mathcal{G},\mathcal{G}'\}$ | $\theta_1=\theta_2$ | 53.16±7.93 | 63.67±1.08 | 44.83±2.04 | 73.46±1.07 | 68.92±1.53 | 37.14±2.13 | 39.85±2.87 | 32.06±1.12 | 57.03±8.70 | 67.30±4.67 | 75.10±5.85 | 64.84±1.45 | 67.82±0.62 | 86.47±0.66 | 9.46 |
| GAT | cos-node | $\{\mathcal{G},\mathcal{G}'\}$ | $\theta_1\neq\theta_2$ | 90.03±0.78 | 77.56±2.75 | 50.36±0.70 | 82.72±1.16 | 76.83±1.16 | 38.97±3.12 | 40.56±3.77 | 33.49±1.35 | 70.39±7.34 | 65.95±6.77 | 78.63±6.59 | 86.64±1.78 | 75.32±1.04 | 87.87±0.61 | 4.21 |
| GAT | kNN | $\{\mathcal{G}'\}$ | - | 84.27±5.25 | 68.73±1.47 | 46.05±0.90 | 77.57±1.75 | 71.58±1.62 | 38.82±2.33 | 40.12±3.69 | 33.84±1.07 | 61.68±8.71 | 62.97±7.43 | 74.90±5.86 | 86.77±1.90 | 75.64±1.45 | 88.29±0.48 | 6.50 |
| GAT | kNN | $\{\mathcal{G},\mathcal{G}'\}$ | $\theta_1=\theta_2$ | 53.16±7.93 | 63.67±1.08 | 44.83±2.04 | 73.46±1.07 | 68.92±1.53 | 37.14±2.13 | 39.85±2.87 | 32.06±1.12 | 57.03±8.70 | 67.30±4.67 | 75.10±5.85 | 64.84±1.45 | 67.82±0.62 | 86.47±0.66 | 9.46 |
| GAT | kNN | $\{\mathcal{G},\mathcal{G}'\}$ | $\theta_1\neq\theta_2$ | 89.96±0.79 | 77.23±1.63 | 49.79±0.72 | 82.78±0.95 | 76.67±1.13 | 39.65±2.76 | 41.11±3.92 | 33.54±1.36 | 70.38±7.22 | 65.95±6.52 | 77.84±7.23 | 86.97±1.75 | 75.20±1.55 | 87.97±0.51 | 4.18 |

various GSL components, which include (1) five GSL bases: original features $X$, aggregated features $\hat{A}X$, MLP-pretrained features $\text{MLP}(X)$, GCN-pretrained features $\text{GCN}(X,A)$, GCL (Graph Contrastive Learning)-pretrained features [62] $\text{GCL}(X,A)$; (2) three similarity-based graph construction methods: graphs are constructed via cosine similarity of GSL bases with threshold from the graph level (cos-graph) and node level (cos-node), and k-nearest neighbors (kNN); and (3) three view fusion methods: early fusion $\{\mathcal{G}'\}$, late fusion $\{\mathcal{G},\mathcal{G}'\}$ with parameter sharing $\theta_1=\theta_2$ or not $\theta_1\neq\theta_2$. To ensure a fair comparison of the performance between GNN+GSL, GNN, and MLP, we consider all five GSL bases as input choices and train all models on each GSL bases. The details of all these modules can be found in Appendix E.

Table 1 reports the performance of MLP, GNN baselines, and GNN+GSL across eight datasets under the best of five GSL bases. Notably, under fair comparison conditions, all six baseline GNNs outperform their GNN+GSL counterparts[4]. This suggests that **the incorporation of GSL does not consistently yield performance improvements of GNNs, and in some situations, it even lead to worse results.** Besides, under the same search space of GSL bases, MLP outperforms most GNN+GSL in average rank. This result verifies the dilemma in Section 4.1 that **high-quality GSL bases already provide informative node representations without newly constructed graphs.** Since GSL-based methods may require specific training procedures or more complex model designs, we further examine the performance of state-of-the-art (SOTA) GSL approaches to evaluate the potential of GSL in Appendix F.3, where the results also indicate GSL makes no significant improvement.

As previously mentioned, besides boosting model performance, GSL is often used to enhance the robustness of GNNs [15]. Therefore, under our proposed framework, we conduct fair experiments to study the robuseness of GSL-enhanced GNNs with the same GSL search space. Figure 4 demonstrates the performance of GNNs alongside their GSL-enhanced counterparts on perturbed graphs, incorporating feature noise, edge addition, and edge removal, as suggested by **(author?)** [26]. The curves for GSL-enhanced GNNs (dotted lines) is close to those of the original GNNs (solid lines) across three types of perturbations and four GNN backbones, indicating that the baseline GNNs perform comparably to their GSL-enhanced versions. Therefore, **the similarity-based graph construction may not be indispensable for enhancing model robustness.** See Appendix F.8 for more details.

## 5.2 Ablation Study on Each GSL Component

Since the performance of GNN and GNN+GSL models is comparable under the same bases, we further investigate how different components of GSL influence GNNs in Figure 5, where each result is the averaged performance of four GNN backbones, including GCN, GAT, SGC, and GraphSAGE.

---

[4]Due to page limitation, results of the other two heterophily-oriented GNNs are shown in Appendix F.5, where we can derive the same conclusion as in Table 1.

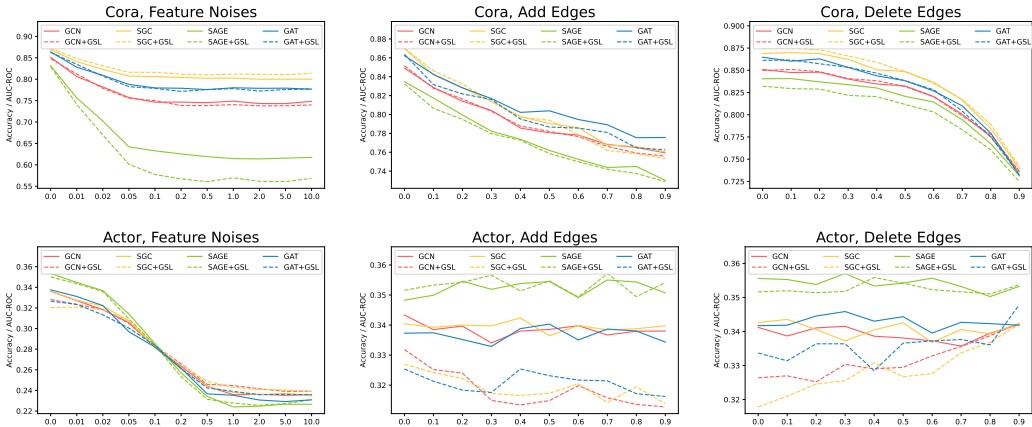

Figure 4: Response to feature noise, edge additions, and edge removals in GNN baselines and their GSL-enhanced counterparts.

The results indicate that: (1) Pretrained node representations, such as $\mathrm{MLP}(\mathbf{X})$ and $\mathrm{GCN}(\mathbf{X}, \mathbf{A})$, significantly enhance GNN performance [5], (2) GSL graph generation has minimal impact on model performance, (3) two view fusion with parameter separation improves GNN performance, and (4) early fusion generally outperforms late fusion. Especially, GSL bases influence model performance most among all the GSL components, verifying our analysis in Section 4 that the quality of GSL bases greatly influences GNN performance, while graph construction has little impact.

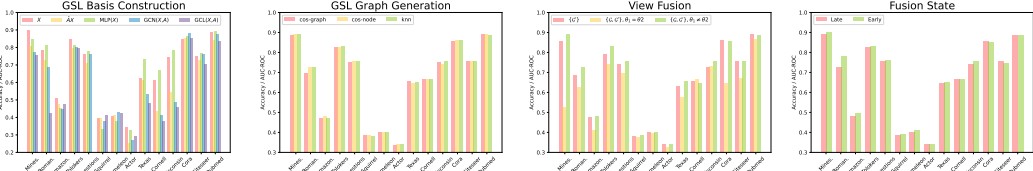

Figure 5: The influences of different GSL components on GNN+GSL.

## 6 Conclusion

In this paper, we disentangle the impact of GSL in GNN performance through our proposed GSL framework. Motivated by the dilemma associated with GSL, we show that it is the pretrained node features that really improve GNN performance instead of the similarity-based graph construction methods. Our research contributes to a deeper understanding of GSL and provides insights for re-evaluating essential components in future GNN designs. Although this paper primarily focuses on the impact of GSL on model performance in node classification tasks, future research could expand this analysis to other graph-related tasks and different types of graphs, as well as theoretically examine the effects of GSL under broader assumptions.

---

[5]See more discussion of GSL bases in Appendix F.2.1.

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

# A  Taxonomy of Graph Structure Learning Methods

We present several representative GSL-based GNNs within our proposed GSL framework in Table 2. Below, we provide a detailed description of each method.

Table 2: Representative GSL methods under our proposed GSL framework

| Method | Bases | Construct | Refinement | View Fusion | Training Mode |
|---|---|---|---|---|---|
| LDS [8] | $X$ | $\{\mathcal{E}' = kNN(B)\}$ +Opt. | Bernoulli($\mathcal{E}'$) | Late Fusion, $\{\mathcal{G}'_1, \mathcal{G}'_2, \ldots, \mathcal{G}'_m\}, \theta_1 = \theta_2$ | 2-stage |
| Geom-GCN [38] | Isomap/Poincare/ Struc2vec($X$, $A$) | $\{\mathcal{E}'|e'_{ij} = |B_i - B_j|\}$ | threshold($\mathcal{E}'$) | Late Fusion, $\{\mathcal{G}, \mathcal{G}'\}, \theta_1 \neq \theta_2$ | Static |
| ProGNN [15] | $\epsilon$ | $\{\mathcal{E}' = Opt(\epsilon)\}$ | Low Rank+Sparsity +Original | No Fusion, $\{\mathcal{G}'\}$ | Joint |
| IDGL [6] | MLP($X$) | $\{\mathcal{E}'|e'_{ij} = \cos(B_i, B_j)\}$ | topk($\mathcal{E}'$) | Early Fusion, $\{\mathcal{G} + \mathcal{G}'\}$ | Joint |
| GRCN [53] | GCN($X$,$A$) | $\{\mathcal{E}'|e'_{ij} = \sigma(B_i B_j^T)\}$ | topk($\mathcal{E}'$), sym($\mathcal{E}'$) | Early Fusion, $\{\mathcal{G} + \mathcal{G}'\}$ | Joint |
| GAug-M [55] | GCN$^{(2)}$($X$,$A$) | $\{\mathcal{E}'|e'_{ij} = \sigma(B_i B_j^T)\}$ | $\mathcal{G}'_+ = $ topk($\mathcal{E}'$), $\mathcal{G}'_- = $ bottom($\mathcal{E}'$) | Early Fusion, $\{\mathcal{G} + \mathcal{G}'_+ - \mathcal{G}'_-\}$ | Joint |
| GAug-O [55] | $X$ | $\{\mathcal{E}'|e'_{ij} = p(e_{ij}|\text{GAE}(B, A))\}$ | Gumbel($\mathcal{E}'$) | Early Fusion, $\{\mathcal{G} + \mathcal{G}'\}$ | Joint |
| SLAPS [7] | MLP($X$) | $\{\mathcal{E}' = kNN(B)\}$ | norm($\mathcal{E}'$),sym($\mathcal{E}'$) | No Fusion, $\{\mathcal{G}'\}$ | Joint |
| CoGSL [27] | GCN($X$, {$A$, kNN($X$), PPR($X$), Subgraph($X$)}) | $\{\mathcal{E}'|e'_{ij} = p(e_{ij}|\text{MLP}(B, A))\}$ | - | Early Fusion, $\{\mathcal{G}^*| \min \mathcal{L}_{CL}(\mathcal{G}, \mathcal{G}')\}, \theta_1 \neq \theta_2$ | 2-stage |
| GEN [45] | GCN($X$, $A$) | $\{\mathcal{E}' = kNN(B)\}$ | - | Late Fusion, $\{\mathcal{G}'_1, \mathcal{G}'_2, \ldots, \mathcal{G}'_m\}, \theta_1 \neq \theta_2$ | 2-stage |
| STABLE [22] | GCL($X$, $A$) | $\{\mathcal{E}'|e'_{ij} = \cos(B_i, B_j)$ or $\cos(B_i, B_j)\}$ | $\mathcal{G}'_+ = $ topk($\mathcal{E}'$), $\mathcal{G}'_- = $ threshold($\mathcal{E}'$) | Early Fusion, $\{\mathcal{G} + \mathcal{G}'_+ - \mathcal{G}'_-\}$ | Joint |
| SEGSL [63] | $X$ | $\{\mathcal{E}'|\min \mathcal{H}_S, e'_{ij} \in \text{EncTree}(kNN(B))\}$ | - | No Fusion, $\{\mathcal{G}'\}$ | Joint |
| SUBLIME [28] | GCN($X$, $A$) | $\{\mathcal{E}' = Opt(\epsilon)\}$ or $\{\mathcal{E}'|e'_{ij} = \cos/\text{Minkowski}(B_i, B_j)\}$ | topk($\mathcal{E}'$),sym($\mathcal{E}'$),norm($\mathcal{E}'$) | Separation, $\{\mathcal{G}, \mathcal{G}'\}, \theta_1 = \theta_2$ | Joint |
| BM-GCN [11] | $\hat{Y} = $ MLP($X$), $\min \mathcal{L}_{CE}(\hat{Y}, Y)$ | $\{\mathcal{E}' = BQB^T\}$ | norm($\mathcal{E}'$) | Early Fusion, $\{\mathcal{G} \odot \mathcal{G}'\}$ | Joint |
| WSGNN [21] | MLP($X$) | $\{\mathcal{E}'|e'_{ij} = cos(B_i, B_j)\}$ | - | Early Fusion, $\{\mathcal{G} + \mathcal{G}'\}$ | Joint |
| GLCN [14] | $X$ | $\{\mathcal{E}'|e'_{ij} = \phi(|B_i - B_j|)\}$ | norm($\mathcal{E}'$), Original +Sparsity+Smoothness | No Fusion, $\{\mathcal{G}'\}$ | Joint |
| ASC [23] | SpectralCluster($X$) | $\{\mathcal{E}'|e'_{ij} = \|B_i - B_j\|\}$ | topk($\mathcal{E}'$) | No Fusion, $\{\mathcal{G}'\}$ | Static |
| WRGAT [43] | GCN($X$, $A$) | $\{\mathcal{E}'|e'_{ij} \cdot Opt(B)\}$ | Sparsity + MultiHop | Early Fusion $\{\mathcal{G} + \mathcal{G}'\}$ | Static |
| HOG-GCN [46] | GCN($X$, $A$) | $\{\mathcal{E}'|e'_{ij} = \sigma(B_i B_j^T)\}$ | Sparsity + Smoothness | No Fusion $\{\mathcal{G}'\}$ | Joint |
| GGCN [51] | MLP($X$) | $\{\mathcal{E}'|e'_{ij} = \cos(B_i, B_j)\}$ | Low Rank + Sparsity | Early Fusion, $\{\mathcal{G} + \mathcal{G}'\}$ | Joint |
| GloGNN [25] | MLP($X$) | $\{\mathcal{E}' = Opt(B)\}$ | Sparsity+MultiHop | No Fusion, $\{\mathcal{G}'\}$ | Joint |
| HiGNN [57] | $\hat{Y} = $ GCN($X$, $A$), $\min \mathcal{L}_{CE}(\hat{Y}, Y)$ | $\{\mathcal{E}' = e'_{ij} = \cos(B_i, B_j))\}$ | topk($\mathcal{E}'$), sym($\mathcal{E}'$) | Late Fusion, $\{\mathcal{G}, \mathcal{G}'\}, \theta_1 \neq \theta_2$ | Static |

**LDS** [8]. The GSL bases in LDS is constructed as node features $X$ and the GSL graph $\mathcal{G}'$ is initialized using a k-Nearest-Neighbors algorithm based on $B$. Then, $\mathcal{G}'$ is updated with a loss function of node classification. Then multiple graphs are sampled based on $\mathcal{G}'$ with a Bernoulli function and used to update the model parameters. The $\mathcal{G}'$ construction and model parameters are updated as a 2-stage mode.

**Geom-GCN** [38]. Geom-GCN constructs the GSL bases from several graph-aware node embedding strategies using both of the $X$ and $A$: Isomap [], Poincare [], and struc2vec []. Then, new graphs are constructed by filtering node pairs with a higher similarity measured by Euclidean distance $\{\mathcal{E}'|e'_{ij} = |B_i - B_j| < \delta\}$ where $\delta$ is a threshold. Finally, both of the aggregated message from $\mathcal{G}$ and $\mathcal{G}'$ are fused after applying graph convolution layers with no parameter sharing. The $\mathcal{G}'$ is not updated through the training process.

**ProGNN** [15]. The $\mathcal{G}'$ in ProGNN is purely learned by optimization without GSL bases. It optimizes the $\mathcal{G}'$ using low rank, sparsity, and similarity with the original graphs $\mathcal{G}$. It outputs a single graph $\mathcal{G}'$ without fusion and updates the $\mathcal{G}'$ together with model parameters.

**IDGL** [6]. The GSL bases in LDS is constructed by linear transformation of node features MLP($X$). Then, a GSL graph $\mathcal{G}'$ is constructed using cosine similarity with topk threshold refinement. The early fusion is applied by fusing GSL graph $\mathcal{G}'$ with original graph $\mathcal{G}$ before training. The GSL graph $\mathcal{G}'$ is trained with model parameters jointly.

**GRCN** [53]. GRCN constructs GSL bases by node embeddings of graph convolution GCN($X$, $A$). Then, the GSL graph $\mathcal{G}'$ is constructed by a kernel function with topk and symmetrization refinement $\{\mathcal{E}'|e'_{ij} = \sigma(B_i B_j) > \delta\}$. The final graph is obtained by early fusion and the GSL graph $\mathcal{G}'$ is updated together with model parameters.

**GAug-M** and **GAug-O** [55]. GAug-M constructs GSL bases using a 2-layer graph convolution GCN$^{(2)}$($X$, $A$). Then, the GSL graph $\mathcal{G}'$ is constructed by a kernel function. The final graph is obtained by adding some edges with highest probabilities and removing some edges with lowest probabilities on $\mathcal{G}$. GAug-O selects node features as GSL bases $X$, then trains a Graph Auto-Encoder to predict edges as $\mathcal{G}'$. Then, after gumbel sampling, the GSL graph $\mathcal{G}'$ is fused with original graph $\mathcal{G}$ before training. The $\mathcal{G}'$ in both of the GAug-M and GAug-O is updated together with model parameters.

**SLAPS** [7]. SLAPS constructs the GSL bases by applying MLP(X) followed by a k-nearest neighbors (kNN) algorithm based on node feature similarities. The GSL graph $\mathcal{G}'$ is then processed by an adjacency processor that symmetrizes and normalizes the adjacency matrix to ensure non-negativity and symmetry. The final graph is obtained of the generated graph $\mathcal{G}'$ with the node features without fusion. Additionally, a self-supervised denoising autoencoder $L_{DAE} = L(X_i, GNN_{DAE}(\hat{X}_i; \theta_{GNN_{DAE}}))$ is introduced to address the supervision starvation problem, updating $\mathcal{G}'$ together with the model parameters.

**CoGSL** [27]. CoGSL constructs GSL bases using two views, one of them is the Origin graph. Another is selected from the Adjacency matrix $A$, Diffusion matrix $PPR(X)$, the KNN graph $KNN(X)$ and the Subgraph of the Origin. GCNs are applied to these views to obtain node embeddings. The GSL graph is constructed by applying a linear transformation to the node embeddings of each node pair to estimate the connection probability between them. This connection probability is then added to the original view to finalize the graph. The refinement $\mathcal{E}'|e'_{ij} = p(e_{ij}|\text{MLP}(\mathbf{B}, \mathbf{A}))$ step involves maximizing the mutual information between the two selected views and the newly constructed graph. InfoNCE loss is used to optimize the connection probability, where the same node serves as a positive sample, and different nodes serve as negative samples. The final graph $\mathcal{G}'$ is obtained via early fusion of the selected views, and the GSL graph is updated with model parameters.

**GEN** [45]. GEN constructs the GSL bases by generating kNN graphs though several GCN layer, utilizing node representations from different layers. These kNN graphs are then combined using a Stochastic Block Model (SBM) to create a new graph $\mathcal{G}'$. The GSL graph $\mathcal{G}'$ is refined iteratively through Bayesian inference to maximize posterior probabilities $P(G, \alpha, \beta|O, Z, Y_l) = \frac{P(O|G,\alpha,\beta)P(G,\alpha,\beta)P(O,Z,Y_l)}{P(O,Z,Y_l)}$, considering both the original graph and node embeddings. The final graph is obtained by feeding the graph $Q$ back into the GCN for further optimization. The iterative process updates both the GSL graph and GCN parameters as a 2-stage mode, providing mutual reinforcement between the graph estimation and model learning.

**STABLE** [22]. STABLE constructs the GSL bases by generating augmentations based on node similarity through kNN graph and perturbing edges to simulate adversarial attacks. The GSL graph $\mathcal{G}'$ is constructed by refining the structure using contrastive learning between positive samples (slightly perturbed graphs) and negative samples (undesirable views generated by feature shuffling). The refinement step applies a top-k filtering strategy on the node similarity matrix to retain helpful edges while removing adversarial ones. The final graph is obtained through early fusion, and the GSL graph $\mathcal{G}'$ is updated together with model parameters during joint training

**SE-GSL** [63]. SE-GSL constructs the GSL bases using a kNN graph fused with the original graph. The GSL graph $\mathcal{G}'$ is constructed through a structural entropy minimization process that extracts hierarchical community structures in the form of an encoding tree. The final graph is optimized by sampling node pairs from the encoding tree and generating new edges based on the minimized entropy structure. The refined graph is then used for downstream tasks, and the GSL graph $\mathcal{G}'$ is updated jointly with model parameters during training.

**SUBLIME** [28]. SUBLIME constructs the GSL bases using both an anchor view (original graph) and a learner view (new graph). The new graph is initialized through kNN and further optimized either by parameter-based methods (using models like MLP, GCN, or GAT) or by non-parameter-based approaches (using cosine similarity or Minkowski distance). After obtaining the new graph, post-processing operations such as top-k filtering, symmetrization, and degree-based regularization are applied to ensure the graph's sparsity and structure. The GSL graph $\mathcal{G}'$ is refined by applying contrastive learning between the anchor and learner views, incorporating edge drop and feature masking to generate node embeddings. The final graph is used in downstream tasks, and both views are updated together with model parameters in a joint training process.

**BM-GCN** [11]. BM-GCN constructs the GSL bases by introducing soft labels for nodes enbedding $\mathbf{B} = softmax(\sigma(MLP(X)))$ via a multilayer perceptron $\mathcal{L}_{MLP} = \sum_{v_i \in \mathcal{V}} f(B_i, Y_i)$. These soft labels are then used to compute a block matrix (H) , which models the connection probabilities between different node classes. The GSL graph $\mathcal{G}'$ is constructed by creating a block similarity matrix $Q = HH^T$ from the block matrix $Y_s = Y_i, B_i|\forall v_i \in \mathcal{T}_y, \forall v_j \notin \mathcal{T}_y, H = (Y_s^T A Y_s) \circ (Y_s^T A E)$, reflecting similarities between classes. The new graph is optimized using $BQB^T$ and further fused with the original graph $A + \beta I$ for downstream tasks. The final graph is obtained by optimizing $\mathcal{G}'$

through degree-based regularization and top-k filtering. The GSL graph $\mathcal{G}'$ is updated together with model parameters during joint training.

**WSGNN** [21]. WSGNN introduces a two-branch graph structure learning method, where each branch operates on different aspects of the graph: Branch AZ learns node labels from the new graph structure, while Branch ZA learns the new graph structure from the labels. The GSL bases is constructed using the observed graph $A_{obs}$ and node features $X$. The new graph $A'$ is inferred via cosine similarity between node embeddings. After constructing two separate views from each branch, the final graph is obtained by averaging the graphs from both branches. The refinement process ensures sparsity through cosine-based edge calculation $\mathcal{E}'|e'_{ij} = cos(\mathbf{B}_i, \mathbf{B}_j)$. Finally, both views undergo early fusion, with graph structure and node labels optimized jointly using a composite loss function that includes ELBO for structure prediction and cross-entropy loss for label prediction. The final GSL graph $\mathcal{G}'$ is updated during joint training.

**GLCN** [14]. GLCN constructs the GSL bases by computing pairwise distances between node features and passing them through an MLP to obtain a block similarity score. This score is then processed with a softmax function to generate an $n \times n$ probability matrix that serves as the learned graph structure. The graph is refined using regularization techniques to ensure sparsity and feature smoothness $L_{GL} = \sum_{i,j=1}^{n} ||x_i - x_j||_2^2 S_{ij} + \gamma||S||_F^2 + \beta||S - A||_F^2$. The learned graph is then used for downstream graph tasks, where the task loss and the graph regularization loss are jointly optimized during joint training

**ASC** [23]. ASC constructs the GSL bases is formed by using pseudo-eigenvectors from spectral clustering. They divide the Laplacian spectrum into slices, with each slice corresponding to an embedding matrix. The GSL graph $\mathcal{G}'$ is constructed by adaptive spectral clustering, where pseudo-eigenvectors are weighted based on alignment with node labels Where $f_i^{\mathcal{Z}}$. For refinement, they apply top-K edge selection by minimizing node embedding distance and maximizing homophily $\underset{\mathcal{Z}}{argmin} \sum_{i,j \in V_Y}(d(f_i^Z, f_j^Z), 1(y_i, y_j))$. This final restructed graph is training without fusion. Finally, the GSL graph is updated together with the model parameters.

**WRGAT** [43]. WRGAT constructs the GSL bases using the node features and a weighted relational GNN (WRGNN) framework that fuses structural and proximity information. A multi-relational graph is built by assigning different types of edges based on the structural equivalence of nodes at various neighborhood levels. This framework adapts to both assortative and disassortative mixing patterns, which helps improve node classification tasks. The GSL graph $\mathcal{G}'$ is refined through attention-based message passing across these relational edges, and early fusion of proximity and structural features is used. The GSL graph $\mathcal{G}'$ is trained jointly with the model parameters to optimize the node classification task.

**HOG-GCN** [46]. HOG-GCN constructs the GSL bases by incorporating both topological information and node attributes to estimate a homophily degree matrix $S = BB^T, B = softmax(Z_m), Z_m^{(l)} = \sigma(Z_m^{(l-1)W_m^{(l)}})$. The GSL graph $\mathcal{G}'$ is constructed using a homophily-guided propagation mechanism, which adapts the feature propagation weights between neighborhoods based on the homophily degree matrix $Z^{(l)} = \sigma(\mu Z^{(l-1)}W_e^{(l)} + \xi \hat{D}^{(-1)}A_k \odot HZ^{(l-1)}W_n^{(l)})$. For refinement, the graph incorporates both k-order structures and class-aware information to model the homophily and heterophily relationships between nodes. The final graph is obtained through joint fusion of topological and attribute-based homophily degrees, and both graph structure and model parameters are updated during joint training.

**GGCN** [51]. GGCN constructs the GSL bases using node features and structural properties such as node-level homophily $h_i$ and relative degree $\bar{r}_i$. It incorporates structure-based edge correction by learning new edge weights derived from structural properties like node degree, and feature-based edge correction by learning signed edge weights from node features, allowing for positive and negative influences between neighbors. The GSL graph $\mathcal{G}'$ is constructed by combining signed and unsigned edge information, aiming to capture both homophily and heterophily. The refinement process uses edge correction and decaying aggregation to mitigate oversmoothing and heterophily problems. The final graph is updated with early fusion, and the GSL graph $\mathcal{G}'$ is optimized during joint training

**GloGNN** [25]. GloGNN constructs its GSL bases using node embeddings derived from MLP, combining both low-pass and high-pass convolutional filters. A coefficient matrix $Z^{(l)}$ is used to characterize the relationship between nodes and is optimized to capture both feature and structural

similarities $H_X^{(0)} = (1 - \alpha)H_X^{(0)} + \alpha H_A^{(0)}$. Refinement is achieved via top-k selection based on the multi-hop adjacency matrix, and the matrix is symmetrized. The final graph is obtained through global aggregation of nodes, capturing both local and distant homophilous nodes. This graph is then used in downstream tasks, where the GSL graph $\mathcal{G}'$ is jointly optimized with the model parameters.

**HiGNN** [57]. HiGNN constructs its GSL bases by utilizing heterophilous information as node neighbor distributions, which represent the likelihood of neighboring nodes belonging to different classes $\mathcal{H}_u = [p_1, p_2, ..., p_c], where \ p_i = \frac{|v|v \in \mathcal{N}_u, y_v = i|}{|\mathcal{N}_u|}$. A new graph structure $\mathcal{G}'$ is constructed by linking nodes with similar heterophilous distributions using cosine similarity. The refinement involves selecting top-k edges based on the similarity score and applying symmetrization. The final graph is fused with the original adjacency matrix $A$ and the newly constructed adjacency matrix $A'$ via late fusion during message passing, where the node embeddings from both $A$ and $A'$ are combined with a balance parameter $\lambda$. The graph $\mathcal{G}'$ and node embeddings are updated during static training.

# B  Contextual Stochastic Block Models with Homophily

To study the behavior of GNNs, CSBM-H [33, 37] have been proposed to create synthetic graphs with a controlled homophily degree. Specifically, in CSBM-H, for a node $u$ with label $y$, its features $\boldsymbol{X_u} \in \mathbb{R}^M$ are sampled from a class-wised Gaussian distribution $\boldsymbol{X_u} \sim \boldsymbol{N}_{Y_u}(\boldsymbol{\mu}_{Y_u}, \boldsymbol{\Sigma}_{Y_u})$ with $\boldsymbol{\mu}_{Y_u} \in \mathbb{R}^F$ and $\boldsymbol{\Sigma}_{Y_u} \in \mathbb{R}^{F \times F}$, where each dimension of $\boldsymbol{X_u}$ is independent from each other, *i.e.,*$\boldsymbol{\Sigma}_{Y_u} = \text{diag}(\mathbb{R}_{\geq 0}^n)$. Then, to generate graph structure $\mathcal{G}$ with given homophily degree $h$ with the range of $[0, 1]$, the node $u$ has the probability $h$ to connect intra-class nodes and the probability $\frac{1-h}{C-1}$ to connect inter-class nodes. After applying neighbor sampling, both of the node homophily $h_{node}$ and edge homophily $h_{edge}$ in $\mathcal{G}$ are approximately equal to $h$.

# C  Proof of Theorem

**Theorem 4.1**  *Given a graph $\mathcal{G} = \{\mathcal{V}, \mathcal{E}\}$ with node labels $\mathbf{Y}$ and node features $\mathbf{X}$, the accuracy of graph convolution in node classification is upper bounded by the mutual information between the node label $Y$ and the aggregated node features $H$:*

$$P_A \leq \frac{I(Y; H) + \log 2}{\log(C)} \tag{5}$$

*Proof.* For an arbitrary node $u$, the aggregated node features can be derived as $H_u = \frac{1}{|\mathcal{N}_u|} \sum_{v \in \mathcal{N}_u} X_v$ following the graph convolution operation. For a classifier predicting labels based on $H_u$, we have $\hat{Y}_u = \text{cls}(H_u)$. Consequently, the Markov chain $Y \rightarrow H \rightarrow \hat{Y}$ holds. By applying Fano's inequality [9], we obtain

$$H(Y|H) \leq H_b(P_E) + P_E \log(C - 1) \tag{6}$$

where $P_E$ represents the error rate and $H_b(\cdot)$ is the binary entropy function. Rearranging this inequality gives us a lower bound on $P_E$:

$$P_E \geq \frac{H(Y|H) - H_b(P_E)}{\log(C - 1)} \tag{7}$$

Since $H(Y|H) = H(Y) - I(Y; H) = \log(C) - I(Y; H)$ and $H_b(P_E) \leq \log 2$, we can substitute these terms into the equation:

$$P_E \geq 1 - \frac{I(Y; H) + \log 2}{\log(C)} \tag{8}$$

Finally, by expressing the accuracy rate $P_A$, we find:

$$P_A = 1 - P_E \leq \frac{I(Y; H) + \log 2}{\log(C)} \tag{9}$$

This concludes the proof of Theorem 4.1.

**Proposition 4.2** *Consider a graph $\mathcal{G} = \{\mathcal{V}, \mathcal{E}\}$ characterized by node labels $Y$ and $n$-dimensional node bases $\mathbf{B} = \{B_1, B_2, \ldots, B_n\}$ with $C$ classes. Each base $B_i$ is independent and follows a class-dependent Gaussian distribution, i.e., $B_i \sim \mathcal{N}(\mu_Y, \sigma_Y)$. A new graph $\mathcal{G}' = \{\mathcal{V}, \mathcal{E}'\}$ is generated using a non-parametric method based on the bases $\mathbf{B}$. For the aggregated bases $\boldsymbol{B}'$ on $\mathcal{G}'$, we have $\inf I(Y; \boldsymbol{B}') \leq \inf I(Y; \boldsymbol{B})$.*

*Proof.* Let's first consider the mutual information for $i$-th node base $B_i$. For a non-parametric GSL method, we have the probability that class $k$ connects with class $j$ as:

$$p_{k,j} = \frac{g(B_i^k, B_i^j)}{\sum_{q=1}^{C} g(B_i^k, B_i^q)} \tag{10}$$

where $g(\cdot)$ is a non-parametric measurement of the probability of new connections, such as cosine similarity or Minkowski Distance. Then, we can get aggregated bases from the new graph by the operation of graph convolution [37, 33]:

$$B_i'^k = \sum_{q=1}^{C} p_{k,q} B_i^q \tag{11}$$

Therefore, the Markow chain $Y \to B_i \to B_i'$ holds. From data processing inequality [3], we have

$$I(Y; B_i') \leq I(Y, B_i) \tag{12}$$

To extend this conclusion to multi-dimensional variables, we apply the chain rule of mutual information

$$I(Y; \mathbf{B}) = I(Y; \{B_1, \ldots, B_n\}) = \sum_{i=1}^{n} I(Y; B_i \mid \{B_1, \ldots, B_{i-1}\})$$
$$I(Y; \mathbf{B}') = I(Y; \{B_1', \ldots, B_n'\}) = \sum_{i=1}^{n} I(Y; B_i' \mid \{B_1', \ldots, B_{i-1}'\}) \tag{13}$$

Due to the property that conditioning reduces entropy, we have

$$I(Y; B_i \mid \{B_1, \ldots, B_{i-1}\}) \geq I(Y; B_i)$$
$$I(Y; B_i' \mid \{B_1', \ldots, B_{i-1}'\}) \geq I(Y; B_i') \tag{14}$$

Thus, we have

$$\inf I(Y; \mathbf{B}) = \sum_{i=1}^{n} I(Y; B_i) \text{ and } \inf I(Y; \mathbf{B}') = \sum_{i=1}^{n} I(Y; B_i') \tag{15}$$

where $\inf$ represents infimum. Since $I(Y; B_i') \leq I(Y, B_i)$ holds for each $i$, we have

$$\inf I(Y; \boldsymbol{B}') \leq \inf I(Y; \boldsymbol{B}) \tag{16}$$

This concludes the proof of Proposition 4.2.

# D  Dataset Details

The datasets used in our experiments include heterophilous graphs: Squirrel, Chameleon, Actor, Texas, Cornell, and Wisconsin [38, 42], homophilous graphs: Cora, PubMed, and Citeseer [52], and Minesweeper, Roman-empire, Amazon-ratings, Tolokers, and Questions [39]. The dataset statistics are shown in 3. The descriptions of all the datasets are given below:

**Cora**, **Citeseer**, and **Pubmed** datasets are widely used citation networks in graph structure learning research. In each dataset, nodes represent academic papers, while edges capture citation relationships

Table 3: Dataset Statistics

| Dataset | #Nodes | #Edges | #Classes | #Features | Edge Homophily |
|---|---|---|---|---|---|
| Cora | 2,708 | 5,278 | 7 | 1,433 | 0.81 |
| Pubmed | 19,717 | 44,324 | 3 | 500 | 0.80 |
| Citeseer | 3,327 | 4,552 | 6 | 3,703 | 0.74 |
| Roman-empire | 22,662 | 32,927 | 18 | 300 | 0.05 |
| Amazon-ratings | 24,492 | 93,050 | 5 | 300 | 0.38 |
| Minesweeper | 10,000 | 39,402 | 2 | 7 | 0.68 |
| Tolokers | 11,758 | 529,000 | 2 | 10 | 0.59 |
| Questions | 48,921 | 153,540 | 2 | 301 | 0.84 |
| Cornell | 183 | 295 | 5 | 1,703 | 0.30 |
| Chameleon | 2,277 | 36,101 | 5 | 2,325 | 0.23 |
| Wisconsin | 251 | 466 | 5 | 1,703 | 0.21 |
| Texas | 183 | 309 | 5 | 1,703 | 0.11 |
| Squirrel | 5,201 | 216,933 | 5 | 2,089 | 0.22 |
| Actor | 7,600 | 33,544 | 5 | 931 | 0.22 |

between them. The node features are bag-of-words vectors derived from the paper's content, and each node is assigned a label based on its research topic. These datasets offer a structured framework to evaluate GNN models on classification tasks within citation networks.

**Roman-Empire** is constructed from the Roman Empire Wikipedia article, with nodes representing words and edges formed by either word adjacency or dependency relations. It contains 22.7K nodes and 32.9K edges. The task is to classify words by their syntactic roles, and node features are fastText embeddings. The graph is chain-like, with an average degree of 2.9 and a large diameter of 6824. Adjusted homophily is low ($h_{adj}$ = -0.05), making it useful for GNN evaluation under low homophily and sparse connectivity.

**Amazon-Ratings** is based on Amazon's product co-purchasing network, this dataset includes nodes as products (books, CDs, DVDs, etc.) and edges linking frequently co-purchased items. It consists of the largest connected component of the graph's 5-core. The goal is to predict product ratings grouped into five classes.

**Minesweeper** is a synthetic dataset resembling the Minesweeper game, nodes in a 100x100 grid represent cells, with edges connecting adjacent cells. The task is to identify mines (20% of nodes). Node features indicate neighboring mine counts, with 50% of features missing. The average degree is 7.88, and the graph has near-zero homophily due to random mine placement.

**Tolokers** is derived from the Toloka crowdsourcing platform, where nodes represent workers connected by shared tasks. The graph has 11.8K nodes and an average degree of 88.28. The task is to predict which workers have been banned, using profile and task performance features. The graph is much denser than others in the benchmark.

**Questions** is based on user interactions from Yandex Q, this dataset focuses on users interested in medicine. Nodes are users, and edges represent questions answered between users. It contains 48.9K nodes with an average degree of 6.28. The task is to predict user activity at the end of a one-year period, with fastText embeddings from user descriptions as features. The graph is highly imbalanced (97% active users).

**Texas**, **Wisconsin**, **Cornell** are part of the WebKB project, representing web pages from university computer science departments. Nodes correspond to web pages, and edges represent hyperlinks between them. The node features are bag-of-words vectors from the web page content, and the labels classify each page into one of five categories: student, project, course, staff, and faculty.

**Chameleon**, **Squirrel** are page-page networks based on specific topics from Wikipedia. Nodes represent web pages, and edges correspond to mutual links between them. Node features are derived from the page content, and the classification task is based on average monthly traffic. These datasets are characterized by high heterophily, making them challenging for traditional GNN models.

**Actor** is an induced subgraph from a film-director-actor-writer network. Nodes represent actors, and edges are created when two actors co-occur on the same Wikipedia page. The task is to classify actors into five categories based on the keywords associated with their Wikipedia pages.

## E Implementation Details

We implement GSL on 6 baseline GNNs with a variety of GSL approaches from the perspective of GSL bases, GSL graph construction, and view fusion. The baseline GNNs include:

- **GCN** [17] performs layer-wise propagation of node features and aggregates information from neighboring nodes to capture local graph structures. Each layer applies a convolution operation to update node embeddings, combining the node's features with its neighbors.
- **GAT** [44] employs self-attention to learn dynamic attention coefficients between nodes and their neighbors. These coefficients are normalized using softmax, and the final node representation is computed as a weighted sum of the neighbor features. Multi-head attention is used to enhance stability and expressiveness, with the number of attention heads set to 8 by default in our experiments.
- **SAGE** [10] uses an inductive framework to aggregate features from a node's local neighborhood, allowing it to generalize to unseen nodes. The aggregation function, set to mean in our experiments, efficiently combines neighbor information at each layer.
- **SGC** [47] simplifies the GCN model by removing non-linear activations and collapsing multiple layers into a single linear transformation. This reduction in complexity accelerates training. Node features are propagated using precomputed matrices, making the model faster and more efficient. In our experiments, the number of k-hops in SGC is set to 2 by default.
- **MixHop** [2] extends traditional GNNs by allowing nodes to aggregate information from neighbors at multiple distances within a single layer. Instead of only considering immediate neighbors, MixHop raises the adjacency matrix to different powers, capturing diverse topological signals. In our experiments, we follow the original paper's setup by using three propagation levels.
- **ACMGCN** [32] introduces an adaptive channel mixing mechanism to dynamically learn and combine information from different channels of node features. By leveraging attention-based feature transformation, ACMGCN enhances representation learning for graphs with diverse structural properties. In our experiments, we use the default channel mixing setup as described in the original paper.

The GSL bases $B$ includes the following options:

- $B = X$: The original node features are used as the GSL bases.
- $B = \hat{A}X$: Aggregated node features from 1-hop neighbors, normalized by node degree, are used as the GSL bases.
- $B = \text{MLP}(X)$: Pretrained MLP embeddings are used as the GSL bases. A 2-layer MLP is trained using node features and labels on the training set for 1000 epochs per run. The hidden layer size is set to 128, the learning rate to $1e^{-2}$, the dropout rate to 0.5, and the weight decay to $5e^{-4}$. All parameters are optimized with Adam. After training, node embeddings are extracted from the last hidden layer, with a dimension of 128, prior to classifier input.
- $B = \text{GCN}(X, A)$: Pretrained node embeddings are obtained from a 2-layer GCN model, following the same training procedure as for the MLP embeddings.
- $B = \text{GCL}(X, A)$: Pretrained node embeddings are derived from a Graph Contrastive Learning (GCL) model without supervision, following the same training process as the MLP embeddings. GRACE [62] is used as the GCL model, with 2 views and 2 layers. The edge and feature dropout rates in each view are set to 0.2.

The approaches for the construction of GSL graph $\mathcal{G}'$ includes:

- Cos-graph: $\mathcal{G}' = \{e_{ij} | \cos(B_i, B_j) > \delta, i \in \mathcal{V}, j \in \mathcal{V}\}$. This method calculates the cosine similarity between all node pairs in the original graph $\mathcal{G}$. Node pairs with a similarity higher than the threshold $\delta$ are selected as the edge set for the GSL graph $\mathcal{G}'$.

- Cos-node: $\mathcal{G}' = \bigcup_{i \in \mathcal{V}} \{\{e_i j\} | \cos(\boldsymbol{B_i}, \boldsymbol{B_j}) > \delta_i, j \in \mathcal{N}_i\}$. Unlike Cos-graph, which operates at the graph level, Cos-node constructs $\mathcal{G}'$ at the node level. To prevent nodes from being left without neighbors (which may occur in Cos-graph), Cos-node selects neighbors based on node-level cosine similarity, ensuring each node has sufficient connections.

- kNN: $\mathcal{G}' = \text{kNN}(\boldsymbol{B})$. This method constructs a kNN graph using the k-Nearest Neighbors algorithm based on the GSL bases $\boldsymbol{B}$.

The view fusion in GSL includes:

- $\{\mathcal{G}'\}$: This approach uses only the GSL graph $\mathcal{G}'$ for subsequent GNN training, completely ignoring the original graph $\mathcal{G}$.

- $\{\mathcal{G}, \mathcal{G}'\}, \theta_1 = \theta_2$. Both the GSL graph $\mathcal{G}'$ and the original graph $\mathcal{G}$ are used for GNN training, with parameter sharing across each layer of the GNN.

- $\{\mathcal{G}, \mathcal{G}'\}, \theta_1 \neq \theta_2$. Both the GSL graph $\mathcal{G}'$ and the original graph $\mathcal{G}$ are used for GNN training, but with separate model parameters for each graph.

Especially, for graphs with two views, the fusion stage in GSL includes:

- Early Fusion: $\mathcal{G} + \mathcal{G}'$.Combine the two graphs, $\mathcal{G}$ and $\mathcal{G}'$, into a single new graph prior to GNN training.

- Late Fusion: $\boldsymbol{H} + \boldsymbol{H}'$. After training the GNN on the original graph $\mathcal{G}$ and the GSL graph $\mathcal{G}'$, merge the node embeddings, $\mathbf{H}$ and $\mathbf{H}'$, before passing them to the classifiers.

In addition to the original models based on $4$ baseline GNNs, we implement GNN+GSL (GSL-augmented GNNs) by combining the aforementioned GSL modules, resulting in multiple variants for each type of GNN. For all models, we explore hyperparameters including hidden dimensions from the set $\{64, 128, 256\}$, learning rates from $\{$1e-2, 1e-3, 1e-4$\}$, weight decay values from $\{$0, 1e-5, 1e-3$\}$, the number of layers from $\{2, 3\}$, and dropout rates from $\{0.2, 0.4, 0.6, 0.8\}$. All the experiments are conducted on a Linux server(Operation system: Ubuntu 16.04.7 LTS) with one NVIDIA Tesla V100 card.

For GSL graph generation, we also search for additional hyperparameters to ensure the performance quality of the GSL-augmented GNN. Specifically, for Cos-graph and Cos-node, we control the parameter $\delta$ to vary the ratio of the number of edges in $\mathcal{G}'$ to the number of edges in $\mathcal{G}$ across the set $\{0.1, 0.5, 1, 5\}$. For kNN, we investigate the number of neighbors from the set $\{2, 3, 5, 10\}$.

## F    Additional Experiment Results

### F.1    Impact of GSL Bases on GNN baselines

In Figure 6, we illustrate the influence of 5 GSL bases on the performance of 4 GNNs across both homophilous and heterophilous graphs. The results indicate that MLP-pretrained features, denoted as $\text{MLP}(\boldsymbol{X})$, significantly enhance GNN performance compared to raw features $\boldsymbol{X}$ across 6 out of 9 datasets. These improvements stem from the self-training process applied to node inputs, suggesting that various self-training strategies could be employed with different graph datasets to further enhance GNN performance. Many GSL-enhanced GNNs leverage trained GSL bases to improve model performance, whereas GNN baselines utilize raw node features as GSL bases for comparison. This raises concerns about the fairness of previous comparisons between GNNs using original node features and those employing GNN+GSL, underscoring the importance of high-quality GSL bases. Additionally, we observe that GCN and GCL-pretrained features tend to degrade GNN performance on heterophilous datasets. This degradation is attributed to the increased noise within heterophilous datasets, leading to lower-quality GSL bases that can negatively impact GNN performance.

### F.2    Impact of each GSl component on GNN+GSL

#### F.2.1    GSL Bases

In addition to the analysis of the impact of GSL bases shown in Figure 5, Figure 7 presents further results on the performance of various GSL bases ($\mathbf{X}$, $\hat{\mathbf{A}}\mathbf{X}$, $MLP(\mathbf{X})$, $GCN(\mathbf{X}, \mathbf{A})$, $GCL(\mathbf{X}, \mathbf{A})$)

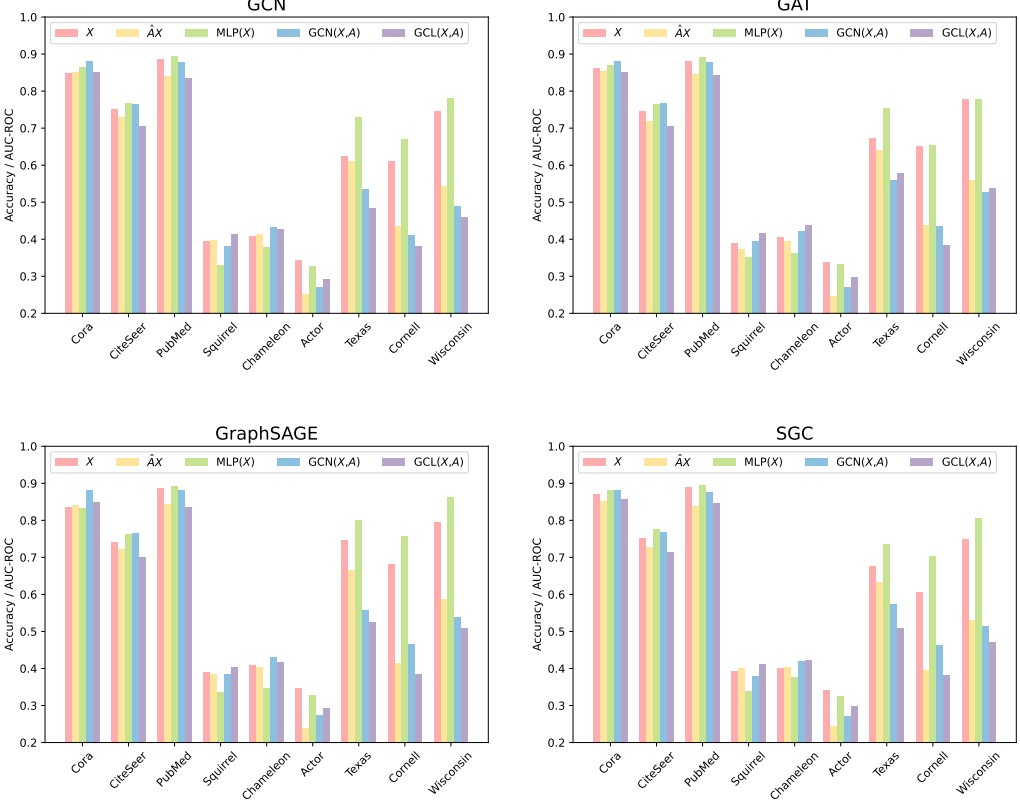

Figure 6: Influences of different GSL bases to GNNs.

across GAT, SGC, and GraphSAGE. The results are consistent with those observed in GCN and MLP, where the original node features do not always yield the best input. Some pretrained features, such as $MLP(\mathbf{X})$ on the Texas, Cornell, and Wisconsin datasets, demonstrate significant improvement compared to the original features $\mathbf{X}$, highlighting the necessity of self-training. Since many GSL methods [57, 43] utilize self-training during the training process, a fair comparison of these GSL methods and baseline GNNs should be conducted in the context of self-training, such as by using pretrained node features as input, as shown in Table 1.

### F.2.2 GSL Graph Generation

Figure 8 compares the Cos-graph, Cos-node, and kNN methods for GSL graph generation. Across most datasets, the performance differences among these methods are minimal. In certain datasets, such as Roman-empire and Pubmed, the models exhibit comparable performance regardless of the graph generation technique employed. This suggests that variations in graph generation have a limited effect on overall performance.

### F.2.3 View Fusion

Figure 9 illustrates the impact of different view fusion approaches, comparing the use of only the GSL graph $\mathcal{G}'$, the combination of the original graph $\mathcal{G}$ with $\mathcal{G}'$ using shared parameters $\theta_1 = \theta_2$, and the use of separate parameters $\theta_1 \neq \theta_2$. Notably, using only the GSL graph $\mathcal{G}'$ underperforms compared to employing both graph views with separate model parameters. This indicates that incorporating information from the original graph $\mathcal{G}$ is beneficial for maximizing GNN+GSL performance. Furthermore, for the two graph views, parameter sharing significantly underperforms parameter separation. We speculate that the messages aggregated under $\mathcal{G}$ and $\mathcal{G}'$ differ considerably, suggesting that different graphs should be treated with distinct model parameters.

### F.2.4 Fusion Stage

Figure 10 compares early fusion and late fusion for GNN+GSL with multiple graph views. The performance difference between the two fusion states is often minimal. While early fusion tends to perform slightly better on complex datasets like Actor and Pubmed, the overall impact of switching between early and late fusion is limited across most datasets. For simpler datasets like Minesweeper and Amazon, both fusion methods yield nearly identical performance, indicating that the choice of fusion state does not drastically alter the model's outcome in most cases.

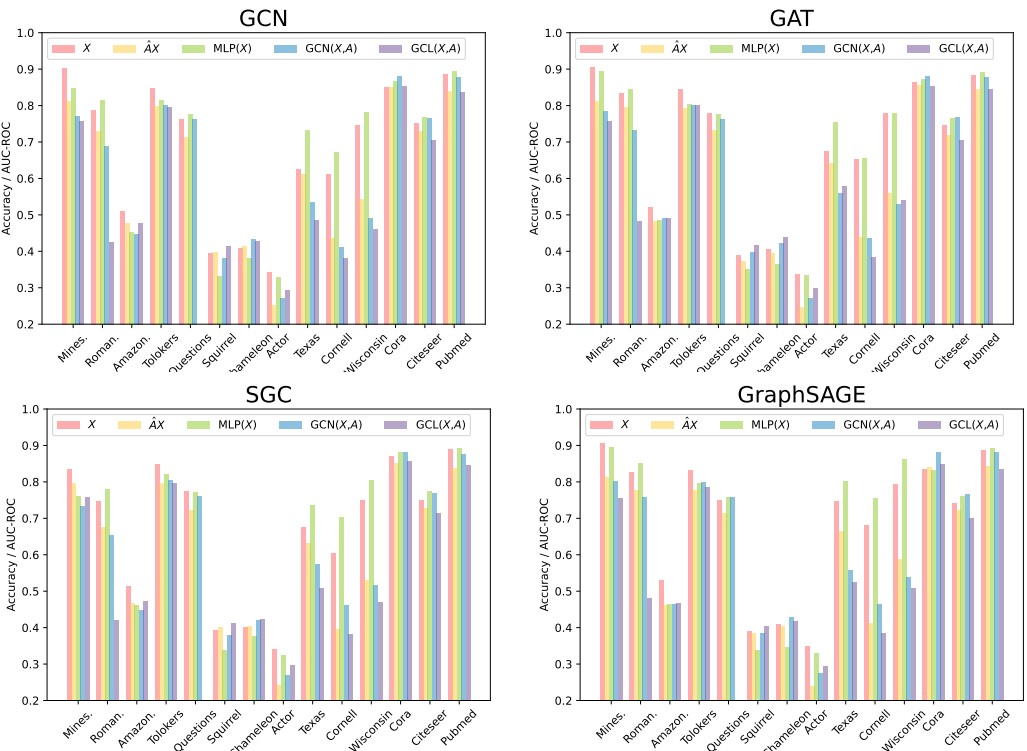

Figure 7: Influences of different GSL bases to more GNNs.

### F.3 Removing GSL in SOTA GNNs

**Settings**    To fairly reassess the impact of GSL in state-of-the-art (SOTA) methods, we compare the performance of SOTA models with their SOTA-GSL counterparts within the same hyperparameter search space. These GSL-based SOTA models include GAug [55], GEN [45], GRCN [53], IDGL [6], NodeFormer [48], GloGNN [25], WRGAT [43], and WRGCN [43]. Corresponding to the analysis of GCN and MLP in Section 4.2, the SOTA-GSL methods include two variants: (1) SOTA, $\mathcal{G}' = \mathcal{G}$, which replaces the GSL graph $\mathcal{G}'$ with the original graph $\mathcal{G}$; and (2) SOTA, $\mathcal{G}' = $ MLP, which substitutes the graph convolution layers of GSL $\mathcal{G}'$ with MLP layers. We train each model for 1000 epochs and search the hidden dimensions from the set {16, 32, 64, 128, 256, 512}, learning rate from {1e-1, 1e-2, 1e-3, 1e-4, 1e-5}, weight decay values from {5e-4, 5e-5, 5e-6, 5e-7, 0}, the number of layers from {1, 2, 3}, and dropout rates from {0.2, 0.4, 0.6, 0.8}. The hyperparameters of the above methods are shown in Table 5. The model-specific hyperparameters are shown as follows:

In **GRCN**, the hyperparameter K determines the number of nearest neighbors used to create a sparse graph from a dense similarity graph which helps balance efficiency and accuracy.We set the k as 5.

In **GAug**, the alpha is a hyperparameter that regulates the influence of the edge predictor on the original graph. We set the alpha as 0.1.

In **IDGL**, The parameter graph_learn_num_pers defines the number of perspectives for evaluating node similarities in the graph learning process. The parameter num_anchors specifies the number of

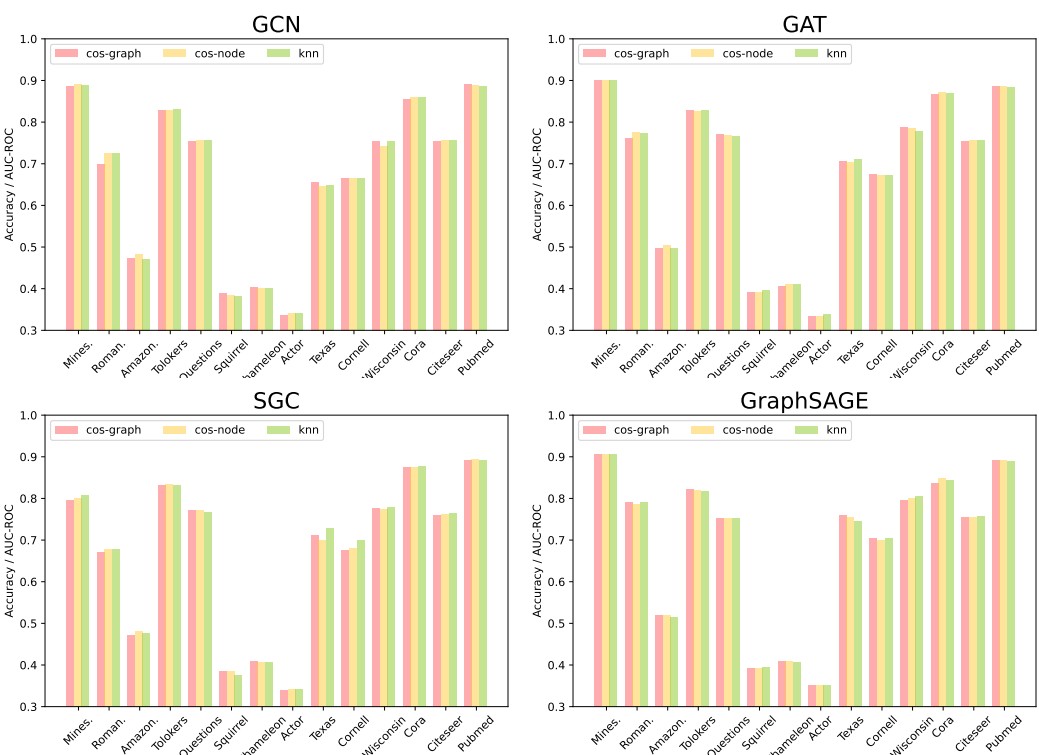

Figure 8: Influences of the approaches of GSL generation to GNN+GSL.

anchor points used to reduce computational complexity and improve scalability in graph structure learning. The graph_skip_conn parameter controls the proportion of skip connections, preserving information from the original graph during new graph structure learning. The update_adj_ratio parameter determines the proportion of the adjacency matrix updated at each iteration, influencing the dynamic adjustment of the graph structure. We set the graph_learn_num_pers as 6, num_anchors as 500, graph_skip_conn as 0.7, and update_adj_ratio as 0.3.

In **NodeFormer**, The parameter k determines the number of neighbors considered for each node in constructing the local graph structure, influencing the strength of node connections and the propagation of features. The parameter tolerance controls the degree of error tolerance during optimization. A larger tolerance allows more flexibility in the search space near local optima, while a smaller one results in stricter convergence. The number of attention heads in a graph attention network (GAT). Multi-head attention enables the model to focus on different subspace representations simultaneously, enhancing the diversity and stability of the representations. We set the k as 10, lambda as 0.01, and n_heads as 4.

In **GEN**, the parameter K in KNN refers to the number of nearest neighbors used to construct the graph structure, determining how many adjacent nodes are selected. The parameter tolerance defines the acceptable range of error during optimization, controlling the convergence criteria of the model. The parameter threshold determines the edge weight threshold in the graph, deciding which edges to retain in the graph structure. We set the k as 10, tolerance as 0.01, and threshold as 0.5.

In **GloGNN**, we set the Delta as 0.9, Gamma as 0.8, alpha as 0.5, beta as 2000, and orders as 5. Delta adjusts the balance between local and global node embeddings. Gamma controls the significance of global aggregation versus local information. Alpha balances the contributions of node features and graph structure. Beta regularizes the model, preventing overfitting. Order defines how many layers of neighbors are considered.

In **WRGAT**, we set the number of attention heads as 2 and the negative slope as 0.2. The number of attention heads determines how many attention mechanisms are used. The negative slope modifies the LeakyReLU activation.

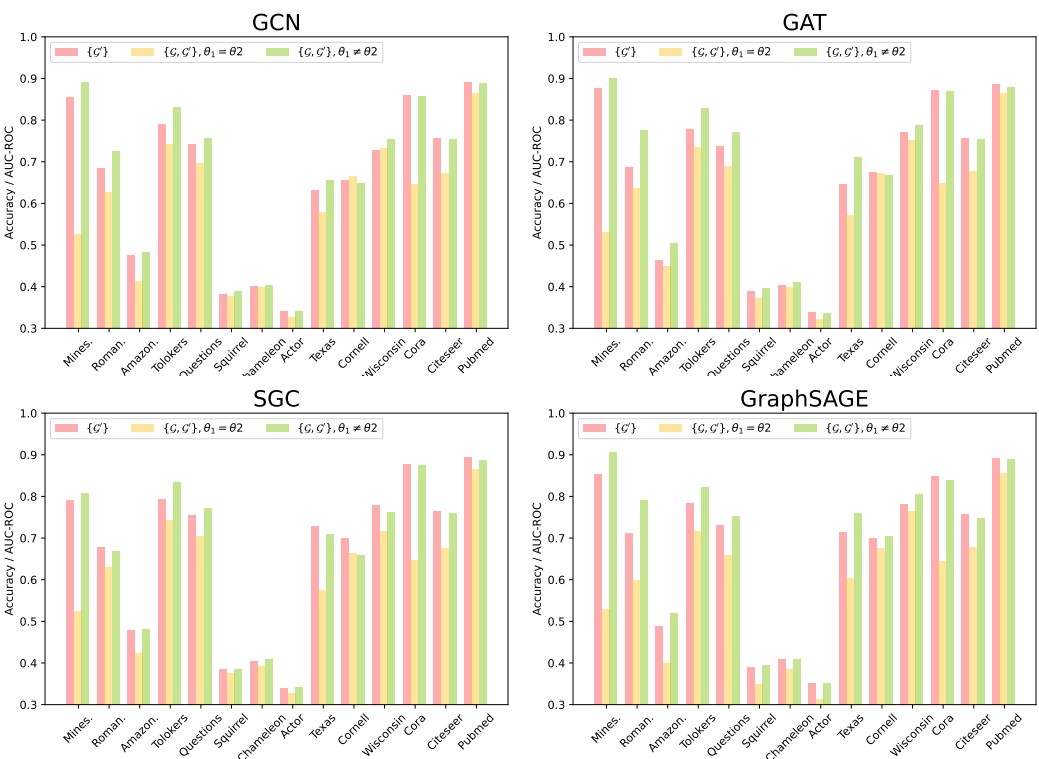

Figure 9: Influences of the approaches of view fusion in GSL to GNN+GSL.

**Results.** The results are presented in Table 4, where "OOM" denotes out-of-memory. It is evident that removing GSL does not diminish model performance; in fact, it is often comparable to or even exceeds the original results. Furthermore, GSL-based SOTA methods require significantly more GPU memory and longer running times compared to their non-GSL counterparts. Based on these findings, we conclude that GSL not only fails to enhance performance across most datasets but also increases model complexity. In conjunction with the results in Table 1, we assert that GSL may be unnecessary for effective GNN design in most cases.

Table 4: Model Performance and training time per epoch of SOTA methods and SOTA-GSL. The results for methods marked with "*" are reported in **(author?)** [58].

| Model | Questions AUC | Time | Minesweeper AUC | Time | Roman-empire Acc | Time | Amazon-ratings Acc | Time | Tolokers AUC | Time | Cora Acc | Time | Pubmed Acc | Time | Citeseer Acc | Time |
|---|---|---|---|---|---|---|---|---|---|---|---|---|---|---|---|---|
| GAug* | OOM | - | 77.93±0.64 | - | OOM | - | 48.42±0.39 | - | OOM | - | 82.48±0.66 | 7s | 78.73±0.77 | 20s | 72.79±0.86 | 10s |
| GAug, $\mathcal{G}'=\mathcal{G}$ | OOM | - | 80.56±0.36 | 11s | OOM | - | 48.45±0.37 | 12s | OOM | - | 81.73±0.38 | 1s | 79.38±0.46 | 6s | 72.34±0.18 | 2s |
| GAug, $\mathcal{G}'=$ MLP | OOM | - | 64.31±1.40 | 4.8s | OOM | - | 48.05±0.66 | 37s | OOM | - | 78.90±0.00 | 3.2s | 77.40±0.00 | 8.1s | 72.91±0.32 | 9s |
| GEN* | OOM | - | 79.56±1.09 | 260s | OOM | - | 49.17±0.68 | - | OOM | - | 81.66±0.91 | 214s | 78.49±3.98 | 1384s | 73.21±0.62 | 470s |
| GEN, $\mathcal{G}'=\mathcal{G}$ | OOM | - | 80.81±0.23 | 75s | OOM | - | 50.08±0.30 | 130s | OOM | - | 82.16±0.37 | 39s | 80.49±0.13 | 114s | 71.52±0.34 | 25s |
| GEN, $\mathcal{G}'=$ MLP | OOM | - | 71.81±0.98 | 12s | OOM | - | 49.29±0.65 | 49s | OOM | - | 80.20±0.00 | 140s | 66.80±0.00 | 1592s | 73.50±0.00 | 310s |
| GRCN* | 74.50±0.84 | - | 72.57±0.49 | 60s | 44.41±0.41 | 180s | 50.06±0.38 | 220s | 71.27±0.42 | 37s | 84.61±0.34 | 13s | 79.30±0.34 | 17s | 72.34±0.34 | 20s |
| GRCN, $\mathcal{G}'=\mathcal{G}$ | 75.69±0.52 | 8s | 71.15±0.05 | 10s | 45.84±0.52 | 8s | 46.07±1.02 | 10s | 71.73±0.42 | 10s | 81.66±1.10 | 2s | 79.35±0.26 | 3s | 69.55±1.28 | 2s |
| GRCN, $\mathcal{G}'=$ MLP | 63.59±2.35 | 3.9s | 72.18±1.09 | 2s | 45.89±0.83 | 7.5s | 48.77±0.60 | 8.1s | 70.45±1.39 | 8s | 79.40±0.00 | 1.3s | 78.10±0.00 | 5s | 71.40±0.00 | 4.2s |
| IDGL* | OOM | - | 50.00±0.00 | 157s | 47.10±0.65 | 186s | 45.87±0.58 | - | 50.00±0.00 | 279s | 84.19±0.61 | 123s | 82.78±0.44 | 146s | 73.26±0.53 | 332s |
| IDGL, $\mathcal{G}'=\mathcal{G}$ | OOM | - | 50.00±0.00 | 51s | 41.24±0.86 | 42s | OOM | - | 50.00±0.00 | 52s | 82.43±0.45 | 13s | 73.50±1.85 | 23s | 73.13±0.49 | 36s |
| IDGL, $\mathcal{G}'=$ MLP | OOM | - | 79.56±1.26 | 13.7s | 50.35±0.36 | 35s | 39.93±0.88 | 15s | 71.55±1.08 | 11s | 83.20±0.00 | 6.6s | 79.20±0.00 | 13s | 72.60±0.00 | 13.9s |
| NodeFormer* | OOM | - | 77.29±1.71 | - | 56.54±3.73 | - | 41.33±1.25 | - | OOM | - | 78.81±1.21 | 213s | 78.38±1.94 | - | 70.39±2.04 | 219s |
| NodeFormer, $\mathcal{G}'=\mathcal{G}$ | OOM | - | 80.66±0.82 | 215s | 68.37±1.95 | 236s | OOM | - | OOM | - | 77.01±1.99 | 152s | OOM | - | 70.82±0.13 | 139s |
| NodeFormer, $\mathcal{G}'=$ MLP | OOM | - | 80.04±1.42 | 21s | 53.08±2.37 | 7.2s | 71.55±1.08 | 26s | OOM | - | 78.82±0.00 | 8s | 76.30±0.00 | 127s | 72.80±0.00 | 15s |
| GloGNN | 68.67±1.07 | 66.6s | 52.45±0.30 | 13.0s | 66.21±0.17 | 26.1s | 50.72±0.88 | 31.1s | 79.81±0.20 | 47.4s | 78.07±1.66 | 6.6s | 87.88±0.26 | 18.2s | 71.95±1.90 | 21.8s |
| GloGNN, $\mathcal{G}'=\mathcal{G}$ | 68.32±1.23 | 49.4s | 52.30±0.21 | 3.6s | 66.03±0.14 | 15.3s | 50.23±0.83 | 21.7s | 80.02±0.16 | 25.1s | 73.49±2.01 | 5.1s | 87.62±0.20 | 14.4s | 72.27±2.08 | 21.2s |
| GloGNN, $\mathcal{G}'=$ MLP | 69.69±0.22 | 25.7s | 52.30±0.20 | 2.1s | 66.49±0.16 | 12.4s | 49.56±0.73 | 12.3s | 74.85±0.12 | 2.8s | 73.93±1.81 | 3.2s | 87.64±0.27 | 10.2s | 72.09±1.81 | 13.8s |
| WRGAT | OOM | - | 90.22±0.64 | 168.0s | OOM | - | OOM | - | 78.69±1.21 | 153.0s | 84.28±1.52 | 19.5s | 88.82±0.50 | 421.6s | 73.50±1.41 | 22.1s |
| WRGAT, $\mathcal{G}'=\mathcal{G}$ | 74.67±0.95 | 64.1s | 89.79±0.37 | 18.6s | OOM | - | 50.41±0.53 | 49.9s | 78.81±0.89 | 47.0s | 83.48±1.48 | 3.4s | 83.22±0.43 | 26.5s | 73.22±1.90 | 4.7s |
| WRGAT, $\mathcal{G}'=$ MLP | 68.07±2.62 | 75.8s | 87.08±2.11 | 16.2s | OOM | - | 41.38±1.46 | 24.4s | 76.41±1.25 | 37.7s | 76.99±1.10 | 2.9s | 80.27±6.23 | 23.9s | 65.28±2.11 | 4.5s |
| WRGCN | 74.70±1.71 | 358.3s | 90.63±0.64 | 40.9s | OOM | - | 52.76±0.95 | 508.4s | 82.68±0.82 | 52.3s | 88.30±1.46 | 23.7s | OOM | - | 73.74±1.60 | 54.2s |
| WRGCN, $\mathcal{G}'=\mathcal{G}$ | 75.91±1.30 | 43.3s | 90.65±0.49 | 5.5s | OOM | - | 52.54±0.56 | 50.1s | 82.65±0.86 | 15.6s | 88.32±0.79 | 3.9s | 89.26±0.45 | 19.4s | 74.45±1.51 | 10.5s |
| WRGCN, $\mathcal{G}'=$ MLP | 64.59±1.48 | 23.1s | 70.66±1.37 | 7.7s | OOM | - | 37.05±0.46 | 8.0s | 69.10±0.91 | 12.2s | 70.00±3.59 | 2.2s | 67.29±2.49 | 9.9s | 70.84±1.36 | 4.1s |

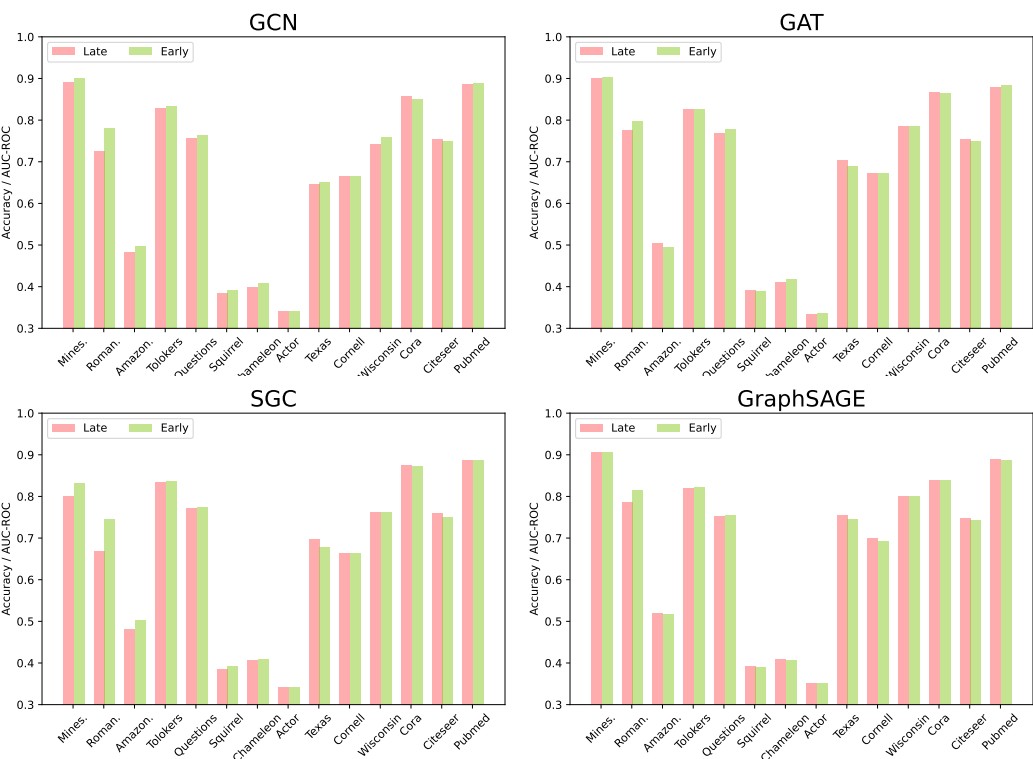

Figure 10: Influences of the states of view fusion in GSL to GNN+GSL.

### F.4 Quality of GSL Graphs

Previous studies [25, 57] suggest that GSL constructs graphs with properties that improve intra-class node connectivity, which can be measured by homophily. This improvement can be visualized by inspecting graph structures with nodes sorted by their class labels. A graph that appears closer to a block diagonal matrix indicates stronger intra-class connectivity. However, this enhancement may not always be essential and can be achieved through non-GSL methods as well. In Figure 11, we visualize the original and reconstructed structures of a heterophilous graph from the Wisconsin dataset. The GSL graphs are constructed using various bases: $\mathbf{X}$, $\hat{\mathbf{A}}\mathbf{X}$, $\mathrm{MLP}(\mathbf{X})$, $\mathrm{GCN}(\mathbf{X}, \mathbf{A})$, and $\mathrm{GCL}(\mathbf{X}, \mathbf{A})$. We also include reconstructed graphs using a simple method that samples edges between nodes of the same class based on label predictions, *i.e.,* $\hat{\mathbf{Y}} = \mathrm{GCN}(\mathbf{X}, \mathbf{A})$ or $\hat{\mathbf{Y}} = \mathrm{MLP}(\mathbf{X}, \mathbf{A})$. Figure 11 demonstrates that, although GSL improves intra-class connectivity, the improvement is not as substantial as that achieved by non-GSL methods, as seen in the last two subfigures. Thus, the improvement in homophily within GSL graphs is unnecessary, as it can be easily achieved through simple methods.

### F.5 Heterophily-oriented GNN with GSL

We also include heterophily-oriented GNNs, specifically ACMGNN [32] and MixHop [2], in our experiments that incorporate GSL into GNN baselines. These experiments follow the same setup as described in Table 1. The results, presented in Table 6, demonstrate that, under fair comparison conditions, both ACMGNN and MixHop outperform their GNN+GFS counterparts. This suggests that adding GSL to these heterophily-oriented GNNs may be unnecessary.

### F.6 Trainable GSL

In Table 7, we present the results of applying trainable GSL to baseline GNNs. Specifically, we select the best-performing GSL variants, as shown in Tables 1 and 6, for each backbone GNN. The best-performing method is highlighted in bold, while the runner-up is indicated with an underline.

Table 5: Hyperparameters for GSL-enhanced SOTA methods and their counterparts by replacing or removing new graphs.

| Dataset | Model | Learning Rate | Weight Decay | Dropout | Hidden Dim | Num of Layers |
|---|---|---|---|---|---|---|
| Cora | GAug | 1e-4 | 5e-7 | 0.8 | 512 | 2 |
| | GAug, $\mathcal{G}'=\mathcal{G}$ | 1e-4 | 5e-7 | 0.8 | 512 | 2 |
| | GAug, $\mathcal{G}'=$ MLP | 1e-4 | 5e-7 | 0.8 | 512 | 2 |
| | GEN | 1e-2 | 5e-4 | 0.5 | 16 | 2 |
| | GEN, $\mathcal{G}'=\mathcal{G}$ | 1e-2 | 5e-4 | 0.5 | 16 | 2 |
| | GEN, $\mathcal{G}'=$ MLP | 1e-2 | 5e-4 | 0.5 | 16 | 2 |
| | GRCN | 1e-3 | 5e-3 | 0.5 | 256 | 2 |
| | GRCN, $\mathcal{G}'=\mathcal{G}$ | 1e-3 | 5e-3 | 0.5 | 256 | 2 |
| | GRCN, $\mathcal{G}'=$ MLP | 1e-3 | 5e-3 | 0.5 | 256 | 2 |
| | IDGL | 1e-2 | 5e-4 | 0.5 | 512 | 2 |
| | IDGL, $\mathcal{G}'=\mathcal{G}$ | 1e-2 | 5e-4 | 0.5 | 512 | 2 |
| | IDGL, $\mathcal{G}'=$ MLP | 1e-2 | 5e-4 | 0.5 | 512 | 2 |
| | NodeFormer | 1e-2 | 5e-4 | 0.2 | 64 | 2 |
| | NodeFormer, $\mathcal{G}'=\mathcal{G}$ | 1e-2 | 5e-4 | 0.2 | 64 | 2 |
| | NodeFormer, $\mathcal{G}'=$ MLP | 1e-2 | 5e-4 | 0.2 | 64 | 2 |
| | GloGNN | 1e-2 | 5e-5 | 0.5 | 64 | 1 |
| | GloGNN, $\mathcal{G}'=\mathcal{G}$ | 1e-2 | 5e-5 | 0.5 | 64 | 1 |
| | GloGNN, $\mathcal{G}'=$ MLP | 1e-2 | 5e-5 | 0.5 | 64 | 1 |
| | WRGAT | 1e-2 | 1e-5 | 0.5 | 128 | 2 |
| | WRGAT, $\mathcal{G}'=\mathcal{G}$ | 1e-2 | 5e-5 | 0.5 | 128 | 2 |
| | WRGAT, $\mathcal{G}'=$ MLP | 1e-2 | 1e-5 | 0.5 | 128 | 2 |
| | WRGCN | 1e-2 | 1e-5 | 0.5 | 128 | 2 |
| | WRGCN, $\mathcal{G}'=\mathcal{G}$ | 1e-2 | 5e-5 | 0.5 | 128 | 2 |
| | WRGCN, $\mathcal{G}'=$ MLP | 1e-2 | 1e-5 | 0.5 | 128 | 2 |

| Dataset | Model | Learning Rate | Weight Decay | Dropout | Hidden Dim | Num of Layers |
|---|---|---|---|---|---|---|
| PubMed | GAug | 1e-2 | 5e-4 | 0.5 | 128 | 2 |
| | GAug, $\mathcal{G}'=\mathcal{G}$ | 1e-2 | 5e-4 | 0.5 | 128 | 2 |
| | GAug, $\mathcal{G}'=$ MLP | 1e-2 | 5e-4 | 0.5 | 128 | 2 |
| | GEN | 1e-3 | 5e-4 | 0.2 | 32 | 2 |
| | GEN, $\mathcal{G}'=\mathcal{G}$ | 1e-3 | 5e-4 | 0.2 | 32 | 2 |
| | GEN, $\mathcal{G}'=$ MLP | 1e-3 | 5e-4 | 0.2 | 32 | 2 |
| | GRCN | 1e-3 | 5e-3 | 0.5 | 32 | 2 |
| | GRCN, $\mathcal{G}'=\mathcal{G}$ | 1e-3 | 5e-3 | 0.5 | 32 | 2 |
| | GRCN, $\mathcal{G}'=$ MLP | 1e-3 | 5e-3 | 0.5 | 32 | 2 |
| | IDGL | 1e-2 | 5e-4 | 0.5 | 16 | 2 |
| | IDGL, $\mathcal{G}'=\mathcal{G}$ | 1e-2 | 5e-4 | 0.5 | 16 | 2 |
| | IDGL, $\mathcal{G}'=$ MLP | 1e-2 | 5e-4 | 0.5 | 16 | 2 |
| | NodeFormer | 1e-3 | 5e-4 | 0.2 | 64 | 2 |
| | NodeFormer, $\mathcal{G}'=\mathcal{G}$ | 1e-3 | 5e-4 | 0.2 | 64 | 2 |
| | NodeFormer, $\mathcal{G}'=$ MLP | 1e-3 | 5e-4 | 0.2 | 32 | 2 |
| | GloGNN | 1e-3 | 5e-5 | 0.7 | 64 | 3 |
| | GloGNN, $\mathcal{G}'=\mathcal{G}$ | 1e-3 | 5e-5 | 0.7 | 64 | 3 |
| | GloGNN, $\mathcal{G}'=$ MLP | 1e-3 | 5e-5 | 0.7 | 64 | 3 |
| | WRGAT | 1e-2 | 5e-5 | 0.5 | 64 | 2 |
| | WRGAT, $\mathcal{G}'=\mathcal{G}$ | 1e-2 | 1e-5 | 0.5 | 64 | 2 |
| | WRGAT, $\mathcal{G}'=$ MLP | 1e-2 | 5e-5 | 0.5 | 64 | 2 |
| | WRGCN | 1e-2 | 5e-5 | 0.5 | 64 | 2 |
| | WRGCN, $\mathcal{G}'=\mathcal{G}$ | 1e-2 | 5e-5 | 0.5 | 64 | 2 |
| | WRGCN, $\mathcal{G}'=$ MLP | 1e-2 | 5e-5 | 0.5 | 64 | 2 |

| Dataset | Model | Learning Rate | Weight Decay | Dropout | Hidden Dim | Num of Layers |
|---|---|---|---|---|---|---|
| Citeseer | GAug | 1e-4 | 5e-7 | 0.8 | 512 | 2 |
| | GAug, $\mathcal{G}'=\mathcal{G}$ | 1e-4 | 5e-7 | 0.8 | 512 | 2 |
| | GAug, $\mathcal{G}'=$ MLP | 1e-4 | 5e-7 | 0.8 | 512 | 2 |
| | GEN | 1e-2 | 5e-4 | 0.5 | 16 | 2 |
| | GEN, $\mathcal{G}'=\mathcal{G}$ | 1e-2 | 5e-4 | 0.5 | 16 | 2 |
| | GEN, $\mathcal{G}'=$ MLP | 1e-2 | 5e-4 | 0.5 | 16 | 2 |
| | GRCN | 1e-3 | 5e-3 | 0.8 | 512 | 3 |
| | GRCN, $\mathcal{G}'=\mathcal{G}$ | 1e-3 | 5e-3 | 0.8 | 512 | 3 |
| | GRCN, $\mathcal{G}'=$ MLP | 1e-2 | 5e-3 | 0.5 | 256 | 3 |
| | IDGL | 1e-2 | 5e-4 | 0.5 | 32 | 2 |
| | IDGL, $\mathcal{G}'=\mathcal{G}$ | 1e-3 | 5e-4 | 0.5 | 16 | 2 |
| | IDGL, $\mathcal{G}'=$ MLP | 1e-3 | 5e-4 | 0.5 | 16 | 2 |
| | NodeFormer | 1e-2 | 5e-4 | 0.2 | 64 | 2 |
| | NodeFormer, $\mathcal{G}'=\mathcal{G}$ | 1e-2 | 5e-4 | 0.2 | 64 | 2 |
| | NodeFormer, $\mathcal{G}'=$ MLP | 1e-2 | 5e-4 | 0.2 | 64 | 2 |
| | GloGNN | 1e-2 | 1e-5 | 0.7 | 64 | 2 |
| | GloGNN, $\mathcal{G}'=\mathcal{G}$ | 1e-2 | 1e-5 | 0.7 | 64 | 2 |
| | GloGNN, $\mathcal{G}'=$ MLP | 1e-2 | 1e-5 | 0.7 | 64 | 2 |
| | WRGAT | 1e-2 | 5e-5 | 0.5 | 128 | 2 |
| | WRGAT, $\mathcal{G}'=\mathcal{G}$ | 1e-2 | 5e-5 | 0.5 | 128 | 2 |
| | WRGAT, $\mathcal{G}'=$ MLP | 1e-2 | 5e-5 | 0.5 | 128 | 2 |
| | WRGCN | 1e-2 | 5e-5 | 0.3 | 128 | 2 |
| | WRGCN, $\mathcal{G}'=\mathcal{G}$ | 1e-2 | 5e-5 | 0.5 | 128 | 2 |
| | WRGCN, $\mathcal{G}'=$ MLP | 1e-2 | 1e-5 | 0.5 | 128 | 1 |

| Dataset | Model | Learning Rate | Weight Decay | Dropout | Hidden Dim | Num of Layers |
|---|---|---|---|---|---|---|
| Minesweeper | GAug | 1e-3 | 5e-6 | 0.8 | 256 | 3 |
| | GAug, $\mathcal{G}'=\mathcal{G}$ | 1e-3 | 5e-6 | 0.8 | 256 | 3 |
| | GAug, $\mathcal{G}'=$ MLP | 1e-3 | 5e-6 | 0.8 | 256 | 3 |
| | GEN | 1e-4 | 5e-6 | 0.8 | 256 | 3 |
| | GEN, $\mathcal{G}'=\mathcal{G}$ | 1e-4 | 5e-6 | 0.8 | 256 | 3 |
| | GEN, $\mathcal{G}'=$ MLP | 1e-4 | 5e-6 | 0.8 | 256 | 3 |
| | GRCN | 1e-3 | 5e-7 | 0.2 | 128 | 2 |
| | GRCN, $\mathcal{G}'=\mathcal{G}$ | 1e-3 | 5e-6 | 0.2 | 128 | 2 |
| | GRCN, $\mathcal{G}'=$ MLP | 1e-3 | 5e-6 | 0.2 | 128 | 2 |
| | IDGL | 1e-1 | 5e-6 | 0.2 | 128 | 3 |
| | IDGL, $\mathcal{G}'=\mathcal{G}$ | 1e-1 | 5e-6 | 0.2 | 128 | 3 |
| | IDGL, $\mathcal{G}'=$ MLP | 1e-1 | 5e-6 | 0.2 | 128 | 3 |
| | NodeFormer | 1e-2 | 5e-4 | 0.8 | 32 | 2 |
| | NodeFormer, $\mathcal{G}'=\mathcal{G}$ | 1e-2 | 5e-4 | 0.8 | 32 | 2 |
| | NodeFormer, $\mathcal{G}'=$ MLP | 1e-2 | 5e-4 | 0.8 | 32 | 2 |
| | GloGNN | 1e-2 | 5e-4 | 0.5 | 512 | 5 |
| | GloGNN, $\mathcal{G}'=\mathcal{G}$ | 1e-2 | 5e-4 | 0.5 | 512 | 5 |
| | GloGNN, $\mathcal{G}'=$ MLP | 1e-2 | 5e-4 | 0.5 | 512 | 5 |
| | WRGAT | 1e-2 | 5e-5 | 0.5 | 128 | 2 |
| | WRGAT, $\mathcal{G}'=\mathcal{G}$ | 1e-2 | 5e-5 | 0.5 | 128 | 2 |
| | WRGAT, $\mathcal{G}'=$ MLP | 1e-2 | 5e-5 | 0.5 | 128 | 2 |
| | WRGCN | 1e-2 | 5e-5 | 0.5 | 128 | 2 |
| | WRGCN, $\mathcal{G}'=\mathcal{G}$ | 1e-2 | 5e-5 | 0.5 | 128 | 2 |
| | WRGCN, $\mathcal{G}'=$ MLP | 1e-2 | 5e-5 | 0.5 | 128 | 2 |

| Dataset | Model | Learning Rate | Weight Decay | Dropout | Hidden Dim | Num of Layers |
|---|---|---|---|---|---|---|
| Roman-empire | GAug | 1e-1 | 5e-5 | 0.5 | 32 | 2 |
| | GAug, $\mathcal{G}'=\mathcal{G}$ | 1e-1 | 5e-5 | 0.5 | 32 | 2 |
| | GAug, $\mathcal{G}'=$ MLP | 1e-1 | 5e-5 | 0.5 | 32 | 2 |
| | GEN | 1e-2 | 5e-7 | 0.2 | 128 | 2 |
| | GEN, $\mathcal{G}'=\mathcal{G}$ | 1e-2 | 5e-7 | 0.2 | 128 | 2 |
| | GEN, $\mathcal{G}'=$ MLP | 1e-2 | 5e-7 | 0.2 | 128 | 2 |
| | GRCN | 1e-3 | 5e-5 | 0.5 | 128 | 2 |
| | GRCN, $\mathcal{G}'=\mathcal{G}$ | 1e-2 | 5e-5 | 0.5 | 128 | 2 |
| | GRCN, $\mathcal{G}'=$ MLP | 1e-2 | 5e-5 | 0.5 | 128 | 2 |
| | IDGL | 1e-1 | 5e-5 | 0.5 | 128 | 2 |
| | IDGL, $\mathcal{G}'=\mathcal{G}$ | 1e-1 | 5e-5 | 0.5 | 128 | 2 |
| | IDGL, $\mathcal{G}'=$ MLP | 1e-1 | 5e-5 | 0.5 | 128 | 2 |
| | NodeFormer | 1e-3 | 5e-6 | 0.2 | 128 | 3 |
| | NodeFormer, $\mathcal{G}'=\mathcal{G}$ | 1e-3 | 5e-6 | 0.2 | 128 | 3 |
| | NodeFormer, $\mathcal{G}'=$ MLP | 1e-3 | 5e-5 | 0.8 | 128 | 3 |
| | GloGNN | 1e-2 | 5e-5 | 0.7 | 128 | 3 |
| | GloGNN, $\mathcal{G}'=\mathcal{G}$ | 1e-2 | 5e-5 | 0.7 | 128 | 3 |
| | GloGNN, $\mathcal{G}'=$ MLP | 1e-2 | 5e-5 | 0.7 | 128 | 2 |
| | WRGAT | 1e-2 | 5e-5 | 0.5 | 128 | 2 |
| | WRGAT, $\mathcal{G}'=\mathcal{G}$ | 1e-2 | 1e-5 | 0.5 | 128 | 2 |
| | WRGAT, $\mathcal{G}'=$ MLP | 1e-2 | 5e-5 | 0.5 | 128 | 2 |
| | WRGCN | 1e-2 | 5e-5 | 0.5 | 128 | 2 |
| | WRGCN, $\mathcal{G}'=\mathcal{G}$ | 1e-2 | 5e-5 | 0.5 | 128 | 2 |
| | WRGCN, $\mathcal{G}'=$ MLP | 1e-2 | 5e-5 | 0.5 | 128 | 2 |

| Dataset | Model | Learning Rate | Weight Decay | Dropout | Hidden Dim | Num of Layers |
|---|---|---|---|---|---|---|
| Amazon-ratings | GAug | 1e-2 | 5e-7 | 0.2 | 128 | 2 |
| | GAug, $\mathcal{G}'=\mathcal{G}$ | 1e-2 | 5e-7 | 0.2 | 128 | 2 |
| | GAug, $\mathcal{G}'=$ MLP | 1e-2 | 5e-7 | 0.2 | 128 | 2 |
| | GEN | 1e-2 | 5e-7 | 0.2 | 128 | 2 |
| | GEN, $\mathcal{G}'=\mathcal{G}$ | 1e-2 | 5e-7 | 0.2 | 128 | 2 |
| | GEN, $\mathcal{G}'=$ MLP | 1e-2 | 5e-7 | 0.2 | 128 | 2 |
| | GRCN | 1e-3 | 5e-7 | 0.2 | 128 | 2 |
| | GRCN, $\mathcal{G}'=\mathcal{G}$ | 1e-2 | 5e-7 | 0.2 | 64 | 2 |
| | GRCN, $\mathcal{G}'=$ MLP | 1e-2 | 5e-7 | 0.2 | 64 | 2 |
| | IDGL | 1e-2 | 5e-7 | 0.2 | 128 | 2 |
| | IDGL, $\mathcal{G}'=\mathcal{G}$ | 1e-2 | 5e-7 | 0.2 | 128 | 2 |
| | IDGL, $\mathcal{G}'=$ MLP | 1e-2 | 5e-7 | 0.2 | 128 | 2 |
| | NodeFormer | 1e-4 | 5e-5 | 0.5 | 64 | 3 |
| | NodeFormer, $\mathcal{G}'=\mathcal{G}$ | 1e-4 | 5e-5 | 0.5 | 64 | 3 |
| | NodeFormer, $\mathcal{G}'=$ MLP | 1e-4 | 5e-5 | 0.5 | 64 | 3 |
| | GloGNN | 1e-2 | 5e-5 | 0.3 | 128 | 3 |
| | GloGNN, $\mathcal{G}'=\mathcal{G}$ | 1e-2 | 5e-5 | 0.3 | 128 | 3 |
| | GloGNN, $\mathcal{G}'=$ MLP | 1e-2 | 5e-5 | 0.3 | 128 | 3 |
| | WRGAT | 1e-2 | 5e-5 | 0.3 | 128 | 2 |
| | WRGAT, $\mathcal{G}'=\mathcal{G}$ | 1e-2 | 1e-5 | 0.3 | 128 | 2 |
| | WRGAT, $\mathcal{G}'=$ MLP | 1e-2 | 5e-5 | 0.3 | 128 | 2 |
| | WRGCN | 1e-2 | 5e-5 | 0.7 | 128 | 3 |
| | WRGCN, $\mathcal{G}'=\mathcal{G}$ | 1e-2 | 5e-5 | 0.7 | 128 | 3 |
| | WRGCN, $\mathcal{G}'=$ MLP | 1e-2 | 5e-5 | 0.7 | 128 | 3 |

| Dataset | Model | Learning Rate | Weight Decay | Dropout | Hidden Dim | Num of Layers |
|---|---|---|---|---|---|---|
| Questions | GAug | 1e-2 | 5e-4 | 0.5 | 64 | 3 |
| | GAug, $\mathcal{G}'=\mathcal{G}$ | 1e-2 | 5e-4 | 0.5 | 64 | 3 |
| | GAug, $\mathcal{G}'=$ MLP | 1e-2 | 5e-4 | 0.5 | 64 | 3 |
| | GEN | 1e-2 | 5e-7 | 0.2 | 256 | 2 |
| | GEN, $\mathcal{G}'=\mathcal{G}$ | 1e-2 | 5e-7 | 0.2 | 256 | 2 |
| | GEN, $\mathcal{G}'=$ MLP | 1e-2 | 5e-7 | 0.2 | 256 | 2 |
| | GRCN | 1e-2 | 5e-6 | 0.5 | 64 | 2 |
| | GRCN, $\mathcal{G}'=\mathcal{G}$ | 1e-2 | 5e-6 | 0.5 | 64 | 2 |
| | GRCN, $\mathcal{G}'=$ MLP | 1e-2 | 5e-6 | 0.5 | 64 | 2 |
| | IDGL | 1e-2 | 5e-7 | 0.2 | 128 | 2 |
| | IDGL, $\mathcal{G}'=\mathcal{G}$ | 1e-2 | 5e-7 | 0.2 | 128 | 2 |
| | IDGL, $\mathcal{G}'=$ MLP | 1e-2 | 5e-7 | 0.2 | 128 | 2 |
| | NodeFormer | 1e-4 | 5e-3 | 0.5 | 128 | 3 |
| | NodeFormer, $\mathcal{G}'=\mathcal{G}$ | 1e-4 | 5e-3 | 0.5 | 64 | 3 |
| | NodeFormer, $\mathcal{G}'=$ MLP | 1e-4 | 5e-3 | 0.5 | 64 | 3 |
| | GloGNN | 1e-2 | 5e-5 | 0.7 | 128 | 3 |
| | GloGNN, $\mathcal{G}'=\mathcal{G}$ | 1e-2 | 5e-5 | 0.7 | 128 | 3 |
| | GloGNN, $\mathcal{G}'=$ MLP | 1e-2 | 5e-5 | 0.7 | 128 | 3 |
| | WRGAT | 5e-3 | 5e-5 | 0.3 | 64 | 2 |
| | WRGAT, $\mathcal{G}'=\mathcal{G}$ | 5e-3 | 1e-5 | 0.3 | 64 | 2 |
| | WRGAT, $\mathcal{G}'=$ MLP | 5e-3 | 5e-5 | 0.3 | 64 | 2 |
| | WRGCN | 5e-3 | 5e-5 | 0.7 | 64 | 2 |
| | WRGCN, $\mathcal{G}'=\mathcal{G}$ | 5e-3 | 5e-5 | 0.7 | 64 | 2 |
| | WRGCN, $\mathcal{G}'=$ MLP | 5e-3 | 1e-5 | 0.7 | 64 | 2 |

| Dataset | Model | Learning Rate | Weight Decay | Dropout | Hidden Dim | Num of Layers |
|---|---|---|---|---|---|---|
| Tolokers | GAug | 1e-1 | 5e-5 | 0.5 | 32 | 2 |
| | GAug, $\mathcal{G}'=\mathcal{G}$ | 1e-1 | 5e-5 | 0.5 | 32 | 2 |
| | GAug, $\mathcal{G}'=$ MLP | 1e-1 | 5e-5 | 0.5 | 32 | 2 |
| | GEN | 1e-2 | 5e-5 | 0.2 | 128 | 2 |
| | GEN, $\mathcal{G}'=\mathcal{G}$ | 1e-2 | 5e-6 | 0.2 | 128 | 2 |
| | GEN, $\mathcal{G}'=$ MLP | 1e-2 | 5e-6 | 0.2 | 128 | 2 |
| | GRCN | 1e-2 | 5e-5 | 0.5 | 32 | 2 |
| | GRCN, $\mathcal{G}'=\mathcal{G}$ | 1e-2 | 5e-6 | 0.5 | 32 | 2 |
| | GRCN, $\mathcal{G}'=$ MLP | 1e-1 | 5e-6 | 0.5 | 64 | 2 |
| | IDGL | 1e-2 | 5e-4 | 0.5 | 64 | 2 |
| | IDGL, $\mathcal{G}'=\mathcal{G}$ | 1e-2 | 5e-4 | 0.5 | 64 | 2 |
| | IDGL, $\mathcal{G}'=$ MLP | 1e-2 | 5e-4 | 0.5 | 64 | 2 |
| | NodeFormer | 1e-2 | 5e-4 | 0.2 | 64 | 2 |
| | NodeFormer, $\mathcal{G}'=\mathcal{G}$ | 1e-2 | 5e-4 | 0.2 | 64 | 2 |
| | NodeFormer, $\mathcal{G}'=$ MLP | 1e-2 | 5e-4 | 0.2 | 64 | 2 |
| | GloGNN | 1e-2 | 5e-5 | 0.3 | 128 | 3 |
| | GloGNN, $\mathcal{G}'=\mathcal{G}$ | 1e-2 | 5e-5 | 0.3 | 128 | 3 |
| | GloGNN, $\mathcal{G}'=$ MLP | 1e-2 | 5e-5 | 0.3 | 128 | 3 |
| | WRGAT | 1e-2 | 5e-5 | 0.5 | 128 | 2 |
| | WRGAT, $\mathcal{G}'=\mathcal{G}$ | 1e-2 | 1e-5 | 0.5 | 128 | 2 |
| | WRGAT, $\mathcal{G}'=$ MLP | 1e-2 | 5e-5 | 0.5 | 128 | 2 |
| | WRGCN | 1e-2 | 5e-5 | 0.5 | 128 | 1 |
| | WRGCN, $\mathcal{G}'=\mathcal{G}$ | 1e-2 | 5e-5 | 0.5 | 128 | 2 |
| | WRGCN, $\mathcal{G}'=$ MLP | 1e-2 | 5e-5 | 0.5 | 128 | 2 |

"OOM" refers to "out of memory." The results demonstrate the following: (1) The average rank indicates that trainable GSL improves GNN performance on 5 out of 6 GNN backbones; (2) Although trainable GSL outperforms non-trainable GSL, it remains inferior to GNN backbones without GSL, indicating that GSL could be unnecessary in improving GNN performance on node classification.

## F.7 Performance on Graph classification

In addition to the node classification experiments, we further investigate whether GSL consistently improves GNN performance in graph classification. Specifically, we conduct ablation experiments by replacing the GSL graph with the original graph, following the methodology outlined in [26]. As shown in 8, removing GSL from 4 state-of-the-art GNNs, including ProGNN [15], GEN [45], GRCN [53], and IDGL [6], results in significantly reduced training time. At the same time, the GNN

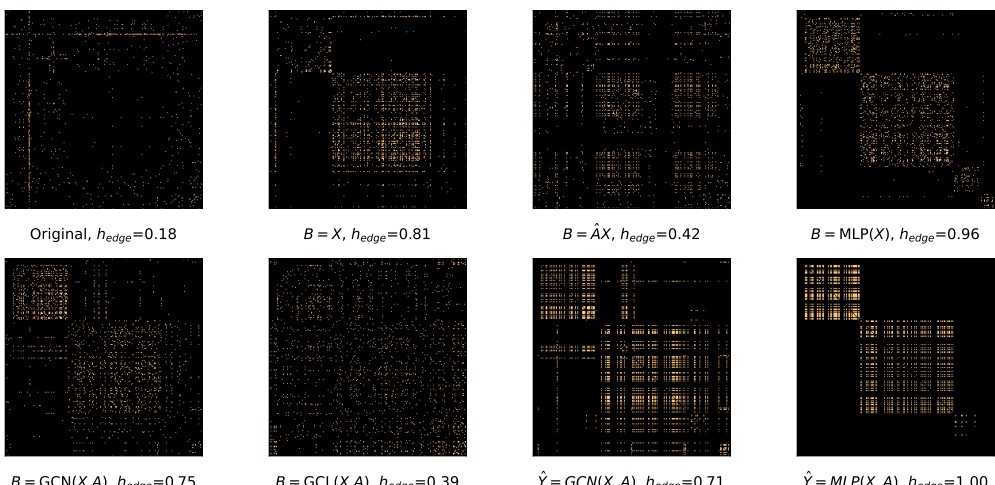

Figure 11: Visualization of original graph and reconstructed graphs on Wisconsin

Table 6: Performance of heterophily-oriented GNNs with GNN+GSL

| Model | Construct | Fusion | Param Sharing | Mines. | Roman. | Amazon. | Tolokers | Questions | Squirrel | Chameleon | Actor | Texas | Cornell | Wisconsin | Cora | CiteSeer | PubMed | Rank |
|---|---|---|---|---|---|---|---|---|---|---|---|---|---|---|---|---|---|---|
| MLP | - | - | - | 79.55±1.23 | 65.45±0.99 | 46.65±0.83 | 75.94±1.38 | 74.92±1.39 | 39.29±2.22 | 43.57±4.18 | 35.40±1.38 | 80.46±6.44 | 73.78±7.34 | 85.88±7.78 | 87.97±1.80 | 76.68±2.10 | 87.39±2.18 | 2.93 |
| ACMGNN | - | - | - | 90.56±1.03 | 84.86±0.73 | 52.07±1.72 | 84.41±1.12 | 77.72±1.59 | 41.53±2.43 | 44.65±4.43 | 34.86±1.22 | 82.62±5.97 | 75.68±8.99 | 87.65±7.15 | 88.23±1.81 | 76.63±2.34 | 89.37±0.56 | 1.21 |
| ACMGNN | cos-graph | $\{\mathcal{G}'\}$ | - | 47.36±3.47 | 60.97±0.76 | 41.50±0.75 | 70.21±1.51 | 67.32±1.37 | 38.12±1.92 | 39.90±3.64 | 33.43±0.95 | 59.80±6.99 | 59.46±8.35 | 71.57±6.68 | 77.47±2.41 | 73.68±0.97 | 87.19±0.38 | 7.64 |
| ACMGNN | cos-graph | $\{\mathcal{G},\mathcal{G}'\}$ | $\theta_1=\theta_2$ | 52.74±5.22 | 51.18±2.12 | 33.11±1.38 | 69.06±4.65 | 62.30±3.23 | 31.58±4.39 | 38.79±4.73 | 29.06±2.60 | 54.10±7.59 | 59.19±8.87 | 70.39±9.58 | 59.74±1.87 | 65.17±1.94 | 79.53±1.69 | 9.86 |
| ACMGNN | cos-graph | $\{\mathcal{G},\mathcal{G}'\}$ | $\theta_1\neq\theta_2$ | 87.46±1.02 | 74.63±0.76 | 49.35±0.58 | 81.63±0.87 | 73.84±1.41 | 38.54±1.89 | 41.16±4.18 | 34.23±0.98 | 67.67±5.97 | 70.00±5.90 | 80.78±5.21 | 80.83±1.84 | 73.43±1.47 | 88.98±0.47 | 3.64 |
| ACMGNN | cos-node | $\{\mathcal{G}'\}$ | - | 52.83±3.52 | 61.26±0.62 | 42.47±0.53 | 74.14±1.14 | 72.23±1.36 | 38.23±1.97 | 40.77±3.68 | 34.74±0.90 | 61.45±6.13 | 63.51±5.87 | 74.31±6.43 | 75.84±2.93 | 73.05±1.18 | 87.22±0.41 | 6.21 |
| ACMGNN | cos-node | $\{\mathcal{G},\mathcal{G}'\}$ | $\theta_1=\theta_2$ | 52.74±5.22 | 51.18±2.12 | 33.11±1.38 | 69.06±4.65 | 62.30±3.23 | 31.58±4.39 | 38.79±4.73 | 29.06±2.60 | 54.10±7.59 | 59.19±8.87 | 70.39±9.58 | 59.74±1.87 | 65.17±1.94 | 79.53±1.69 | 9.86 |
| ACMGNN | cos-node | $\{\mathcal{G},\mathcal{G}'\}$ | $\theta_1\neq\theta_2$ | 87.80±0.97 | 73.55±0.51 | 49.04±0.57 | 80.74±0.92 | 74.11±1.40 | 39.19±2.12 | 40.28±4.30 | 34.19±1.16 | 69.86±5.56 | 69.46±7.21 | 80.39±5.23 | 80.33±1.90 | 73.31±1.26 | 88.94±0.36 | 4.07 |
| ACMGNN | kNN | $\{\mathcal{G}'\}$ | - | 51.68±3.38 | 60.86±0.87 | 41.68±0.95 | 71.31±0.64 | 69.56±1.41 | 38.58±1.96 | 40.56±2.34 | 34.88±0.77 | 62.51±6.16 | 62.07±5.95 | 76.47±4.43 | 75.99±2.85 | 70.20±1.51 | 87.20±0.45 | 6.64 |
| ACMGNN | kNN | $\{\mathcal{G},\mathcal{G}'\}$ | $\theta_1=\theta_2$ | 52.74±5.22 | 51.18±2.12 | 33.11±1.38 | 69.06±4.65 | 62.30±3.23 | 31.58±4.39 | 38.79±4.73 | 29.06±2.60 | 54.10±7.59 | 59.19±8.87 | 70.39±9.58 | 59.74±1.87 | 65.17±1.94 | 79.53±1.69 | 9.86 |
| ACMGNN | kNN | $\{\mathcal{G},\mathcal{G}'\}$ | $\theta_1\neq\theta_2$ | 87.59±0.88 | 73.21±0.63 | 49.06±0.53 | 81.34±0.85 | 73.95±1.35 | 39.18±2.18 | 41.70±3.71 | 34.67±1.11 | 68.48±5.78 | 68.92±5.87 | 80.20±3.13 | 80.46±2.26 | 73.14±1.31 | 88.87±0.51 | 4.07 |
| MLP | - | - | - | 79.55±1.23 | 65.45±0.99 | 46.65±0.83 | 75.94±1.38 | 74.92±1.39 | 39.29±2.22 | 43.57±4.18 | 35.40±1.38 | 80.46±6.44 | 73.78±7.34 | 85.88±7.78 | 87.97±1.80 | 76.68±2.10 | 87.39±2.18 | 2.29 |
| MixHop | - | - | - | 90.10±5.59 | 81.70±0.89 | 50.95±0.71 | 84.56±1.19 | 77.66±1.24 | 41.22±2.66 | 43.11±4.73 | 33.59±1.23 | 72.54±8.98 | 62.43±9.54 | 75.88±8.27 | 87.76±1.94 | 76.51±1.93 | 89.42±0.81 | 1.86 |
| MixHop | cos-graph | $\{\mathcal{G}'\}$ | - | 64.75±4.59 | 51.83±0.53 | 41.47±2.00 | 68.78±1.94 | 71.45±1.38 | 37.75±2.41 | 37.79±2.10 | 31.77±1.75 | 55.72±6.39 | 60.27±5.85 | 70.20±4.60 | 84.42±1.35 | 74.20±0.83 | 88.74±0.29 | 8.21 |
| MixHop | cos-graph | $\{\mathcal{G},\mathcal{G}'\}$ | $\theta_1=\theta_2$ | 54.22±10.75 | 63.50±0.86 | 44.21±1.36 | 74.22±2.21 | 70.64±1.32 | 37.16±1.34 | 39.06±3.08 | 32.24±1.33 | 58.16±9.18 | 66.22±5.59 | 73.73±7.80 | 65.14±2.62 | 68.66±1.24 | 86.63±0.51 | 7.54 |
| MixHop | cos-graph | $\{\mathcal{G},\mathcal{G}'\}$ | $\theta_1\neq\theta_2$ | 84.71±1.19 | 55.41±1.63 | 43.37±0.75 | 74.41±1.33 | 69.63±2.03 | 37.64±2.19 | 38.71±4.36 | 31.73±1.77 | 61.13±7.96 | 61.35±7.10 | 75.29±6.00 | 85.42±1.21 | 74.57±1.34 | 88.16±0.46 | 6.50 |
| MixHop | cos-node | $\{\mathcal{G}'\}$ | - | 60.56±7.08 | 51.74±0.68 | 42.71±0.97 | 74.27±1.84 | 72.83±1.12 | 38.35±1.99 | 38.88±3.00 | 33.05±1.04 | 58.42±6.52 | 60.27±5.98 | 71.57±4.91 | 83.22±1.16 | 74.11±1.12 | 88.23±0.45 | 6.71 |
| MixHop | cos-node | $\{\mathcal{G},\mathcal{G}'\}$ | $\theta_1=\theta_2$ | 54.22±10.75 | 63.50±0.86 | 44.21±1.36 | 74.22±2.21 | 70.64±1.32 | 37.16±1.34 | 39.06±3.08 | 32.24±1.33 | 58.16±9.18 | 66.22±5.59 | 73.73±7.80 | 65.14±2.62 | 68.66±1.24 | 86.63±0.51 | 7.64 |
| MixHop | cos-node | $\{\mathcal{G},\mathcal{G}'\}$ | $\theta_1\neq\theta_2$ | 85.43±0.57 | 55.95±2.35 | 44.15±0.59 | 76.54±0.91 | 72.03±2.45 | 37.47±2.07 | 39.52±3.33 | 32.50±1.10 | 60.61±8.73 | 62.97±6.75 | 75.10±6.20 | 85.36±0.89 | 74.68±1.13 | 88.18±0.52 | 4.79 |
| MixHop | kNN | $\{\mathcal{G}'\}$ | - | 59.50±6.26 | 50.39±0.72 | 42.07±0.93 | 70.49±1.70 | 69.57±1.32 | 38.07±1.72 | 38.76±2.91 | 33.23±1.30 | 59.25±4.49 | 57.30±6.96 | 69.22±7.22 | 83.99±1.28 | 74.96±1.18 | 87.99±0.40 | 8.00 |
| MixHop | kNN | $\{\mathcal{G},\mathcal{G}'\}$ | $\theta_1=\theta_2$ | 54.22±10.75 | 63.50±0.86 | 44.21±1.36 | 74.22±2.21 | 70.64±1.32 | 37.16±1.34 | 39.06±3.08 | 32.24±1.33 | 58.16±9.18 | 66.22±5.59 | 73.73±7.80 | 65.14±2.62 | 68.66±1.24 | 86.63±0.51 | 7.54 |
| MixHop | kNN | $\{\mathcal{G},\mathcal{G}'\}$ | $\theta_1\neq\theta_2$ | 85.53±0.50 | 57.48±1.98 | 43.28±0.68 | 77.24±1.61 | 70.34±1.76 | 38.15±2.01 | 40.12±3.76 | 32.30±1.53 | 60.05±9.45 | 63.51±7.56 | 74.90±8.21 | 85.18±1.26 | 74.59±1.19 | 88.20±0.57 | 4.93 |

performance remains comparable to that of the GSL-enhanced counterparts. This suggests that GSL does not consistently enhance GNN performance in graph classification. Due to page limitations, we only tested a few methods in this paper. We believe it would be valuable to explore additional state-of-the-art methods, datasets, and theoretical justifications for the effectiveness of GSL in graph classification in future work.

## F.8 Robustness of GSL

We investigate the robustness of GSL with GNNs using 3 types of graph perturbation strategies:

- **Additive Feature Noise**: We randomly inject noise into node features, where the noise follows a normal distribution $N(0, \sigma^2)$. The level of noise is controlled by $\sigma$, taking values from the set $[0, 0.01, 0.02, 0.05, 0.1, 0.2, 0.5, 1.0, 2.0, 3.0, 10.0]$.

- **Edge Addition**: We randomly add edges to the graph structure, with the ratio of added edges proportional to the original number of edges, ranging from $r \in [0.0, 0.1, 0.2, 0.3, 0.4, 0.5, 0.6, 0.7, 0.8, 0.9]$.

- **Edge Removal**: We randomly remove edges from the graph structure, with the ratio of removed edges also proportional to the original number of edges, following the same range as in edge addition.

We then measure model performance through accuracy or AUC-ROC in node classification.

Figure 12 illustrates the differences in model performance between GNN baselines and their GSL-enhanced counterparts across additional datasets beyond those shown in Figure 4. Generally, the performance of GSL is comparable to or even worse than that of the GNN baselines for all three types of perturbed graphs. Notably, model performance is not consistently stable for structural

Table 7: Performance of GNNs with their counterparts of trainable GSL.

| Model | GSL Type | Mines. | Roman. | Amazon. | Tolokers | Questions | Cora | CiteSeer | PubMed | Rank |
|---|---|---|---|---|---|---|---|---|---|---|
| GCN | No GSL | **90.07±5.79** | **81.46±1.25** | **50.89±1.16** | **84.61±0.99** | **77.68±1.10** | 87.97±1.51 | 76.75±2.30 | **89.47±0.64** | 1.19 |
| | Trainable GSL | **90.07±0.58** | 78.76±0.46 | **50.89±0.65** | **84.61±0.65** | OOM | 84.92±1.51 | 74.89±1.13 | 88.66±0.45 | 2.31 |
| | Non-trainable GSL | 89.17±0.68 | 72.63±1.45 | 48.31±0.96 | 82.91±0.97 | 75.56±1.05 | 85.69±1.73 | 75.49±1.42 | 88.72±0.71 | 2.50 |
| SGC | No GSL | **83.45±4.47** | **78.04±0.69** | **51.38±0.68** | **84.88±1.13** | **77.39±1.23** | 88.10±1.89 | 77.52±2.20 | **89.39±0.62** | 1.19 |
| | Trainable GSL | **83.45±1.03** | 74.74±0.57 | **51.38±0.57** | **84.88±0.65** | OOM | 86.99±1.64 | 75.13±1.26 | 88.94±0.31 | 2.31 |
| | Non-trainable GSL | 79.03±3.76 | 67.84±1.87 | 47.93±0.94 | 78.09±1.84 | 75.46±1.43 | 87.47±1.86 | 76.36±1.27 | 89.37±0.41 | 2.50 |
| GraphSAGE | No GSL | 90.66±0.88 | **85.02±0.97** | 52.93±0.83 | **83.31±1.12** | 75.95±1.41 | 88.13±1.77 | 76.65±2.00 | 89.18±0.65 | 1.31 |
| | Trainable GSL | 90.66±0.58 | 82.54±0.60 | 52.93±0.59 | **83.31±0.50** | OOM | 83.48±1.69 | 74.18±1.02 | 88.67±0.39 | 2.44 |
| | Non-trainable GSL | **90.67±0.66** | 79.02±1.21 | 52.10±0.84 | 82.17±0.89 | 75.38±0.96 | 83.60±1.78 | 74.39±1.35 | 88.88±0.50 | 2.25 |
| GAT | No GSL | **90.41±1.34** | **84.51±0.84** | 52.00±2.84 | **84.37±0.96** | **77.78±1.27** | 88.02±1.92 | 76.77±2.02 | 89.21±0.67 | 1.19 |
| | Trainable GSL | **90.41±0.61** | 83.10±0.58 | 52.10±0.62 | 84.35±0.56 | OOM | 86.23±1.58 | 74.39±1.14 | 88.13±0.56 | 2.19 |
| | Non-trainable GSL | 89.96±0.79 | 77.23±1.63 | 49.79±0.72 | 82.78±0.95 | 76.67±1.13 | 86.97±1.75 | 75.20±1.55 | 87.97±0.51 | 2.62 |
| ACMGNN | No GSL | **90.56±1.03** | **84.86±0.73** | 52.07±1.72 | **84.41±1.12** | **77.72±1.59** | 88.23±1.81 | 76.63±2.34 | 89.37±0.56 | 1.06 |
| | Trainable GSL | **90.56±0.63** | 81.90±0.71 | 51.87±0.44 | 84.40±0.79 | OOM | 81.16±1.81 | 73.91±1.16 | 88.55±0.39 | 2.19 |
| | Non-trainable GSL | 87.46±1.02 | 74.63±0.76 | 49.35±0.58 | 81.63±0.87 | 73.84±1.41 | 80.83±1.84 | 73.43±1.47 | 88.98±0.47 | 2.75 |
| MixHop | No GSL | **90.10±5.59** | **81.70±0.89** | 50.95±0.71 | **84.56±1.19** | 77.66±1.24 | 87.76±1.94 | 76.51±1.93 | **89.42±0.81** | 1.12 |
| | Trainable GSL | **90.10±0.52** | 79.07±0.75 | 50.95±0.71 | 84.55±0.67 | OOM | 84.84±1.28 | 74.45±1.11 | 88.48±0.62 | 2.25 |
| | Non-trainable GSL | 85.43±0.57 | 55.95±2.35 | 44.15±0.59 | 76.54±0.91 | 72.03±2.45 | 85.36±0.89 | 74.68±1.13 | 88.18±0.52 | 2.62 |

Table 8: Ablation study of GSL-enhanced methods for graph classification.

| Model | Cora | | PubMed | | CiteSeer | |
|---|---|---|---|---|---|---|
| | AUC | Time | AUC | Time | Acc | Time |
| ProGNN | 76.28±0.52 | 959s | OOM | - | 67.14±0.23 | 1776s |
| ProGNN,w/o. GSL | 78.96±0.64 | 30s | 75.80±0.95 | 326s | 67.24±1.48 | 44s |
| GEN | 79.88 ± 0.93 | 219s | OOM | - | 66.98 ± 1.28 | 320s |
| GEN,w/o. GSL | 78.32 ± 1.21 | 3s | 76.94 ± 0.40 | 47s | 64.66 ± 1.46 | 3s |
| GRCN | 83.04 ± 0.33 | 56s | 74.55 ± 0.96 | 249s | 70.85 ± 0.87 | 113s |
| GRCN,w/o. GSL | 71.82 ± 0.61 | 9s | 74.18 ± 0.63 | 28s | 58.33 ± 0.17 | 24s |
| IDGL | 83.32 ± 0.59 | 144s | OOM | - | 70.57 ± 0.26 | 330s |
| IDGL,w/o. GSL | 83.32 ± 0.59 | 129s | OOM | - | 71.12 ± 0.31 | 401s |

perturbations in heterophilous graphs. We attribute this inconsistency to the non-informative nature of the structural information in these graphs, which leads to diminished responses to edge addition or removal. Despite this, GSL still fails to consistently outperform GNN baselines.

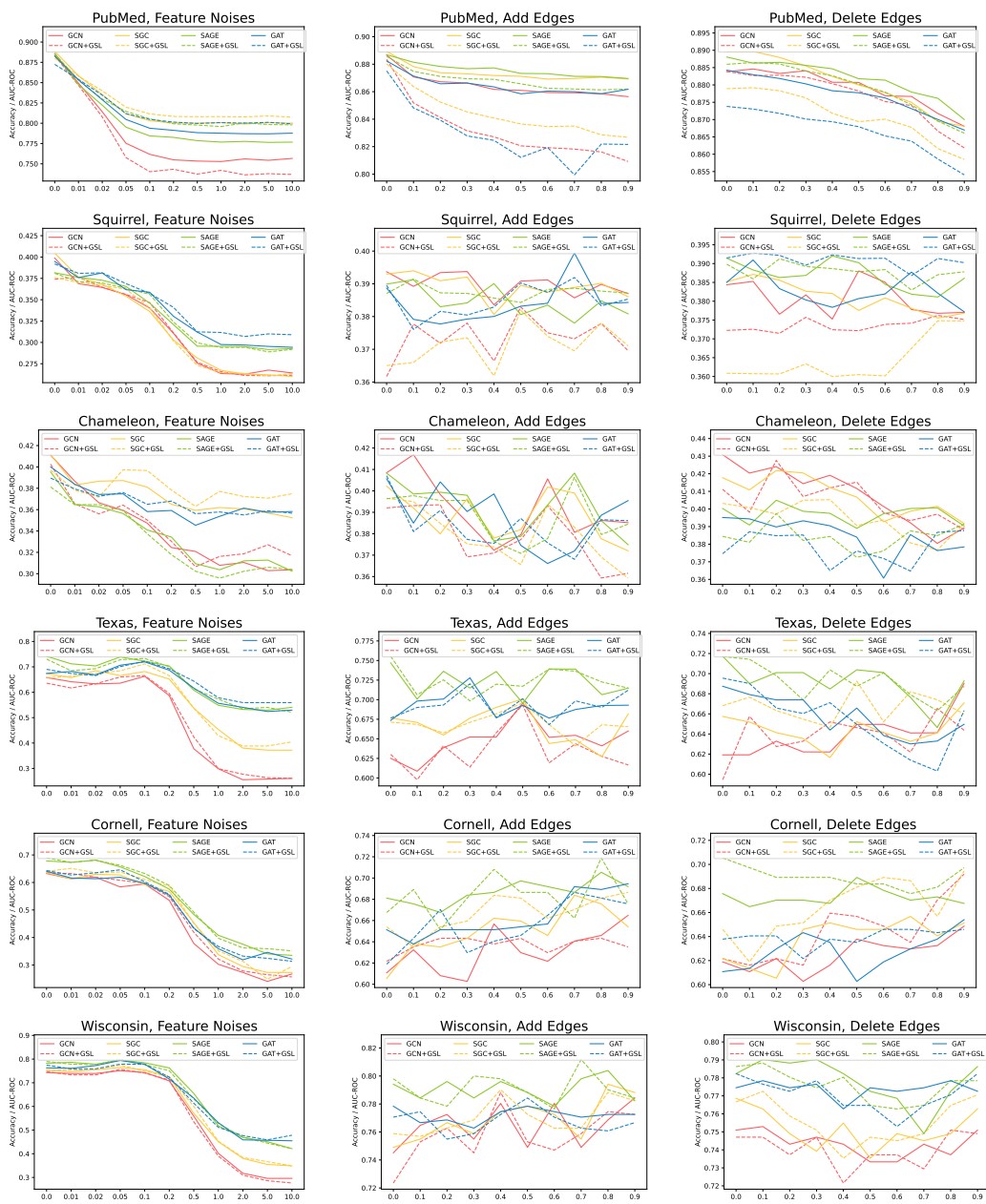

Figure 12: Response to feature noise, edge additions, and edge removals in GNN baselines and their GSL-enhanced counterparts.

