# OpenReview forum: "Disentangling and Re-evaluating The Effectiveness of Graph Structure Learning For GNNs"
_NeurIPS.cc/2025/Datasets_and_Benchmarks_Track — Submitted to NeurIPS 2025 Datasets and Benchmarks Track_

### Official Review · Reviewer_afkP · 2025-06-24

**Rating:** 4
**Confidence:** 4

**Summary:**

The paper disentangles the graph structure learning for GNNs into a unified framework with three steps: GSL bases, new graph construction, and view fusion. Besides, the paper states that it is the GSL base not graph construction and view fusion plays the most important role in enhancing the GNN performance. The empirical finding is also supported by the theoretical analysis.

**Additional Feedback:**

1. While the claim that GSL bases dominate performance is well-supported, this conclusion raises a fundamental conceptual question: If the GSL bases, essentially node representations, are the primary contributors to performance gains, does the problem reduce to general graph representation learning? In that case, the boundary between GSL and node representation learning becomes blurred, and the independent significance of GSL, particularly the graph construction component, is called into question.
2. C for number of classes in equation 4 should be defined in Theorem 4.1。
3. What is reference 1?

**Dataset Code Accessibility:**

Partly

**Dataset Code Comments:**

The paper mentions extensive experiments involving 450 GSL variants and provides detailed empirical results. However, code for replicating the analysis, especially the MI computation method in Section 4.2, is not included. Releasing this code is crucial for reproducibility, particularly given the complexity of MI estimation and GSL variant construction.

**Ethical Considerations:**

No, there are no or only very minor ethics concerns

**Final Justification:**

Authors resolve all my questions and I will keep my score positive.

**Limitations Weaknesses:**

1. The definition of successful new edge is a kind of strong assumption that intra-class connections are always desirable, which may oversimplify real-world scenarios, especially for heterophilous graphs.
2. The concept of GSL bases is kind of messy. The paper first says GSL bases are the processed node embeddings in the introduction section, then in Figure 2 and Section 4.1 the paper says the similarity of neighbor distribution is GSL bases. It would be better if there is an official definition on GSL bases.
3. The methodology for mutual information (MI) estimation is under-specified. How is the MI calculated and in practical use?

**Strengths Contributions:**

1. The paper is overall well-written and easy to follow.
2. The decomposition of GSL framework is reasonable and provides a clear and comprehensive understanding of the process.
3. The conclusion is very interesting and authors try to explain the empirical observations theoretically, which makes the conclusion persuasive and reliable.

---

> ### Author Rebuttal · Authors · 2025-07-31
>
> #### **R1: assumption of desirable intra-class connects in successful new edge**
>
> Thank you for your valuable feedback. The example presented in Figure 2 serves as an illustrative motivation, not a rigorous definition. Its purpose is to highlight the crucial importance of high-quality GSL bases and demonstrate why constructing entirely new graphs may not always be necessary. Following this motivation in Section 4.1, we provide further empirical studies in Section 4.2 and theoretical analysis in Section 4.3 to substantiate this claim, with the proof in Appendix C being free of such illustrative assumptions.
>
> Furthermore, prior research on heterophilous graphs [1,2] indicates that newly constructed graphs that enhance GNN performance consistently exhibit increased homophily, meaning a higher ratio of intra-class connections. This property can thus be generally used as a metric to assess the quality of a GSL graph.
>
> #### **R2: concept of GSL base**
>
> Thanks for your suggestion. We introduced the concept of GSL base, defined as pre-processed node embeddings used for new structure construction, in Section 3.1. In Lines 166-167 of Section 4.1, we used neighborhood distributions as GSL bases to illustrate examples of successful and unsuccessful GSL in Figure 2. We've now revised our paper to include a formal definition of GSL bases.
>
> #### **R3: calculation of mutual information**
>
> As mentioned in Lines 103-108, we measure mutual information (MI) using entropy estimation derived from k-nearest neighbors distances, following [3,4,5]. This numerical method for MI calculation is applied in Section 4.2. However, in Section 4.3, we don't explicitly calculate MI, as demonstrated by our proofs in Appendix C.
>
> #### **R4: additional feedback**
>
> 1. In this paper, we don't refute all types of GSL. As highlighted in Lines 15, 51, and 74, our primary focus is on similarity-based GSL. We agree with your understanding that this particular type of GSL is questionable. We attribute the reported superior performance of these previous similarity-based GSL methods over GNN baselines to an unfair setting: they incorporate additional input information that is absent in the GNN baselines. Because this additional, unacknowledged information can inflate performance metrics, it becomes imperative to re-evaluate the core mechanisms of GSL. Rather than simply stacking complex modules onto models, we aim to disentangle GSL and rethink its crucial components to reduce complexity and truly understand what makes it effective.
> 2. & 3. We have added the notation of C in Theorem 4.1 and fix reference 1 in our paper. Thank you very much.
>
> **References**
>
> [1] X. Li, R. Zhu, Y. Cheng, C. Shan, S. Luo, D. Li, and W. Qian. Finding global homophily in graph neural networks when meeting heterophily. In International Conference on Machine Learning, pages 13242–13256. PMLR, 2022.
>
> [2] Y. Zheng, J. Xu, and L. Chen. Learn from heterophily: Heterophilous information-enhanced graph neural network. arXiv preprint arXiv:2403.17351, 2024.
>
> [3] L. F. Kozachenko and N. N. Leonenko. Sample estimate of the entropy of a random vector. 381 Problemy Peredachi Informatsii, 23(2):9–16, 1987. 10
>
> [4] A. Kraskov, H. Stögbauer, and P. Grassberger. Estimating mutual information. Physical Review E—Statistical, Nonlinear, and Soft Matter Physics, 69(6):066138, 2004.
>
> [5] B. C. Ross. Mutual information between discrete and continuous data sets. PloS one, 9(2):e87357, 2014.

---

### Official Review · Reviewer_ySWB · 2025-06-25

**Rating:** 4
**Confidence:** 3

**Summary:**

This paper presents a comprehensive re-evaluation of Graph Structure Learning methods. The authors propose a three-step framework for decomposing GSL into: base generation, new graph construction, and view fusion. Their central claim is that most of the performance gains attributed to GSL do not come from the new graphs constructed (e.g. via similarity), but from the pretrained or enhanced node embeddings produced during the GSL base generation step. Theoretical analysis via mutual information and extensive experiments across 450 variants are showed to support this claim. The results show that many GSL-enhanced models perform comparably or worse than their corresponding GNN baselines when evaluated fairly using the same GSL bases. The paper concludes with a call for simplified GSL pipelines and reevaluation of where actual gains in GNN performance originate.

**Dataset Code Accessibility:**

Yes

**Ethical Considerations:**

No, there are no or only very minor ethics concerns

**Final Justification:**

I appreciate the authors’ responses for addressing my concerns. I think this work presents a good contribution.
While I had minor reservations initially, these were resolved during the discussion phase, and I recommend acceptance.

**Limitations Weaknesses:**

I note that the experiments are already comprehensive (with 450 variants) and supports the current claim. To suggest some possible future work.

- While the authors discuss computational overheads in Sec 4.4, there is little quantitative comparison of runtime/memory usage across GSL variants. This can be another add on to benefit practitioners.

- All experiments are on node classification tasks. I feel that other tasks such as link prediction, graph classification, etc. How structural information is used in those settings can be different from node classification.

Minor: I found the numbers in Table 1 are too small to read. The authors may want to present the results in a more readable way.

**Strengths Contributions:**

- The paper introduces a conceptually clean decomposition of GSL: base generation, graph construction, and view fusion. This helps better organize GSL methods, which often mixes different operations without isolating their individual contributions.

- The authors conduct a massive empirical study involving 450 GSL variants across 6 GNN backbones. The setup is also more rigorous and controls for unfair advantages, for example, using better input embeddings for GSL methods than for baselines.

- The use of mutual information bounds provides theoretical support for the empirical observations. Graph convolutions over newly constructed graphs do not necessarily increase the information content about labels compared to the original GSL bases .

- This paper challenges a prevailing assumption in the GSL literature that similarity-based graph construction is inherently beneficial. This work encourages the community to reevaluate GSL architectures and conduct fairer baselines.

---

> ### Author Rebuttal · Authors · 2025-07-31
>
> #### **R1: report computational usage**
>
> Thank you for your suggestion. We have included the runtime of GSL methods in Table 4. These results indicate that GSL significantly increases runtime compared to non-GSL counterparts. Given the comparable performance of non-GSL methods, we believe it is important to acknowledge that the performance gains observed in previous similarity-based GSL methods are largely attributable to the high quality of their input features.
>
> | Model              | Cora AUC   | Cora Time | Pubmed AUC | Pubmed Time | CiteSeer AUC | CiteSeer Time |
> | ------------------ | ---------- | --------- | ---------- | ----------- | ------------ | ------------- |
> | GAug*              | 81.73±0.66 | 20s       | 78.38±0.46 | 6s          | 72.79±0.86   | 10s           |
> | GAug, G′=G         | 81.73±0.38 | 7s        | 78.38±0.77 | 6s          | 72.79±0.18   | 2s            |
> | GAug, G′=MLP       | 78.90±0.00 | 3.2s      | 77.40±0.00 | 8.1s        | 72.91±0.62   | 9s            |
> | GEN*               | 81.66±0.91 | 214s      | 78.49±3.98 | 1384s       | 73.21±0.62   | 470s          |
> | GEN, G′=G          | 82.16±0.37 | 39s       | 80.49±0.13 | 114s        | 71.52±0.34   | 25s           |
> | GEN, G′=MLP        | 80.20±0.00 | 146s      | 66.80±0.00 | 192s        | 75.90±0.00   | 310s          |
> | GRCN*              | 84.61±0.34 | 13s       | 79.30±0.34 | 17s         | 72.34±0.34   | 20s           |
> | GRCN, G′=G         | 81.66±1.10 | 2s        | 79.35±0.26 | 3s          | 69.55±1.28   | 2s            |
> | GRCN, G′=MLP       | 79.40±0.00 | 1.3s      | 78.10±0.00 | 5s          | 71.40±0.00   | 4.2s          |
> | IDGL*              | 84.19±0.61 | 123s      | 82.78±0.44 | 146s        | 73.53±0.53   | 332s          |
> | IDGL, G′=G         | 82.43±0.45 | 13s       | 73.50±1.85 | 23s         | 73.13±0.49   | 36s           |
> | IDGL, G′=MLP       | 83.20±0.00 | 6.6s      | 79.20±0.00 | 13s         | 72.60±0.00   | 13.9s         |
> | NodeFormer*        | 78.81±1.21 | 213s      | 78.38±1.94 | -           | 70.39±2.04   | 219s          |
> | NodeFormer, G′=G   | 77.01±1.99 | 152s      | OOM        | -           | 70.22±0.13   | 139s          |
> | NodeFormer, G′=MLP | 78.80±0.00 | 8s        | 76.30±0.00 | 127s        | 72.80±0.00   | 15s           |
> | GloGNN*            | 78.87±1.66 | 66.6s     | 78.90±0.26 | 18.2s       | 71.95±1.90   | 21.8s         |
> | GloGNN, G′=G       | 73.49±0.01 | 5.1s      | 87.62±0.20 | 14.4s       | 72.72±0.08   | 21.2s         |
> | GloGNN, G′=MLP     | 74.83±0.12 | 2.8s      | 87.64±0.27 | 10.2s       | 72.09±1.81   | 13.8s         |
> | WRGAT              | 84.28±1.52 | 19.5s     | 88.82±0.50 | 421.6s      | 73.50±1.41   | 22.1s         |
> | WRGAT, G′=G        | 83.48±1.48 | 3.9s      | 88.92±0.43 | 26.5s       | 73.21±1.90   | 4.7s          |
> | WRGAT, G′=MLP      | 77.99±1.10 | 2.3s      | 80.27±6.23 | 3.9s        | 65.28±2.11   | 4.5s          |
> | WRGCN              | 88.30±1.46 | 23.7s     | OOM        | -           | 74.74±1.60   | 54.2s         |
> | WRGCN, G′=G        | 88.32±0.79 | 3.9s      | 89.26±0.45 | 19.4s       | 74.4±1.51    | 10.5s         |
> | WRGCN, G′=MLP      | 70.00±3.59 | 2.2s      | 67.29±2.49 | 9.9s        | 70.84±1.36   | 4.1s          |
>
> #### **R2: more graph tasks**
>
> Thanks for your feedback. The main focus of our paper is node classification as it's a widely accepted benchmark for quantifying GNN performance in graph representation learning [1, 2]. We understand the value of demonstrating the generalizability of our observations.
>
> As you suggested, we're keen to expand our evaluation. We're considering adding another setting or task to explore if our findings hold true in different contexts.
>
> As for link prediction, we believe applying graph structure learning to this task might be redundant. Graph structure learning itself can be viewed as a form of link prediction, where optimized or rule-based methods infer a suitable graph structure. Applying structure learning and then performing link prediction seems counterintuitive. To our knowledge, this approach isn't common within the current landscape of graph learning research.
>
> As for graph classification, we conduct experiments on GCN, GAT, and GraphSAGE on datasets of MUTAG, ENZYMES, PTOTEINS, NCI1, and NCI109 as shown in the table below. While GSL showed slight improvements on the ENZYMES and PROTEINS datasets, we generally didn't find significant evidence of a consistent performance boost from GSL on all datasets in this task. This result is consistent as our experiments on node classification.
>
> | Model              | MUTAG | ENZYMES | PROTEINS | NCI1  | NCI109 |
> | ------------------ | ----- | ------- | -------- | ----- | ------ |
> | GCN w/o. GSL       | 71.39 | 21.20   | 73.31    | 69.80 | 67.69  |
> | GCN w. GSL         | 71.33 | 22.99   | 74.43    | 68.66 | 66.52  |
> | GAT w/o. GSL       | 73.48 | 21.56   | 72.30    | 68.31 | 68.20  |
> | GAT w. GSL         | 71.95 | 21.66   | 74.28    | 67.39 | 67.31  |
> | GraphSAGE w/o. GSL | 67.71 | 25.77   | 72.10    | 68.88 | 65.93  |
> | GraphSAGE w. GSL   | 67.42 | 26.27   | 72.86    | 67.58 | 66.42  |
>
> #### **R3: better presentation of Table 1**
>
> Thank you. We have revised the paper to improve readability by increasing the font size of the results. We appreciate your valuable feedback.
>
> **References**
>
> [1] Khoshraftar S, An A. A survey on graph representation learning methods[J]. ACM Transactions on Intelligent Systems and Technology, 2024, 15(1): 1-55.
>
> [2] Ju W, Fang Z, Gu Y, et al. A comprehensive survey on deep graph representation learning[J]. Neural Networks, 2024: 106207.

---

### Official Review · Reviewer_X7Ua · 2025-07-01

**Rating:** 4
**Confidence:** 4

**Summary:**

This submission critically examines the role of Graph Structure Learning (GSL) in enhancing GNNs. The authors propose a novel framework that decomposes GSL into three key steps: (1) ​​GSL bases generation​​ (processed node embeddings), (2) ​​new graph construction​​, and (3) ​​view fusion​​. Through extensive empirical and theoretical analysis, the paper challenges the conventional belief that similarity-based graph construction significantly improves GNN performance.

**Dataset Code Accessibility:**

Partly

**Ethical Considerations:**

No, there are no or only very minor ethics concerns

**Final Justification:**

The authors' responses provided new experimental results and explanations, which to some extent solved the questions. However, I believe that the author can still provide further detailed explanations and analyses in terms of graph construction.

**Limitations Weaknesses:**

1. The study focuses exclusively on ​​node classification​​, leaving open questions about GSL’s role in other tasks (e.g., link prediction, graph classification) or dynamic graphs.
2. The paper argues that GSL’s benefits primarily come from ​​pretrained GSL bases​​, dismissing graph construction as negligible. However, this may not hold in low-data or highly noisy settings.
3. Similar to the second point, if it doesn't make sense to reconstruct graph structures, does graph structure learning make sense? Without graph structure reconstruction, does that just increase the epcoh of training?
4. The paper critiques similarity-based GSL but does not thoroughly compare against ​​non-similarity-based methods​​ (e.g., diffusion-based approaches).
5. Some approaches based on GSL should be mentioned, they focus on aspects such as fairness, robustness, etc., which are also relevant for GSL. For example: 1. Self-guided robust graph structure refinement. 2. BAB-GSL: Using Bayesian influence with attention mechanism to optimize graph structure in basic views. 3. Learning fair representations via rebalancing graph structure. 4. FairWire: Fair graph generation

**Strengths Contributions:**

1. The paper rigorously questions the necessity of similarity-based GSL, a widely adopted approach, showing that its benefits are often overstated or misattributed. This could reshape how researchers design and evaluate GNNs, shifting focus toward improving node representations (GSL bases) rather than graph construction.

2.  Proposes a novel decomposition of GSL into three steps, providing a clearer lens to analyze its components.

---

> ### Author Rebuttal · Authors · 2025-07-31
>
> #### **R1: more graph tasks**
>
> Thanks for your feedback. In our current submission, we focused on node classification as it's a widely accepted benchmark for quantifying GNN performance in graph representation learning [5,6]. We understand the value of demonstrating the generalizability of our observations. As you suggested, we're keen to expand our evaluation. We're considering adding another setting or task to explore if our findings hold true in different contexts.
>
> As for link prediction, we believe applying graph structure learning to this task might be redundant. Graph structure learning itself can be viewed as a form of link prediction, where optimized or rule-based methods infer a suitable graph structure. Applying structure learning and then performing link prediction seems counterintuitive. To our knowledge, this approach isn't common within the current landscape of graph learning research.
>
> As for dynamic graphs, we acknowledge this as a compelling and important area for future work. Our current framework is primarily designed for static graphs, but extending GSL to effectively handle temporal graph dynamics is a promising direction we plan to explore.
>
> As for graph classification, we conduct experiments on GCN, GAT, and GraphSAGE on datasets of MUTAG, ENZYMES, PTOTEINS, NCI1, and NCI109 as shown in the table below. While GSL showed slight improvements on the ENZYMES and PROTEINS datasets, we generally didn't find significant evidence of a consistent performance boost from GSL in this task. This result is consistent as our experiments on node classification.
>
> | Model              | MUTAG | ENZYMES | PROTEINS | NCI1  | NCI109 |
> | ------------------ | ----- | ------- | -------- | ----- | ------ |
> | GCN w/o. GSL       | 71.39 | 21.20   | 73.31    | 69.80 | 67.69  |
> | GCN w. GSL         | 71.33 | 22.99   | 74.43    | 68.66 | 66.52  |
> | GAT w/o. GSL       | 73.48 | 21.56   | 72.30    | 68.31 | 68.20  |
> | GAT w. GSL         | 71.95 | 21.66   | 74.28    | 67.39 | 67.31  |
> | GraphSAGE w/o. GSL | 67.71 | 25.77   | 72.10    | 68.88 | 65.93  |
> | GraphSAGE w. GSL   | 67.42 | 26.27   | 72.86    | 67.58 | 66.42  |
>
> #### **R2: settings of low-data or highly noisy**
>
> Thank you for your valuable suggestion. While our primary experiments focus on supervised node classification, we agree that these settings are also crucial for GSL In our submission, we presented initial findings on three types of noise in Figure 12. Here, we offer further details and analysis. Our results generally indicate that GSL-enhanced GNNs demonstrate improved robustness against higher ratios of noise in graph structures. However, GSL appears to be less effective in mitigating feature noise. It will be interesting to empirically and theoretically investigate the robustness of GSL-enhanced models in the future.
>
> | Noise Type     | Dataset  | Use GSL  | 0%    | 10%   | 20%   | 30%   | 40%   | 50%   |
> | -------------- | -------- | -------- | ----- | ----- | ----- | ----- | ----- | ----- |
> | Add Edges      | CiteSeer | w/o. GSL | 72.34 | 71.72 | 70.83 | 69.65 | 68.52 | 67.75 |
> |                |          | w. GSL   | 72.79 | 70.97 | 71.88 | 69.86 | 68.66 | 70.48 |
> |                | Cora     | w/o. GSL | 81.73 | 78.65 | 78.18 | 77.98 | 74.33 | 74.53 |
> |                |          | w. GSL   | 82.4  | 79.87 | 79.05 | 78.83 | 77.54 | 77.16 |
> | Delete Edges   | CiteSeer | w/o. GSL | 72.34 | 68.85 | 67.12 | 62.87 | 65.03 | 62.22 |
> |                |          | w. GSL   | 72.79 | 71.62 | 68.08 | 63.96 | 63.68 | 62.08 |
> |                | Cora     | w/o. GSL | 81.73 | 76.83 | 74.07 | 70.53 | 68.67 | 67.7  |
> |                |          | w. GSL   | 82.43 | 79.14 | 77.86 | 73.54 | 72.54 | 70.88 |
> | Feature Noises | CiteSeer | w/o. GSL | 72.34 | 69.83 | 69.37 | 69.72 | 67.24 | 65.87 |
> |                |          | w. GSL   | 72.79 | 71.14 | 71.27 | 69.9  | 67.51 | 67.98 |
> |                | Cora     | w/o. GSL | 81.73 | 78.96 | 80.8  | 78.86 | 77.84 | 76.87 |
> |                |          | w. GSL   | 82.48 | 53.87 | 51.81 | 51.2  | 48.66 | 54.41 |
>
> We further conduct an ablation study on the performance of GSL-enhanced methods. As shown in table below, the performance of these GSL-enhanced methods is comparable to their counterparts without GSL when there are 20 labels per class. However, as we decrease the supervision ratio (such as in scenarios with 5 or 3 labels per class) the GSL-enhanced methods demonstrate improved performance compared to those without GSL. These results indicate that GSL could enhance GNN performance in settings with low supervision. Thanks for your suggestions. We have revised the paper to include this valuable finding. We believe it is interesting to investigate why GSL works beyond general supervised settings.
>
> | Dataset  | Use GSL  | 20 labels | 10 labels | 5 labels | 3 labels |
> | -------- | -------- | --------- | --------- | -------- | -------- |
> | PubMed   | w/o. GSL | 79.03     | 72.4      | 67.97    | 63.21    |
> |          | w. GSL   | 79.27     | 75.92     | 70.5     | 66.12    |
> | CiteSeer | w/o. GSL | 71.7      | 63.33     | 57.47    | 51.73    |
> |          | w. GSL   | 71.72     | 67.36     | 58.65    | 54.59    |
> | Cora     | w/o. GSL | 81.43     | 73.53     | 65.76    | 60.34    |
> |          | w. GSL   | 82.14     | 75.64     | 68.6     | 65.9     |
>
> #### **R3: does graph structure learning still make sense?**
>
> Our primary contribution lies in disentangling GSL into three key components: GSL base construction, new graph construction, and view fusion. Through both theoretical analysis (Section 4) and empirical studies (Section 5), we demonstrate that GNN performance is predominantly influenced by the quality of GSL bases, rather than the explicit construction of new graphs. This implies that high-quality GSL bases are sufficient to provide informative node representations, even without explicitly reconstructing the graph. As for your question, without graph structure construction, we could explore alternative input mechanisms, particularly pre-trained embeddings, to enhance GNN performance. As shown in the first subfigure of Figure 5, the choice of input features can significantly impact GNN performance.
>
> #### **R4: comparison with non-similarity-based methods**
>
> First, the comparison with non-similarity-based methods is not our main goal in this paper. As shown in Line 2, 51, 70, we specify our main scope to similarity-based GSL. We also implement 450 variants of similarity-based GSL in Table 1. Second, we also empirically explore how our conclusion can generalize to non-similarity-based methods. As detailed in Table 4 of our submission, we conducted experiments on a range of established methods that not solely rely on similarity-based methods, including GAug, GEN, GRCN, IDGL, NodeFormer, and GloGNN. The results show that removing GSL does not diminish model performance; in fact, it is often comparable to or even exceeds the original results. Recognizing the rapid proliferation of GSL methods, our choice of non-similarity-based approaches is constrained by the practicalities of time and the targeted scope of this paper. Thanks for your feedback.
>
> #### **R5: mention other GSL approaches**
>
> Thank you for highlighting these important works. We agree that investigating various aspects of GSL, including robustness and fairness, is necessary.
>
> SG-GSR [1] improves GNN robustness by dynamically refining graph structures through edge augmentation based on homophily, feature smoothness, and structural proximity, making models more resilient to perturbations. BAB-GSL [2] approximates ideal graph structures in multi-view settings by combining Bayesian inference with attention mechanisms, enhancing accuracy and robustness in node relationships. SRGNN [3] promotes fairness by rebalancing graph structures using gradient constraints and structural edits, reducing bias in node representations. FairWire [4] tackles structural bias in graph generation by introducing a fairness regularizer for generative models, enabling the creation of fairer synthetic graphs without sacrificing utility.
>
> While these cited works primarily focus on the robustness [1, 2] or fairness [3, 4] of GSL, our study specifically investigates the performance of similarity-based GSL. We acknowledge the value of deeper investigations into GSL across these diverse areas and **have revised our paper to include citations to these relevant works**. Thank you for your valuable feedback.
>
> **References**
>
> [1] In, Yeonjun, et al. "Self-guided robust graph structure refinement." *Proceedings of the ACM Web Conference 2024*. 2024.
>
> [2] Liu, Zhaowei, et al. "BAB-GSL: Using Bayesian influence with attention mechanism to optimize graph structure in basic views." *Neural Networks* 181 (2025): 106785.
>
> [3] Zhang, Guixian, et al. "Learning fair representations via rebalancing graph structure." *Information Processing & Management* 61.1 (2024): 103570.
>
> [4] Kose, Oyku, and Yanning Shen. "FairWire: Fair graph generation." *Advances in Neural Information Processing Systems* 37 (2024): 124451-124478.
>
> [5] Khoshraftar S, An A. A survey on graph representation learning methods[J]. ACM Transactions on Intelligent Systems and Technology, 2024, 15(1): 1-55.
>
> [6] Ju W, Fang Z, Gu Y, et al. A comprehensive survey on deep graph representation learning[J]. Neural Networks, 2024: 106207.

---

> > ### Comment · Reviewer_X7Ua · 2025-08-05
> >
> > Thanks to the authors' responses, which to a certain extent solved my questions. I am willing to increase the score to 4.

---

### Official Review · Reviewer_64Hi · 2025-07-02

**Rating:** 4
**Confidence:** 4

**Summary:**

The paper proposes a three-stage taxonomy of GSL (bases, structure, and fusion), proves an information-theoretic upper-bound indicating that similarity-based graph construction cannot add label information, and benchmark-tests 450 GSL variants across 14 datasets. The empirical results suggest that performance gains are almost entirely attributable to richer node bases rather than to the learned graph itself.

**Dataset Code Accessibility:**

Yes

**Ethical Considerations:**

No, there are no or only very minor ethics concerns

**Final Justification:**

The authors responded to my questions, and I am satisfied with them.

**Limitations Weaknesses:**

1. All experiments are node/graph classification. Regression, link prediction, graph‑level property prediction, and inductive settings are left unexplored, while being important tasks for the graph machine learning community. A short discussion appears in the conclusion but no empirical evidence is given.


2. Proposition 4.2 relies on k‑NN entropy estimators that are known to be noisy in high dimensions. The bound is therefore qualitative and less practical.

3. Edge and feature perturbations follow Li et al. (2023) but do not include adversarial structure attacks, which are a common GSL use case.

4. The phrase “trainable GSL improves GNN performance on 5/6 backbones” in appendix F.6 appears contradictory to the main claim and should be re‑phrased to avoid confusion. Can the authors explain that ?

5. The study focuses on k-NN / cosine graphs. It omits edge-rewiring families such as Diffusion Rewiring (Topping et al., 2022), DIGL, GRAND (Chamberlain et al., 2021) and PTDNet (Bo et al., 2021) that modify topology without similarity metrics, as well as multiple newer publication from recent NeruIPS, ICML and ICLR that build on the concept of graph rewiring. These are conceptually closest to “structure” in the taxonomy.

6. Similar to point 5, a proper comparison with graph transformers should be made. The authors mention NodeFormer, however there are already many types of Graph Transformers, which are also in many cases scalable like PolyNormer (ICLR 24) and others, that should be compared with and considered as a methodology to connect the graph, i.e. for GSL.

Minor:

Citation [1] in the paper is broken (missing the actual citations text).

**Strengths Contributions:**

1. The work addresses a rapidly expanding sub‑field (GSL) and provides the first systematic evidence that its perceived benefits may be overstated. The negative result is important for the community because it can steer future research away from needless graph reconstruction efforts.

2. The disentangling framework and the information‑theoretic upper bound in section 4 offer a fresh perspective.

3. Thorough and extensive experiments, with 14 datasets (homophily & heterophily), four perturbation settings, ablations over bases, construction schemes and fusion states.

4. The paper is well structured and easy to follow.

---

> ### Author Rebuttal · Authors · 2025-07-31
>
> ##### **R1. more graph tasks on link prediction and graph classification tasks, and inductive settings**
>
> Thanks for your feedback. The main focus of our paper is node classification, which has been widely used to quantify GNN performance in graph representation learning [1,2]. We'd like to add additional settings or tasks, as you suggested, to explore if our observations and conclusions can be generalized.
>
> However, for link prediction task, we don't find it quite necessary to apply graph structure learning for it. As Graph structure learning itself can be considered a type of link prediction task, where it uses rule-based or optimized methods to find a reasonable graph structure. These two tasks looks too similar and to the best of our knowledge, no such work uses graph structure learning for the task of link prediction.
>
> As for graph classification, we further investigated whether GSL consistently improves GNN performance. The results below show that GSL performs slightly better on the ENZYMES and PROTEINS datasets. However, generally, there is no clear evidence to show significant improvement from GSL in all datasets.
>
> | Model              | MUTAG | ENZYMES | PROTEINS | NCI1  | NCI109 |
> | ------------------ | ----- | ------- | -------- | ----- | ------ |
> | GCN w/o. GSL       | 71.39 | 21.20   | 73.31    | 69.80 | 67.69  |
> | GCN w. GSL         | 71.33 | 22.99   | 74.43    | 68.66 | 66.52  |
> | GAT w/o. GSL       | 73.48 | 21.56   | 72.30    | 68.31 | 68.20  |
> | GAT w. GSL         | 71.95 | 21.66   | 74.28    | 67.39 | 67.31  |
> | GraphSAGE w/o. GSL | 67.71 | 25.77   | 72.10    | 68.88 | 65.93  |
> | GraphSAGE w. GSL   | 67.42 | 26.27   | 72.86    | 67.58 | 66.42  |
>
> For inductive node classification, we follow [7] by creating disjoint training, validation, and test subgraphs from random node samples and their neighborhoods. Models are trained on training subgraphs, tuned on validation, and evaluated on unseen test subgraphs to simulate the inductive setting. The results below are also similar to what we observed in Table 1. Under the same search space of GSL bases, GNNs with GSL do not significantly outperform GNNs without GSL. Therefore, our conclusion still holds for the inductive setting.
>
> | Model              | Cora  | Citeseer | Pubmed | Chameleon | Squirrel | Actor | Texas | Cornell | Wisconsin |
> | ------------------ | ----- | -------- | ------ | --------- | -------- | ----- | ----- | ------- | --------- |
> | GCN w/o. GSL       | 64.17 | 62.46    | 48.73  | 67.53     | 35.11    | 39.83 | 74.22 | 62.41   | 64.38     |
> | GCN w. GSL         | 63.91 | 61.10    | 46.95  | 67.87     | 35.90    | 40.39 | 72.57 | 60.59   | 64.78     |
> | GraphSAGE w/o. GSL | 63.73 | 61.13    | 47.89  | 67.91     | 33.52    | 39.56 | 74.00 | 61.86   | 63.83     |
> | GraphSAGE w. GSL   | 63.55 | 60.95    | 48.11  | 67.41     | 34.37    | 39.63 | 72.11 | 60.41   | 63.92     |
>
>
> ##### **R2: noises in k-NN entropy estimators**
>
> Proposition 4.2 demonstrates a theoretical upper bound: the mutual information between node labels ($Y$) and similarity-based Graph Structure Learning (GSL) aggregated bases ($B'$) is bounded by the mutual information between node labels and the original bases ($B$), i.e., $I(Y;B') \le I(Y;B)$.
>
> While calculating the exact mutual information typically requires an estimator (like a kNN estimator), which might be noisy as you point out, it's important to clarify that this particular analysis is purely theoretical and not for practical use. **We do not conduct any explicitly calculation of the mutual information for any practical use, we just demonstrate the relation in the inequality with MI**. Instead, as detailed in Line 1027 of Appendix C, we leverage the data processing inequality [3] to prove $I(Y;B'_i) \le I(Y;B_i)$. This approach allows us to establish the theoretical bound without the need for noisy estimations.
>
> ##### **R3: adversarial structure attacks**
>
> We systematically evaluate how GSL affects the performance of GNNs under a range of perturbations, e.g. feature noise, edge additions, and edge removals, as shown in Figure 12. To broaden the scope, we further incorporated adversarial structure attacks as you suggest, following the approach in [4].
>
> Consistently across all tested perturbation types, including the newly added adversarial structure attacks, our results below indicate that the performance of GCN with GSL is slightly better than GCN without GSL. This reinforces our earlier observations regarding feature noise, edge additions, and edge removals.
>
> | Use GSL      | Dataset  | 0%    | 10%   | 20%   | 30%   | 40%   | 50%   |
> | ------------ | -------- | ----- | ----- | ----- | ----- | ----- | ----- |
> | GCN w/o. GSL | CiteSeer | 73.12 | 67.34 | 62.18 | 58.77 | 51.36 | 43.02 |
> | GCN w. GSL   | CiteSeer | 73.21 | 68.09 | 63.88 | 59.12 | 52.01 | 45.37 |
> | GCN w/o. GSL | Cora     | 82.02 | 76.44 | 69.12 | 62.31 | 54.67 | 46.98 |
> | GCN w. GSL   | Cora     | 82.77 | 76.13 | 71.22 | 63.01 | 56.09 | 47.01 |
>
> While **our goal in this work is to investigate the direct impact of GSL on GNN performance (only accuracy)**, we recognize the significant and growing importance of robustness in GNNs. Therefore, we believe that a dedicated and in-depth study into the robustness of GSL itself against various adversarial attacks and noisy environments would be a valuable direction for future research.
>
> ##### **R4: unclear phrase in appendix F.6**
>
> In Appendix F.6, what we are going to show is "trainable GSL outperforms non-trainable GSL" instead of "trainable GSL outperforms no GSL". Therefore, it does not contradict to our main claim. Thanks for your careful review and we will clarify the claim to "trainable GSL is better than non-trainable GSL on 5/6 backbones".
>
> ##### **R5: GSL methods without similarity metrics**
>
> As stated in Lines 1-3 of our paper, this work primarily focuses on the investigation of similarity-based GSL. Our key contributions encompass a novel GSL framework, theoretical analysis, and the comprehensive implementation of 450 distinct GSL variants.
>
> To broaden our scope, we also evaluated non-similarity-based methods like GAug, GEN, GRCN, IDGL, NodeFormer, GloGNN, WRGAT, and WRGCN (Table 4). GloGNN, in particular, directly optimizes the graph structure without similarity measures. Results across multiple GNN backbones (GCN, SGC, GraphSAGE, GAT, ACMGNN, MixHop in Table 7) show that these methods generally offer limited improvement over non-GSL baselines. This reinforces our conclusion that GSL, especially similarity-based, yields mixed gains and should not be assumed universally effective.
>
> We recognize that the field of GSL is rapidly evolving with numerous new methods, like the ones you have listed; however, given the practical constraints of time and the primary scope of this paper, our selection of non-similarity-based methods was necessarily focused on the above mentioned methods. We will add results for your suggested methods in the later version.
>
> ##### **R6: comparison with more types of graph transformers**
>
> We appreciate your suggestion. Our focus on similarity-based GSL led us to test 450 GSL variants (Table 1) and 8 learnable GSL methods, including NodeFormer, to ensure generalizability. To strengthen our evaluation, we added Graph Transformer Networks (GTN) [5] and Polynormer [6], using the same GSL base search space as Table 1. Under these consistent conditions, GSL did not consistently boost the performance of transformer models across datasets. This further supports our main claim: the critical design factor lies in how GSL bases are constructed, rather than in the newly generated graphs themselves.
>
> | Method               | Squirrel | Chameleon | Actor | Texas | Cornell | Wisconsin | Cora  | CiteSeer | PubMed |
> | -------------------- | -------- | --------- | ----- | ----- | ------- | --------- | ----- | -------- | ------ |
> | GTN (w/o GSL)        | 40.82    | 43.18     | 34.81 | 79.35 | 69.19   | 84.71     | 87.89 | 76.80    | 89.26  |
> | GTN (w/ GSL)         | 37.44    | 37.99     | 32.82 | 78.98 | 65.43   | 83.53     | 86.36 | 75.71    | 88.79  |
> | Polynormer (w/o GSL) | 41.24    | 42.81     | 35.01 | 79.34 | 73.24   | 84.51     | 88.04 | 77.10    | 89.09  |
> | Polynormer (w/ GSL)  | 37.83    | 39.18     | 32.59 | 78.97 | 73.27   | 83.10     | 86.17 | 75.53    | 88.74  |
> | MLP                  | 39.29    | 43.57     | 35.40 | 80.46 | 73.78   | 85.88     | 87.97 | 76.68    | 87.39  |
>
> Thanks for your review. We have revised our paper accordingly and include your suggested settings to make our study more comprehensive.
>
> **Reference**
>
> [1] Khoshraftar S, An A. A survey on graph representation learning methods[J]. ACM Transactions on Intelligent Systems and Technology, 2024, 15(1): 1-55.
>
> [2] Ju W, Fang Z, Gu Y, et al. A comprehensive survey on deep graph representation learning[J]. Neural Networks, 2024: 106207.
>
> [3] N. J. Beaudry and R. Renner. An intuitive proof of the data processing inequality. Quantum Info. Comput., 12(5–6):432–441, May 2012.
>
> [4] Z. Li, X. Sun, Y. Luo, Y. Zhu, D. Chen, Y. Luo, X. Zhou, Q. Liu, S. Wu, L. Wang, and J. X. Yu. GSLB: the graph structure learning benchmark. In A. Oh, T. Naumann, A. Globerson, K. Saenko, M. Hardt, and S. Levine, editors, Advances in Neural Information Processing Systems 36: Annual Conference on Neural Information Processing Systems 2023, NeurIPS 2023, New Orleans, LA, USA, December 10 - 16, 2023, 2023.
>
> [5] Yun S, Jeong M, Kim R, et al. Graph transformer networks[J]. Advances in neural information processing systems, 2019, 32.
>
> [6] Deng C, Yue Z, Zhang Z. Polynormer: Polynomial-expressive graph transformer in linear time[J]. The Twelfth International Conference on Learning Representations 2024.
>
> [7] Hamilton, W. L., Ying, Z., & Leskovec, J. (2017). Inductive Representation Learning on Large Graphs. In Advances in Neural Information Processing Systems

---

> > ### Comment · Reviewer_64Hi · 2025-08-04
> >
> > Thank you for the rebuttal. As you have addressed my comments and added the requested experiments, I think that the paper deserves an adjusted score, and I will revise my score from 3 to 4.

---

> > ### Author Response · Authors · 2025-08-04
> > **Thanks for your feedback**
> >
> > We appreciate your helpful feedback and raising your score. Your comments really help us to improve the manuscript, and we will keep polishing our paper. Thanks!

---

### Comment · Area_Chair_XHet · 2025-08-01
**Author-Reviewer Discussions (July 31 - Aug 6)**

Dear Reviewers,
Thank you for providing timely your review report. While Author-Reviewer discussions already started from July 31 to Aug 6, can you please carefully read all the other reviews and the author responses, acknowledge that you have read the author response to your questions/concerns, and check whether you want to raise any further  questions/concerns at your early convenience, so that there is time for back and forth discussion with the authors within the time window, Thank you very much for your time, effort, and contribution to the organization of NeurIPS 20205, best wishes, AC

---

### Comment · Area_Chair_XHet · 2025-08-05
**Deadline for Author-Reviewer discussions**

Dear Reviewers,
While the deadline, August 6, is tomorrow and approaching fast, can you please act quickly with the given time window: acknowledge the author response, examine the responses received and decide whether there are any further concerns/comments. Such author-review discussions play a crucial role in dispelling any concerns you may have and deciding your final rating of the paper, thank you very much for your dedication to the organization of NeurIPS 2025, best wishes, AC

---

### Note · Authors · 2025-08-12

Dear ACs, SACs, PCs and Reviewers,

We would like to thank all of you for your time and efforts to help us improve the paper. We are glad and grateful that all reviewers have finalized their ratings to be positive and have shown interests in our results and analysis on graph structure learning (GSL).

The comments and feedback provided by the reviewers are constructive and straightforward, enabling us to refine some details and clarify some statements. We will include all the discussions and promised updates into the final version.

Our paper introduces a novel and comprehensive framework that decomposes GSL into three distinct components, i.e. base representation generation, graph construction, and view fusion. Through both theoretical analysis and extensive experiments, we demonstrate that the performance gains of similarity-based GSL mainly depend on the quality of the pre-trained bases, rather than the graph construction methods. We provide new perspective to assess GSL and we believe that this contribution is of significant importance  at the prestigious NeurIPS. Thanks.

Best,

Authors

---

### Decision · Program_Chairs · 2025-09-18

**Decision:**

Reject

**Comment:**

This paper has proposed a novel and comprehensive framework for the evaluation of the similarity based graph structure learning. This framework decomposes the evaluation into three components: base generation, graph construction and view fusion. Both the empirical and theoretically analysis show that the base generation and representation plays a more crucial role than the graph construction for example.

Four reports have been received. Various comments and concerns have been raised about evaluation protocols and metrics, k-NN entropy estimation, and definitions of some terminologies. During the Reviewer-Author discussions, more explanations, clarifications, and experimental results have been provided. Overall, even though the reviewers were satisfied with the responses provided and the proposed work may reveal a novel avenue for further research and evaluation for graph based learning, various tasks are expected to be explored in depth and width, such as graph level property prediction, and some conclusions drawn may require further investigation, such as whether the GSL would improve the performance of GNNs.